# Loss of colonic fidelity enables multilineage plasticity and metastasis

Patrizia Cammareri[1,2], Michela Raponi[1,2,3], Yourae Hong[4], Caroline V. Billard[1,2], Nat Peckett[1,2], Yujia Zhu[1,2], Fausto D. Velez-Bravo[4], Nicholas T. Younger[1,5], Donnchadh S. Dunican[5], Sebastian Ö.-G. Pohl[1,2], Aslihan Bastem Akan[1,2], Nora J. Doleschall[1,2], John Falconer[3], Mark White[3,6], Jean Quinn[6], Kathryn Pennel[6], Roberta Garau[1,2], Sudhir B. Malla[3,7], Philip D. Dunne[3,7], Richard R. Meehan[5], Owen J. Sansom[3,6], Joanne Edwards[6], Malcolm G. Dunlop[1,2,5], Farhat V. N. Din[1,2], Sabine Tejpar[4], Colin W. Steele[3,6,8] & Kevin B. Myant[1,2✉]

Cancer cell plasticity enables the acquisition of new phenotypic features and is implicated as a major driver of metastatic progression[1,2]. Metastasis occurs mostly in the absence of additional genetic alterations[3–5], which suggests that epigenetic mechanisms are important[6]. However, they remain poorly defined. Here we identify the chromatin-remodelling enzyme ATRX as a key regulator of colonic lineage fidelity and metastasis in colorectal cancer. *Atrx* loss promotes tumour invasion and metastasis, concomitant with a loss of colonic epithelial identity and the emergence of highly plastic mesenchymal and squamous-like cell states. Combined analysis of chromatin accessibility and enhancer mapping identified impairment of activity of the colonic lineage-specifying transcription factor HNF4A as a key mediator of these observed phenotypes. We identify squamous-like cells in human patient samples and a squamous-like expression signature that correlates with aggressive disease and poor patient prognosis. Collectively, our study defines the epigenetic maintenance of colonic epithelial identity by ATRX and HNF4A as suppressors of lineage plasticity and metastasis in colorectal cancer.

Metastasis poses a substantial challenge in cancer management because it significantly affects patient prognosis and response to therapy[7,8]. The dissemination and spread of cancer cells to distant organs is associated with increased cellular heterogeneity and phenotypic plasticity that mostly occurs in the absence of additional genetic alterations[3–5]. This finding indicates that additional layers of regulation, particularly epigenetic alterations, are responsible for driving the phenotypic changes required for metastasis[6]. Epigenetic modifications, including DNA methylation, post-translational histone modifications and chromatin compaction, collectively influence gene expression. Epigenetic modification is typically involved in the maintenance of cellular identity. However, changes in epigenetic modification patterns can influence lineage determination, thereby providing cells with a heightened degree of plasticity. For example, widespread reprogramming of chromatin modification has been reported in pancreatic cancer metastasis, and alterations in chromatin accessibility have been identified across multiple tumour state transitions, including metastasis[6,9]. Together, these results suggest that epigenetic alterations can enhance plasticity to enable cancer cells to undergo phenotypic transitions that facilitate invasion and colonization at distant sites[10].

Epigenetic modifying enzymes are among the most mutated families of genes in cancer[11,12]. However, comprehensive studies investigating the specific impact of mutated epigenetic modifiers on cellular plasticity and metastasis are limited[13]. Here we investigated the function of the chromatin-remodelling helicase ATRX in colorectal cancer (CRC). We show that loss of *Atrx* promotes metastasis and is associated with the emergence of highly plastic, mesenchymal and squamous-like cell states. Transcriptional analysis identifies loss of expression of genes associated with colonic epithelial identity, which suggests that loss of lineage fidelity is a key step that controls metastatic progression. Mechanistically, *Atrx* loss leads to a loss of chromatin accessibility and enhancer activity at targets of the colonic-specifying transcription factor (TF) HNF4A, which we functionally identify as a key mediator of colonic lineage fidelity. In patient samples, expression of a squamous-like gene expression signature is associated with aggressive, metaplastic CRC subtypes and predicts poor patient outcomes. Together, our data indicate that loss of colonic epithelial identity and acquisition of highly plastic squamous-like cell states are key drivers of CRC metastasis.

## *Atrx* loss induces CRC metastasis

Analysis of the International Cancer Genome Consortium database revealed a prevalence of mutations in epigenetic regulators in CRC,

[1]Institute of Genetics and Cancer, The University of Edinburgh, Western General Hospital, Edinburgh, UK. [2]Cancer Research UK Scotland Centre, Institute of Genetics and Cancer, The University of Edinburgh, Western General Hospital, Edinburgh, UK. [3]Cancer Research UK Scotland Institute, Glasgow, UK. [4]Molecular Digestive Oncology, Department of Oncology, Katholieke Universiteit Leuven, Leuven, Belgium. [5]MRC Human Genetics Unit, Institute of Genetics and Cancer, The University of Edinburgh, Western General Hospital, Edinburgh, UK. [6]School of Cancer Sciences, University of Glasgow, Glasgow, UK. [7]The Patrick G. Johnston Centre for Cancer Research, Queen's University Belfast, Belfast, UK. [8]University Department of Surgery, Glasgow Royal Infirmary, Glasgow, UK. ✉e-mail: kevin.myant@ed.ac.uk

including *ATRX*, what was mutated in around 7% of samples (Extended Data Fig. 1a). Immunohistochemistry (IHC) analysis of a tissue microarray (TMA) of CRC cases revealed that loss of ATRX expression is associated with late-stage, metastatic disease (Extended Data Fig. 1b–d). Consistent with this finding, *ATRX* mutation is more prevalent in the highly aggressive CRIS-B transcriptional CRC subtype[14], in which it correlated with poor prognosis (Extended Data Fig. 1e,f). Together, these data suggest that loss of *ATRX* function has a role in promoting aggressive disease and metastasis.

To investigate this possibility, we used CRISPR–Cas9 genome editing to disrupt *Atrx* in a mouse *Apc*$^{fl/fl}$*Kras*$^{G12D}$*Trp53*$^{fl/fl}$ (AKP) CRC organoid line[15]. AKP organoids have low metastatic potential, which facilitates analyses of the effects of additional perturbations on this sytem[15]. An analysis of pan-cancer mutational data indicated a preponderance of stop codon mutations in the 5′ end of the SNF2 and helicase chromatin-remodelling domains of *ATRX* (in particular, R1426*) (Extended Data Fig. 1g). Therefore, we used a single guide RNA (sgRNA) targeted to this region to knock out *Atrx* in AKP organoids (AKP *Atrx*$^{KO}$) (Extended Data Fig. 1g,h). An AKP control line was also generated using a non-targeting gRNA to control for effects of transduction and Cas9 expression.

We first investigated the effects of *Atrx* loss on CRC metastasis. Tail-vein inoculation or intrasplenic transplantation of dissociated AKP *Atrx*$^{KO}$ organoids led to a significant increase in lung and liver metastatic burden, respectively, compared with AKP controls (Fig. 1a–h). Increased metastatic potential was confirmed with an independent clonal *Atrx*$^{KO}$ organoid line (AKP *Atrx*$^{KO2}$), which also displayed an increase in lung metastasis after tail-vein injection (Extended Data Fig. 1h–l). To determine whether this increase was associated with changes in primary tumour phenotypes, we subcutaneously transplanted the same organoid lines into mice. Although there was no difference in tumour size (Extended Data Fig. 2a,b), histological analyses uncovered considerable differences in tumour histology. AKP tumours had a classical glandular morphology, whereas AKP *Atrx*$^{KO}$ tumours displayed areas of poor differentiation and evidence of tumour cells invading into the surrounding stroma (Extended Data Fig. 2c–e). AKP control tumour cells were predominantly positive for the expression of the epithelial marker E-cadherin, whereas *Atrx* loss led to consistent loss of E-cadherin positivity at invasive regions (Extended Data Fig. 2f,g). Transcriptional and protein analyses also identified an induction of mesenchymal markers in AKP *Atrx*$^{KO}$ tumours (Extended Data Fig. 2h–j). This phenotype, indicative of epithelial-to-mesenchymal transition (EMT), was further investigated in vitro. TGFβ can act as an inducer of EMT and is associated with aggressive, late-stage CRC[14]. Treatment of AKP *Atrx*$^{KO}$ organoids with TGFβ led to the emergence of spreading, spindle-like cells reminiscent of mesenchymal cells and the induction of EMT marker expression compared with controls (Fig. 1i–k). This effect was confirmed with the AKP *Atrx*$^{KO2}$ clonal line (Extended Data Fig. 2k–l) and was observed across a range of TGFβ concentrations but not in response to other EMT inducers such as TNF or IFNγ (Extended Data Fig. 3a–c). This result suggests that *Atrx* deletion leads to a high level of sensitivity to TGFβ-driven EMT induction. We also analysed the response of these organoid lines to epigenetic modifying drugs to determine whether *Atrx* deletion sensitizes cells to such agents. Treatment with inhibitors of both BET (JQ1) and HDAC (FK228) impaired cell viability in both lines, with no increased sensitivity in *Atrx*$^{KO}$ cells (Extended Data Fig. 3d,e).

To determine whether *Atrx* loss promotes metastatic dissemination from primary tumours, mice were implanted with AKP control or AKP *Atrx*$^{KO}$ organoids into the colonic submucosa. Although no differences in overall survival or primary tumour growth were observed, AKP *Atrx*$^{KO}$ tumours led to significantly more metastases than controls, with metastases observed in the liver, lymph nodes and diaphragm (Extended Data Fig. 3f–k). Moreover, analyses of primary tumour histology demonstrated loss of glandular morphology in AKP *Atrx*$^{KO}$ tumours (Extended

Data Fig. 3k–m). To further extend these findings, we deleted *Atrx* in the *Braf*$^{V600E}$*Trp53*$^{fl/fl}$*Notch*$^{ICD}$ (BPN) mouse CRC model (Extended Data Fig. 4a,b). This *Braf*$^{V600E}$-driven model is transcriptionally distinct from the AKP model and resembles the CMS4 CRC subtype[16]. Similar to the AKP model, *Atrx* deletion increased the induction of TGFβ-mediated EMT (Extended Data Fig. 4c–f). After orthotopic transplantation, BPN *Atrx*$^{KO}$ tumours completely lost their glandular morphology and exhibited a poorly differentiated phenotype (Extended Data Fig. 4g,h). *Atrx* deletion also led to metastatic progression in about 25% of mice (Extended Data Fig. 4i,j). Together, these data demonstrate that *Atrx* loss in multiple mouse CRC models promotes an aggressive tumour phenotype associated with sensitivity to EMT induction, tumour invasion and metastatic progression.

## *Atrx* protects colonic epithelial identity

To mechanistically define how *Atrx* loss mediates tumour metastasis, we performed RNA sequencing (RNA-seq) analysis of AKP control and AKP *Atrx*$^{KO}$ organoid lines treated or untreated with TGFβ. Consistent with the observed EMT phenotype, gene set enrichment analysis (GSEA) identified significant enrichment of multiple EMT-related signatures in TGFβ-treated AKP *Atrx*$^{KO}$ organoids, which was confirmed by quantitative PCR with reverse transcription (RT–qPCR) (Extended Data Fig. 4k–l and Supplementary Table 1). Untreated organoids did not show the same global enrichment for EMT signatures or EMT marker expression after TGFβ treatment (Fig. 2a, Extended Data Fig. 4l and Supplementary Table 2). However, untreated AKP *Atrx*$^{KO}$ organoids expressed several genes that are expressed in mesenchymal cells, such as *Twist1*, *Itga5*, *Tfap2c* and *Irx2*. This result suggests that despite being insufficient to induce EMT, *Atrx* loss leads to induction of a partial mesenchymal-like phenotype (Fig. 2a and Extended Data Fig. 4l). Notably, *Atrx* loss precipitated a discernible lack of expression of gene signatures associated with colonic epithelial identity (Extended Data Fig. 4m); for example, targets of the colon-specifying TFs HNF4A, CDX1 and CDX2, and genes more broadly associated with colonic function, such as *Aqp1*, *Cftr* and *Tff3* (Fig. 2a and Extended Data Fig. 4n). Loss of expression of key colonic TFs was also seen in the BPN model (Extended Data Fig. 4o). We expanded our analysis using TissueEnrich, which not only confirmed the loss of colonic epithelial gene expression but also revealed a shift towards the expression of genes typically associated with squamous tissues, specifically the oesophagus, skin and adipose tissue (Fig. 2b). Such genes included highly specific markers of squamous tissues, such as *Ly6d*, *Krt5*, *Cav1*, *Cdkn1c* and *Elf5*, and adipose-specific genes encoding molecules involved in lipid catabolism, storage and localization, such as *Plin5*, *Acacb*, *Cidea* and *Dgat2* (Fig. 2b, Extended Data Fig. 4p and Supplementary Table 3). Among this lineage reprogramming effect, the expression of more broadly expressed epithelial markers, such as *Epcam*, *Cdh1* and *Cldn4*, was maintained (Fig. 2a). The preservation of these more ubiquitous epithelial markers suggests that there is a selective loss of colonic lineage specification, whereas broader epithelial characteristics are maintained.

To further investigate changes in cell state induced by *Atrx* loss, we carried out single-cell RNA-seq of AKP control and AKP *Atrx*$^{KO}$ organoids. Harmony integration and uniform manifold approximation and projection (UMAP) revealed clear separation between the two conditions, which indicated that *Atrx* loss induces considerable changes in the cell transcriptional state (Fig. 2c). Clustering analysis identified 18 cell clusters (Fig. 2d and Extended Data Fig. 5a and Supplementary Table 4), for which we mapped expression of key lineage markers. Consistent with the bulk RNA-seq results, *Hnf4a* and *Cdx1* expression was lost in *Atrx*$^{KO}$ cells, whereas *Twist1* expression was increased (Fig. 2e). Notably, *Epcam* expression was maintained, which reinforced the idea that broad epithelial identity is maintained despite extensive lineage reprogramming.

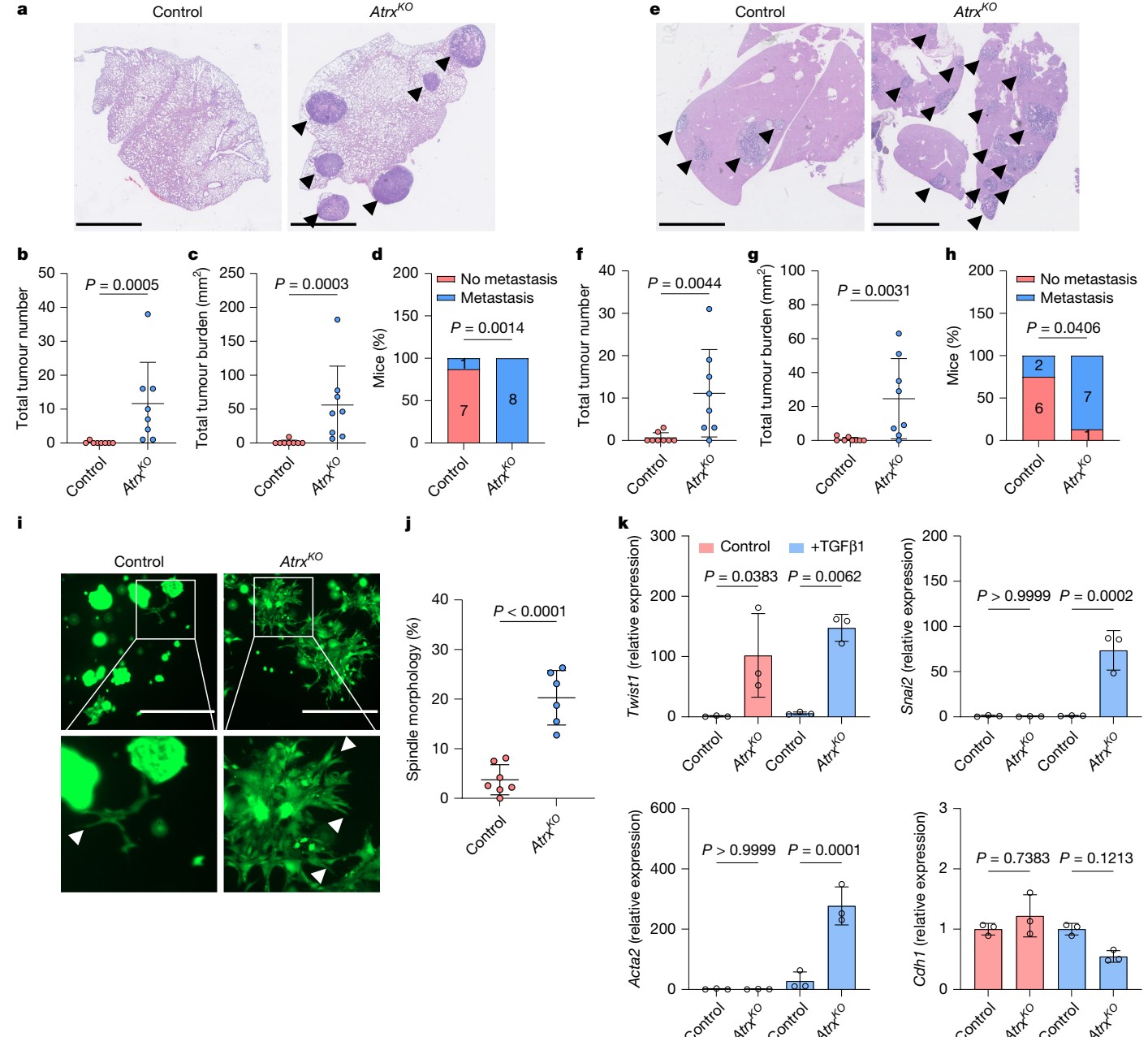

**Fig. 1 | *Atrx* loss promotes metastasis. a**, Representative images of lung metastases (stained with haematoxylin and eosin) from mice injected with AKP control or AKP *Atrx^KO^* organoids through the tail vein. Metastatic nodules are indicated with black arrowheads. **b**, Quantification of the number of lung metastases per mouse (*n* = 8 mice each). **c**, Quantification of total lung tumour burden per mouse (*n* = 8 mice each). **d**, Summary data indicating the presence or absence of lung metastases. The number of mice with or without lung metastases is indicated on the graph (*n* = 8 mice each). **e**, Representative images of liver metastases (stained with haematoxylin and eosin) from mice injected with AKP control or AKP *Atrx^KO^* organoids through intrasplenic injection. Metastatic nodules are indicated with black arrowheads. **f**, Quantification of the number of liver metastases per mouse (*n* = 8 mice each). **g**, Quantification of the total liver tumour burden per mouse (*n* = 8 mice each). **h**, Summary data indicating the presence or absence of liver metastases. The number of mice with or without liver metastases is indicated on the graph

(*n* = 8 mice each). **i**, Fluorescence microscopy of calcein-stained AKP control and AKP *Atrx^KO^* organoids after treatment with 5 ng ml⁻¹ TGFβ (TGFβ1). Spindle-like organoid structures are indicated with white arrowheads. Zoomed areas are outlined by the white boxes. **j**, Quantification of the percentage of AKP control and AKP *Atrx^KO^* organoids adopting a spindle-like morphology after TGFβ treatment (*n* = 7 (control) and 6 (KO) independent experiments). *P* = 0.000027. **k**, RT–qPCR analysis of EMT markers in AKP control and AKP *Atrx^KO^* organoids untreated or treated with 5 ng ml⁻¹ TGFβ (*n* = 3 independent experiments). Gene expression was normalized to *Actb*, and levels relative to untreated AKP control were calculated using the ΔΔ$C_t$ method. Data are the mean ± s.d. (**b,c,f,g,j,k**). *P* values were calculated using two-tailed Mann–Whitney tests (**b,c,f,g**), two-sided Fisher's exact tests (**d,h**), two-tailed Student's *t*-tests (**j**) or ordinary one-way analysis of variance ANOVA with multiple comparisons (**k**). Scale bars, 2.5 mm (**a,e**) or 1,000 μm (**i**).

Given the observed loss of colonic epithelial identity and gain of squamous gene expression observed in the bulk RNA-seq data, we next sought to determine whether *Atrx* deletion induces an adoption of non-intestinal lineage programs. Cluster analysis confirmed the emergence of multiple squamous-like and other non-canonical cell states after *Atrx* loss. Cluster 15 expressed markers of squamous-like epithelia,

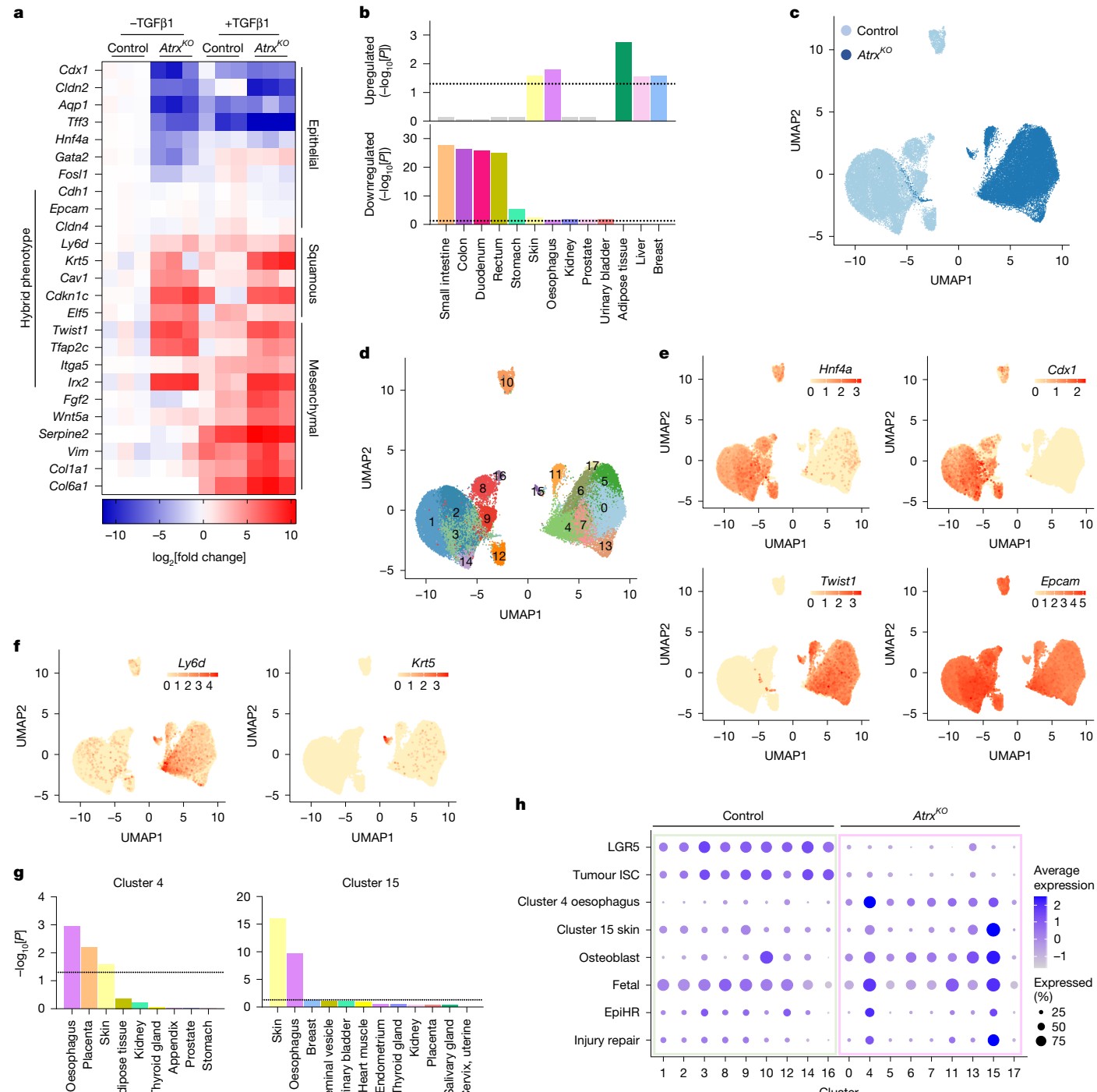

**Fig. 2 | Colonic epithelial identity is perturbed after *Atrx* loss. a**, Heatmap of RNA-seq data from AKP control and AKP *Atrx^KO* organoids with or without TGFβ treatment. Representative genes marking colonic epithelial, squamous and mesenchymal lineages are shown. log₂ fold change values relative to untreated AKP control organoids are indicated by the colour intensity. Genes of multiple lineages co-expressed in AKP *Atrx^KO* organoids are highlighted as 'hybrid phenotype'. **b**, TissueEnrich analysis of genes upregulated and downregulated in AKP *Atrx^KO* organoids compared with AKP controls. Dashed line indicates *P* = 0.05. **c**, UMAP plot of AKP control (23,579 cells) and AKP *Atrx^KO* (25,757 cells) single cells coloured by genotype. **d**, UMAP plot coloured and numbered by cluster in AKP control and AKP *Atrx^KO* single cells. **e**, UMAP plots coloured by the expression of genes used for defining colonic differentiation and EMT in AKP control and AKP *Atrx^KO* single cells. Colour scale indicates expression levels. **f**, UMAP plot coloured by the expression of genes used for defining squamous differentiation in AKP control and AKP *Atrx^KO* single cells. Colour scale indicates expression levels. **g**, TissueEnrich analyses of genes enriched in single-cell RNA-seq clusters 4 and 15. Dashed line indicates *P* = 0.05. **h**, Dot plot of signature scores across all clusters coloured by the average expression and sized by the percentage of cells expressing the signature. Cluster 4 oesophagus and cluster 15 skin signatures are derived from TissueEnrich analyses. Significance was calculated using hypergeometric tests (one-sided) with Benjamini–Hochberg multiple-testing correction (**b**,**g**).

including *Ly6d*, *Trp63*, *Krt5*, *Krt14*, *Krt79* and *Krtdap* (Fig. 2f–h and Supplementary Table 4). Cluster 4 also expressed squamous-like markers, such as *Ly6d*, *Sprr1a* and *Mxd1* (Fig. 2f–h and Supplementary Table 4).

TissueEnrich analysis of markers of these clusters identified cluster 15 as being strongly associated with skin and oesophageal epithelia, whereas cluster 4 showed a weaker association with these cell states, which was

probably due to the lack of keratin expression (Fig. 2g). In *Atrx*[KO] cells, we also observed broad upregulation of osteoblast-like markers such as *Id1*, *Id3*, *Bmp2* and *Wif1*, which indicated a shift towards a mesenchymal-like phenotype (Fig. 2h and Extended Data Fig. 5b). Expression of osteoblast markers was particularly enriched in clusters 13 and 15, which indicated the presence of mixed-lineage populations[17] (Fig. 2h and Extended Data Fig. 5b). Overall, the presence of multiple non-canonical, squamous and mesenchymal-like lineages, which have been recently linked to metastasis[17], highlights the induction of broad cellular reprogramming in *Atrx*[KO] cells, a finding indicative of increased phenotypic plasticity.

We next determined the relationship of these non-canonical lineages to previously identified CRC cell states linked to tumour progression. Canonical, LGR5 stem cell and tumour intestinal stem cell signatures were depleted in *Atrx*[KO] cells, consistent with the loss of colonic identity[17] (Fig. 2h and Extended Data Fig. 5c). Conversely, we found upregulation of fetal intestine, epithelial-specific high-risk and injury-repair programs, which have previously been associated with tumour progression and metastasis[17–19] (Fig. 2h and Extended Data Fig. 5d). These programs were enriched in the squamous-like clusters 4 and 15, which indicated the co-expression of fetal and injury repair and non-canonical cell-lineage states (Fig. 2h). Together, these results suggest that rather than shifting towards a single alternative lineage, *Atrx*[KO] cells adopt a range of non-canonical cell states with distinct clusters exhibiting mixed features of squamous, mesenchymal and fetal intestinal identities. The emergence of these hybrid cell states, which have been linked to tumour progression and metastasis[17], suggests that *Atrx* plays a key role in maintaining colonic epithelial identity and restricting lineage plasticity in CRC.

## *Atrx* suppresses multilineage plasticity

To dissect how *Atrx* loss leads to lineage plasticity, we first investigated the induction of mesenchymal gene expression. Co-immunofluorescence staining for CDH1 and TWIST1 revealed that *Atrx*[KO] organoids express both these epithelial and mesenchymal markers (Extended Data Fig. 6a). Further characterization using fluorescent-activated cell sorting (FACS) for markers of epithelial (EPCAM) and mesenchymal (ITGA5) cells confirmed the expression of both these lineage markers (Extended Data Fig. 6b–d). Notably, EPCAM expression was maintained in *Atrx*[KO] cells, which indicated the transition towards a hybrid EMT cell state rather than full EMT induction (Extended Data Fig. 6e). Subsequent cell sorting on the basis of ITGA5 expression showed that ITGA5[+] cells displayed increased sensitivity to TGFβ-induced EMT and elevated expression of EMT markers, which confirmed the establishment of a hybrid EMT phenotype (Extended Data Fig. 6f–j).

We next investigated the characteristics of the squamous-like cell states induced by *Atrx* loss. FACS analysis showed that expression of the squamous cell marker LY6D was significantly induced in AKP *Atrx*[KO] organoids (Fig. 3a–c). Transcriptional analysis of sorted LY6D[+] cells confirmed the expression of markers of squamous epithelium, such as *Krt5* and *Krt13* (Fig. 3d). These cells had increased expression of *Emp1*, a marker of metastatic CRC[18], and, consistent with a shift away from a colonic epithelial expression profile, they had reduced expression of the canonical colonic stem cell marker *Lgr5* (Extended Data Fig. 7a). To determine whether squamous-like cells were present in vivo, we stained for KRT5 in both subcutaneous transplants and lung metastases from our mouse models. AKP *Atrx*[KO] tumours contained significantly more KRT5[+] cells than control tumours, and metastatic lesions were further enriched compared with subcutaneous tumours (Fig. 3e–g and Extended Data Fig. 7b). Further analysis of these cells revealed an abnormal phenotype compared with CRC cells not expressing KRT5. Individual KRT5[+] cells displayed an elongated morphology, distinct from the typical columnar morphology of colonic epithelium (Extended Data Fig. 7c). Moreover, clusters of KRT5[+]

cells acquired a stratified-like phenotype characteristic of squamous epithelia (Extended Data Fig. 7c). We also observed the occurrence of structures resembling keratin pearls, which are more commonly associated with cancers of squamous cell origin (Extended Data Fig. 7c). Together, these findings indicate that *Atrx* loss leads to the emergence of cells with squamous-like characteristics. Co-immunofluorescence staining for EPCAM and KRT5 in subcutaneous tumours demonstrated an apparent continuum of cell states. We found areas of exclusive KRT5 or EPCAM positivity, but also cells expressing both markers, a result indicative of phenotypic transitions between columnar epithelial and squamous-like states (Fig. 3h). To better understand the relationship between the observed squamous and mesenchymal hybrid cells, LY6D and ITGA5 co-staining was used. This revealed the presence of populations that expressed the individual marker and a highly mixed-lineage cell population that expressed both the squamous and mesenchymal markers (Fig. 3i,j and Extended Data Fig. 7d).

Our observations that *Atrx* loss induces cellular plasticity and mixed-lineage cell states prompted us to explore the stability of the hybrid-lineage populations that emerge. To this end, we sorted the various hybrid epithelial, mesenchymal and squamous cell populations, replated them and reanalysed them 9 days later (Fig. 3k). Regardless of the initial population plated, a resurgence of the other cell populations was observed (Fig. 3l and Extended Data Fig. 7e). This result was validated in vivo, as transplant experiments revealed a similar, dynamic re-emergence of KRT5[+] squamous-like cell populations regardless of the cell population transplanted (Fig. 3m and Extended Data Fig. 7f). Thus, *Atrx* loss confers a high level of plasticity and gives CRC cells the ability to readily shift between columnar epithelial, mesenchymal and squamous-like cell states.

Owing to its previously described function in regulating EMT in cancer, we next tested the role of TGFβ signalling in mediating these lineage transitions. First, we tested the short-term effects of TGFβ treatment on organoid growth. AKP *Atrx*[KO] organoids were partially resistant to the growth-suppressive effects of this treatment (Extended Data Fig. 8a,b). To test the functional requirement for TGFβ signalling in mediating non-canonical lineage plasticity, we deleted *Tgfbr2* in our AKP *Atrx*[KO] model to disrupt TGFβ signalling (Extended Data Fig. 8c). This modification resulted in a reduction in the expression of both squamous (*Krt5* and *Ly6d*) and EMT-associated (*Twist1*) markers, along with a decrease in the proportion of LY6D[+], ITGA5[+] and LY6D[+]ITGA5[+] double-positive cells (Extended Data Fig. 8d–f). Notably, the expression of colon-specifying genes such as *Hnf4a*, *Cdx1* and *Cdx2* was unchanged, which suggested that loss of colonic identity is mostly independent of TGFβ signalling (Extended Data Fig. 8d). Moreover, AKP *Atrx*[KO] cells lacking *Tgfbr2* were unable to undergo EMT in response to TGFβ stimulation (Extended Data Fig. 8g,h). Although these results highlight an important role for TGFβ in promoting plasticity, the persistence of some LY6D[+] populations suggests that alternative mechanisms may contribute to these transitions. Thus, *Atrx* loss leads to an impairment of colonic lineage fidelity and an induction of mesenchymal and squamous-like plasticity in a manner that partially depends on TGFβ signalling.

## *Hnf4a* loss perturbs colonic identity

To unravel the molecular mechanisms underpinning the role of ATRX in maintaining lineage fidelity, we investigated its chromatin-remodelling function by carrying out assay for transposase-accessible chromatin with sequencing (ATAC–seq) on AKP control and AKP *Atrx*[KO] organoids. Principal component analysis revealed prominent separation of organoid genotypes in the first component (Extended Data Fig. 9a), which indicated the presence of differences in ATAC–seq enrichment patterns between the sample groups (Supplementary Table 6). The differences included notable, significant losses at sites associated with colonic epithelial gene expression (Extended Data Fig. 9b). We also observed gain of accessibility at some mesenchymal and squamous cell markers,

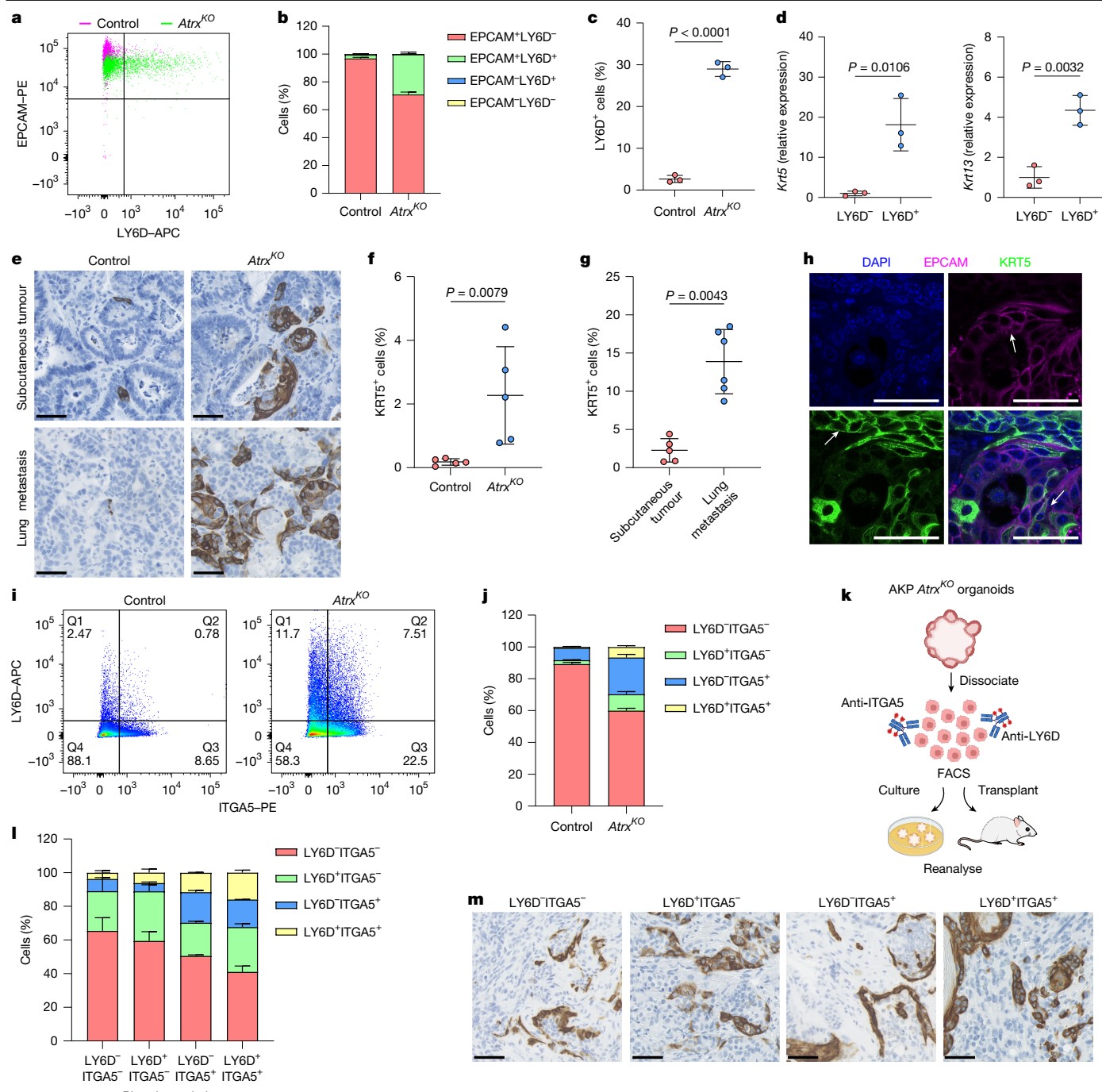

**Fig. 3 | Atrx loss induces squamous-like plasticity. a**, FACS plots for analysing and sorting EPCAM⁺LY6D⁺ cells from AKP control and AKP Atrx^KO organoids. **b**, Quantification of the percentage of cells in each EPCAM and LY6D population (n = 3 independent experiments each). **c**, Quantification of the percentage of LY6D⁺ cells in AKP control and AKP Atrx^KO organoids (n = 3 independent experiments each). P = 0.000019. **d**, RT–qPCR analysis of squamous cell markers in LY6D⁻ and LY6D⁺ cells sorted from AKP Atrx^KO organoids (n = 3 independent experiments each). Gene expression was normalized to Actb, and levels relative to LY6D⁻ cells were calculated using the ΔΔC_t method. **e**, Representative images of KRT5-stained subcutaneous tumours and lung metastases from mice injected with AKP control or AKP Atrx^KO organoids. **f**, Quantification of the percentage of KRT5⁺ cells in subcutaneous tumours from mice injected with AKP control or AKP Atrx^KO organoids (n = 5 mice each). **g**, Quantification of the percentage of KRT5⁺ cells in subcutaneous tumours and lung metastases from mice injected with AKP Atrx^KO organoid cells (n = 5 (subcutaneous) and 6 (lung metastasis) mice). **h**, Representative images of EPCAM and KRT5 co-immunofluorescence in AKP Atrx^KO subcutaneous

tumours. White arrows indicate cells exclusively expressing EPCAM (in EPCAM only panel, magenta), KRT5 (in KRT5 only panel, green) or co-expressing EPCAM and KRT5 (merge). n = 3 biologically independent samples. **i**, FACS plots for analysing and sorting LY6D⁺ITGA5⁺ cells from AKP control and AKP Atrx^KO organoids. **j**, Quantification of the percentage of cells in each LY6D and ITGA5 population (n = 3 independent experiments each). **k**, Schematic of the strategy used to determine the plasticity of different ITGA5 and LY6D expressing cell populations in AKP Atrx^KO organoids. **l**, Quantification of the percentage of cells in each LY6D and ITGA5 population 9 days after plating. The original plated population is noted on the x axis (n = 4 independent experiments each). **m**, Representative images of KRT5-stained subcutaneous tumours from mice injected with different LY6D and ITGA5 populations derived from AKP Atrx^KO organoid cells. The population transplanted is indicated above each image (n = 5 mice each). Scale bars, 50 μm. Data are the mean ± s.d. (**c,d,f,g**). P values were calculated using two-tailed Student's t-test (**c,d**) or two-tailed Mann–Whitney test (**f,g**). Scale bars, 50 μm (**e,h,m**). The schematic in **k** was created using BioRender (https://biorender.com).

consistent with induction of these intermediate cell states (Extended Data Fig. 9c,d). To investigate how these chromatin changes induce the phenotypic alterations we observed, we carried out TF-binding site analysis on sites of altered accessibility. This analysis revealed a marked enrichment in binding sites of the HNF4A, HNF4G, GATA2, GATA3 and GATA5 TFs in regions with decreased accessibility (Extended Data Fig. 9e). To further quantify this differential TF activity, we performed diffTF analysis on our combined ATAC–seq and RNA-seq datasets[20]. This analysis confirmed the loss of HNF4A and HNF4G and GATA binding site accessibility and reduced transcriptional output associated with these TFs (Fig. 4a). Complementary mapping of enhancer activity using H3K27ac CUT&RUN revealed a similar pattern, with widespread loss of enhancer activity after *Atrx* loss (Fig. 4b, Extended Data Fig. 9f and Supplementary Table 7). By comparing these datasets, we revealed that the majority of sites that lose H3K27ac overlapped with regions displaying reduced chromatin accessibility (Fig. 4c). Again, this was apparent at sites associated with colonic epithelial gene expression (Fig. 4d). The integration of these datasets suggests that *Atrx* loss disrupts the chromatin landscape, specifically at sites associated with epithelial gene expression, by directly affecting chromatin accessibility and enhancer activity.

The HNF4 TFs have a crucial role in maintaining normal colonic identity[21,22], which suggests that loss of this activity may result in the impaired lineage fidelity observed after *Atrx* loss. To functionally test this possibility, we carried out HNF4A gain and loss of function experiments. Consistent with the proposal that HNF4A activity maintains colonic epithelial identity, HNF4A overexpression in AKP *Atrx^{KO}* organoids rescued the expression of some of the colonic lineage markers analysed (Extended Data Fig. 9g). This partial rescue suggests that HNF4A activity maintains colonic epithelial identity, probably in cooperation with additional factors. To further investigate this possibility, we generated AKP *Hnf4a^{KO}* organoids (Extended Data Fig. 9h,i) and performed RNA-seq, which revealed widespread changes in gene expression (Extended Data Fig. 9j and Supplementary Table 8). Notably, we observed loss of colon-specific genes such as *Cdx1*, *Muc4*, *Lyz1* and *Tff3*, alongside upregulation of genes associated with non-canonical lineages, including *Twist1*, *Krt4*, *Krt17*, *Krt79*, *Wif1* and *Id3* (Extended Data Fig. 9j and Supplementary Table 8). Comparative analysis of gene expression changes in *Atrx^{KO}* and *Hnf4a^{KO}* organoids demonstrated a strong overlap in downregulated genes, which indicated that ATRX and HNF4A have a key role in maintaining colonic identity (Fig. 4e).

Functionally, *Hnf4a^{KO}* organoids exhibited increased sensitivity to TGFβ-induced EMT (Fig. 4f,g). Moreover, subcutaneous transplantation of these organoids gave rise to tumours that had lost their glandular architecture (Fig. 4h and Extended Data Fig. 9k–l). Together, these findings indicate that loss of HNF4A activity at colonic specifying genes results in loss of colonic epithelial identity, which in turn may create a permissive environment for the emergence of the highly plastic cell states that emerge after *Atrx* loss.

## Squamous phenotype in metastatic disease

To assess the clinical relevance of our findings, we investigated the consequences of loss of colonic identity and squamous-like transitions in human CRC samples. We first asked whether the relationship among ATRX, HNF4A, CDX2 and LY6D expression identified in our mouse model exists in human tumour samples. Staining of a CRC tissue microarray (TMA) consisting of stage I–III primary tumours using antibodies for these four proteins revealed a significant correlation between the expression of ATRX, HNF4A and CDX2, a result in keeping with our mouse model findings (Fig. 5a and Extended Data Fig. 10a–c). Moreover, consistent with a role for ATRX, HNF4A and CDX2 in maintaining lineage fidelity and suppressing squamous-like plasticity, we observed low expression of these proteins in tumours containing LY6D+ cells (Fig. 5b).

We next examined how these lineage transitions evolve in advanced disease by staining matched primary and liver metastasis samples for ATRX, HNF4A, CDX2, LY6D and KRT5. Compared with our stage I–III tumour TMA dataset, squamous-like cells were more prevalent in stage IV tumours, which suggested that squamous-like plasticity becomes more prominent in later stage disease (Fig. 5c,d and Extended Data Fig. 10d,e). However, there was no difference in squamous cell abundance between primary tumours and their matched liver metastases, which indicated that this phenotype is already established before metastatic dissemination (Fig. 5e,f and Extended Data Fig. 10f,g). We also found a modest reduction in HNF4A and ATRX expression in liver metastases compared to their matched primary tumours, which provided further support for a progressive loss of colonic identity in advanced disease (Extended Data Fig. 10h–k).

To functionally validate these observations, we knocked out *ATRX* in human patient-derived CRC organoids and assessed their phenotype (Extended Data Fig. 10l). Consistent with our findings in mouse models and patient samples, deletion of *ATRX* led to a reduction in the expression of the colonic epithelial marker genes *HNF4A* and *CDX1* and an increase in the squamous-like marker *KRT5* (Extended Data Fig. 10m). Moreover, *ATRX^{KO}* organoids showed an enhanced propensity for TGFβ-induced EMT, thereby providing direct functional evidence of a role for *ATRX* in maintaining colonic identity and restricting lineage plasticity in human CRC (Extended Data Fig. 10n,o).

We next asked whether squamous-like gene expression was associated with disease phenotype and clinical outcome. The recently defined intrinsic consensus molecular subtypes (iCMS) separate CRC cases on the basis of epithelial gene expression patterns[23]. Two iCMS exist: iCMS2, which is associated with relatively good prognosis, and iCMS3, which shows evidence of metaplasia and is associated with poor prognosis[23]. We first analysed the prevalence of *ATRX* mutations in these subtypes and found an enrichment of *ATRX* alterations in iCMS3 tumours (Extended Data Fig. 11a). We next generated transcriptional signatures indicative of the squamous-like phenotype observed after *Atrx* loss in our CRC model (*Atrx^{KO}* signature) and the colonic epithelial-like phenotype that is lost (*Atrx^{WT}* signature). The *Atrx^{KO}* signature contained markers of squamous cells (*LY6D*, *KRT5*, *KRT13*, *SPRR1A* and *TRIM29*) and those associated with the basal–squamous pancreatic ductal adenocarcinoma subtype and/or metaplastic CRC (*CAV1*, *AQP3*, *FAM83A*, *F3* and *AQP5*)[24,25]. By contrast, the *Atrx^{WT}* signature contained markers of colonic epithelium (*HNF4A*, *ASCL2*, *CDX1* and *CDX2*). Analyses of previously published patient data[26] using *Atrx^{KO}* and *Atrx^{WT}* signatures separated patients into three distinct clusters: those with high colonic epithelial and low squamous-like expression (HiCol cluster); those with reduced colonic epithelial and increased squamous-like expression (intermediate cluster); and those with high squamous-like and low colonic epithelial expression (HiSquam cluster) (Extended Data Fig. 11b). Analyses of these patients determined that intermediate and HiSquam cluster expression was significantly enriched in patients with right-sided (proximal) disease, mismatch-repair deficiency and *BRAF* mutation (Extended Data Fig. 11c–e). Patients with an advanced tumour stage were also enriched in HiSquam cluster expression, but this did not reach significance (Extended Data Fig. 11f). Consistent with these findings, intermediate and HiSquam cluster expression was associated with iCMS3 tumours (Extended Data Fig. 11g,h). Moreover, when mapped onto single tumours cells[23], the *ATRX^{KO}* score was enriched in iCMS3 cells and the *ATRX^{WT}* score mapped to iCMS2 tumour cells (Fig. 5g). These analyses suggested an enrichment of lineage plasticity in aggressive, right-sided disease. To further investigate this finding, we analysed primary patient biopsy samples for the presence of cells with squamous-like phenotypes. FACS analysis confirmed the presence of tumour cells expressing both LY6D and EPCAM, thereby confirming the presence of this hybrid squamous-like state in primary human CRC samples (Extended Data Fig. 11i and Supplementary Table 9). Consistent with the transcriptional datasets, analysis of multiple

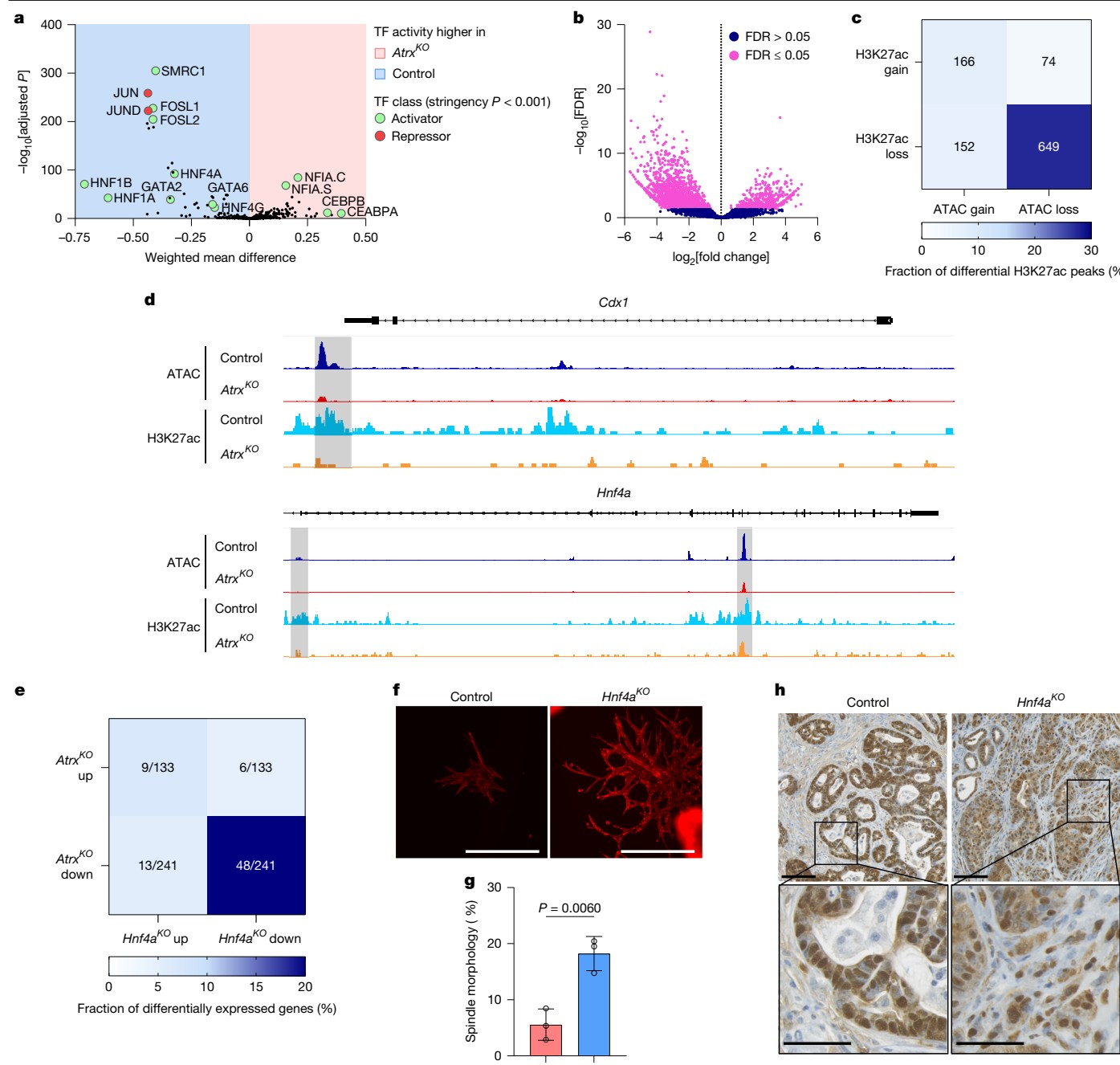

**Fig. 4 | HNF4A activity maintains colonic epithelial identity. a**, DiffTF analysis of combined ATAC–seq and RNA-seq data with selected TFs highlighted. Shaded areas indicate gain (red) or loss (blue) of TF activity in AKP $Atrx^{KO}$ organoids compared with AKP controls. TFs highlighted in green are associated with transcriptional activation and in red with transcriptional repression. **b**, Volcano plot of H3K27ac CUT&RUN data in AKP control and AKP $Atrx^{KO}$ organoids. Each data point represents a H3K27ac binding peak. Significantly altered sites (false discovery rate (FDR) ≤ 0.05) are highlighted in pink. **c**, Table outlining the overlap between ATAC–seq accessibility changes and altered H3K27ac peaks in AKP control and AKP $Atrx^{KO}$ organoids. H3K27ac losses are mostly associated with reduced chromatin accessibility. **d**, Representative Integrative Genomics Viewer (IGV) browser tracks of AKP control and AKP $Atrx^{KO}$ ATAC–seq accessibility and H3K27ac CUT&RUN data. *Cdx1* and *Hnf4a* gene loci are shown. Regions shaded grey have significant loss of chromatin accessibility and corresponding

depletion of H3K27ac. **e**, Table outlining the overlap between RNA-seq gene expression changes in AKP $Atrx^{KO}$ and AKP $Hnf4a^{KO}$ organoids. **f**, Fluorescence microscopy of phalloidin-stained AKP control or AKP $Hnf4a^{KO}$ organoids after treatment with TGFβ (5 ng ml⁻¹). Scale bars, 400 μm. **g**, Quantification of the percentage of AKP control and AKP $Hnf4a^{KO}$ organoids adopting a spindle-like morphology after TGFβ treatment ($n = 3$ independent experiments each). **h**, Representative IHC images of β-catenin-stained subcutaneous tumours from mice injected with AKP control or AKP $Hnf4a^{KO}$ organoid cells ($n = 5$ mice each). β-catenin staining is used to identify tumour cells. Scale bars, 100 μm (overview) and 50 μm (zoom). For **a**, two-sided $P$ values for each transcription factor was calculated with Welch two-sample $t$-tests using the bootstrap approach. Adjusted $P$ values were calculated using the Benjamini–Hochberg method for multiple-testing correction. For **g**, data are mean ± s.d., and $P$ values were calculated using two-tailed Student's $t$-tests.

samples revealed that the proportion of LY6D⁺ cells was significantly increased in right-sided disease (Extended Data Fig. 11j). To determine the prognostic relevance of lineage plasticity, we stratified patients

on the basis of expression of our $Atrx^{KO}$ and $Atrx^{WT}$ expression signatures. Patients with high squamous-like and low colonic epithelial gene expression (HiSquam cluster) had significantly poorer overall survival

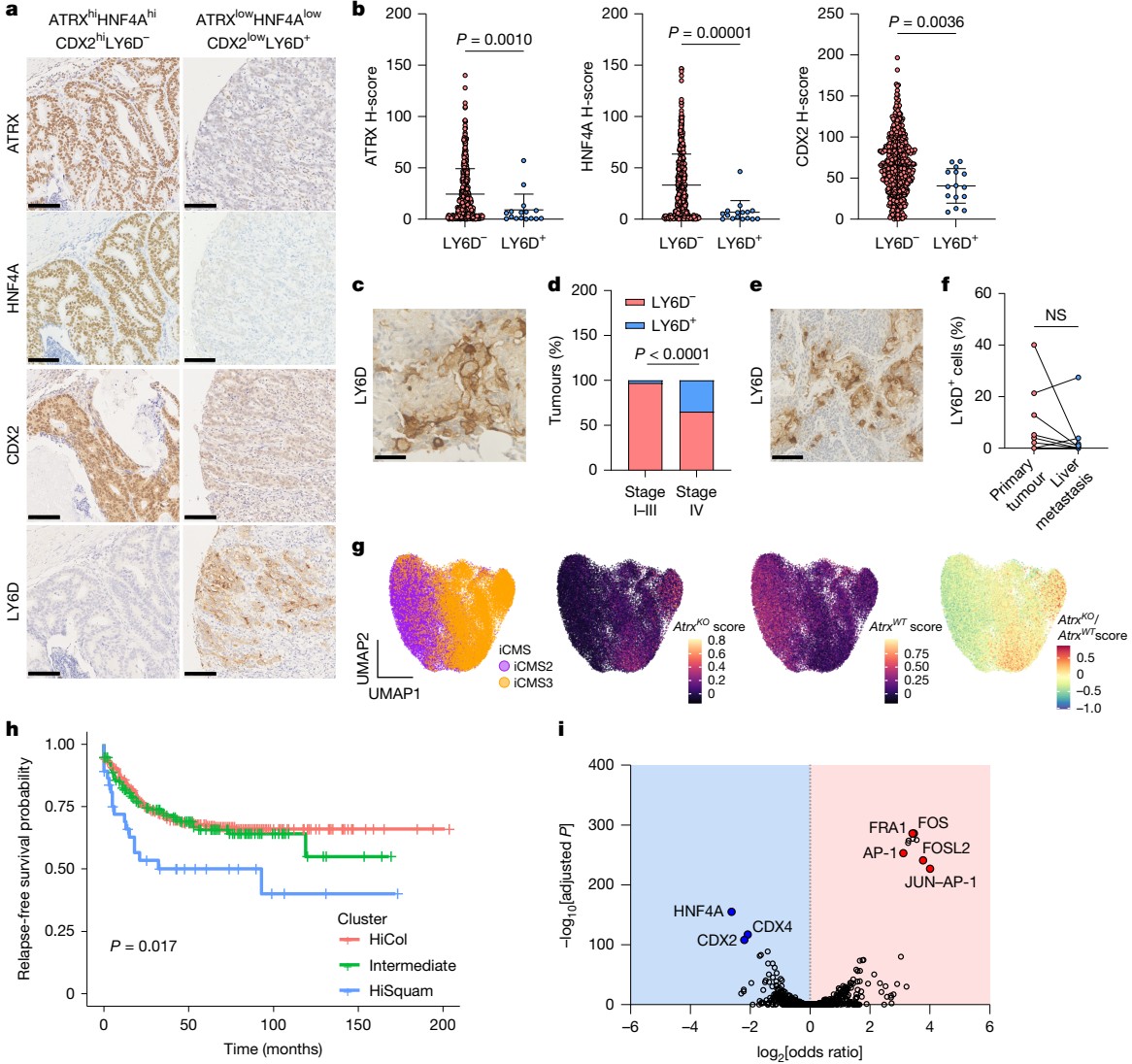

**Fig. 5 | Squamous-like gene expression predicts aggressive disease and poor patient outcome. a**, Representative IHC staining of a human CRC TMA for ATRX, HNF4A, CDX2 and LY6D. Examples of positive and negative staining are shown. Scale bar, 100 μm. **b**, Quantification of ATRX, HNF4A and CDX2 histoscore (H-score) values in LY6D⁻ (<2% cells LY6D⁺) and LY6D⁺ (>2% cells LY6D⁺) tumour cores. *n* = 500 (ATRX, HNF4A) and 509 (CDX2) biologically independent samples. **c**, Representative IHC image of a LY6D-stained human stage IV primary tumour. Scale bars, 50 μm. **d**, Summary data indicating the percentage of human primary tumours at stages I–III versus stage IV positive for LY6D. The percentages are based on >2% cells LY6D⁺ and <2% cells LY6D⁻. *P* = 0.000027. **e**, Representative IHC image of LY6D-stained human liver metastasis. Scale bars, 100 μm. **f**, Summary data indicating the percentage of LY6D⁺ cells in matched human primary tumours and liver metastases. *n* = 17 biologically independent matched samples (7 data points are visible as 11 primary tumour samples have the same value (0) and are overlapping). **g**, UMAP plot of Juanito scRNA-seq dataset overlayed with iCMS designation. For comparison, *Atrx^{KO}*, *Atrx^{WT}* and *Atrx^{KO}/Atrx^{WT}* transcriptional expression scores are overlayed in the same data. **h**, Survival plot of Marisa CRC patient dataset separated on *Atrx*-based gene expression clusters. **i**, Volcano plot of HOMER TF enrichment analysis of TFs with differential motif accessibilities between HiSquam and HiCol signature tumours. Selected TF motifs in regions with reduced accessibility in HiSquam tumours highlighted in blue and TF motifs in regions with increased accessibility in HiSquam tumours highlighted in red. Data are the mean ± s.d. (**b**). *P* values were calculated using two-tailed Mann–Whitney tests (**b**), two-sided Fisher's exact tests (**d**), two-tailed Wilcoxon matched-pairs signed-rank tests (**f**), log-rank (Mantel–Cox) tests (**h**) or two-sided Fisher's exact test and adjusted for multiple comparisons with the Benjamini–Yekutieli method (**i**). NS, not significant.

than patients in the other two transcriptional clusters (Fig. 5h). Thus, the transcriptional phenotype controlled by *Atrx* can define tumour molecular phenotypes that are associated with aggressive disease and poor patient prognosis.

We then asked whether these transcriptional signatures are associated with alterations in chromatin structure by analysing previously published ATAC–seq data from primary human CRC[27]. To this end, we used our *Atrx^{KO}* and *Atrx^{WT}* signatures to stratify patients from The Cancer Genome Atlas (TCGA) with available RNA and ATAC–seq data into high, medium and low on the basis of the single-sample GSEA scores of the signatures. We then compared chromatin accessibility differences between the groups, HiSquam (high for *Atrx^{KO}* and low for *Atrx^{WT}*) and HiCol (low for *Atrx^{KO}* and high for *Atrx^{WT}*), and identified a set of genomic regions with altered accessibility across tumour samples (Extended Data Fig. 12a). TF motif enrichment analysis revealed loss of accessibility at sites that contained motifs for epithelial-specifying genes, including HNF4A and CDX2 in patients with HiSquam tumours (Fig. 5i and Supplementary Table 10). In line with our *Atrx* loss-of-function model, these results demonstrate that loss of activity of colonic lineage-specifying transcriptional regulators is associated with increased lineage plasticity and the expression of squamous-like phenotypic markers in human samples.

## Discussion

Cellular plasticity is a key mechanism that drives cancer metastasis. As metastatic disease often arises in the absence of additional genetic perturbations, epigenetic mechanisms have been proposed as pivotal mediators. Here we identified the chromatin-remodelling enzyme ATRX as a crucial regulator of CRC plasticity and metastasis. Loss of *Atrx* gives rise to a highly plastic cell state, whereby cancer cells acquire both mesenchymal and squamous-like characteristics. Although EMT is a well-recognized aspect of tumour progression, the role of squamous trans-differentiation is poorly understood. Squamous cell transition states have been identified in cancers such as pancreatic, bladder and lung[28–30], but have only recently been described in CRC. A recent study[17] identified non-intestinal cell states, including squamous cells marked by KRT5 expression, in individuals with poor prognosis, metastatic CRC. Thus, the acquisition of non-canonical lineages seems to be an emerging phenomenon in this disease. In this context, our study provides mechanistic insight into how these cell states arise, and places the loss of colonic lineage fidelity and the activity of lineage specifying TFs at the heart of this regulation.

Mechanistically, we showed that loss of chromatin accessibility and enhancer activity at colonic lineage specifying genes mediates loss of colonic epithelial identity. Across all our analyses, the loss of colonic identity was highly notable, which suggests that this is the initiating event controlled by ATRX that, when lost, enables the induction of plasticity. Indeed, the gain in non-canonical lineages, both mesenchymal and squamous, was less evident at the chromatin level. This finding indicates that rather than being driven towards an alternative lineage, once colonic identity is 'softened', the promotion of non-canonical lineages may occur in a stochastic manner, perhaps influenced by environmental factors. This hypothesis is supported by previous studies of pancreatic cancer that have highlighted the importance of the tumour microenvironment in mediating cellular transitions[25]. Our functional studies defined HNF4A as a key protector of the colonic lineage, thereby further emphasizing the importance of lineage-defining TFs in protecting against the acquisition of non-canonical cell states. We provided evidence of a similar mechanism in patient samples, whereby a squamous-like gene expression signature based on genes that are overexpressed after *Atrx* loss predicts aggressive disease and poor prognosis. Notably, tumours that exhibited this squamous-like gene expression pattern had reduced expression of colonic epithelial genes and showed changes in chromatin accessibility indicative of loss of HNF4A activity. Similar findings in other cancers, such as pancreatic, indicate that loss of activity of lineage-specifying TFs may be a ubiquitous mechanism that enables cellular plasticity in aggressive cancers[31]. Together, our study shed light on the mechanisms that facilitate metastasis and promote cellular plasticity. Further understanding of how these processes contribute to cancer progression will pave the way for potential opportunities for therapeutic intervention.

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

# Methods

## Mouse studies

All in vivo experiments were performed in accordance with the UK Home Office regulations (project licence: PP9016178, PP7510272 and PP3908577) and were subject to local ethics review by the animal welfare and ethics board of the University of Edinburgh and University of Glasgow. Mice were housed under a 12-h light–dark cycle at a temperature ranging from 19 to 23 °C. Ambient humidity was 55 ± 10%. Standard diet and water were available ad libitum. Work was performed on the C57BL/6J and the CD1 nude background available from Charles River Laboratories. For the lung metastatic model, 500,000 AKP control or AKP $Atrx^{KO}$ cells were resuspended in 100 µl PBS and injected into the tail vein of female CD1 nude mice. For the liver metastatic model, C57BL/6J mice were anaesthetized with isoflurane and splenic access was achieved using laparotomy. Around 500,000 cells were injected in 50 µl PBS, following which, the incision was closed with staples. Subcutaneous injections into female CD1 nude mice were performed with 50,000 AKP control, AKP $Atrx^{KO}$ or AKP $Hnf4a^{KO}$ cells resuspended in 100 µl basement membrane extract (BME). Subcutaneous injections into female CD1 nude mice were performed with 12,000 flow cytometry-sorted cells resuspended in 100 µl BME.

Colonic submucosal injections were performed using a Karl Storz TELE PACK VET X LED endoscopic video unit. AKP control, AKP $Atrx^{KO}$, BPN control and BPN $Atrx^{KO}$ organoids were dissociated by mechanical pipetting and then were washed with PBS before being injected orthotopically in female CD1 nude mice. Approximately 500 organoids in 70 µl PBS were injected in a single injection. The BPN organoid line (male) was generated in the laboratory of O.J.S.

In accordance with the respective licence and study protocol, mice were humanely euthanized at the reported experimental end points (15 mm diameter for subcutaneous injection) as defined in the relevant licensing documents.

In accordance with the 3Rs (replacement, reduction and refinement), the smallest sample size was chosen that could give a significant difference.

C57BL/6J or CD1 nude mice 6–12 weeks of age were randomly grouped for transplantation experiments. All mice received the same number of cells of a different genotype or phenotype. Experimental groups were determined by the genotype of injected cells (for example, AKP control versus AKP $Atrx^{KO}$). Investigators were blinded to the genotype of tumours when monitoring for clinical signs, when carrying out histological analyses and during data collection. IHC analysis of tumour histology was carried out using QuPath software, with the investigator blinded to the tumour genotype.

## Histopathology, IHC and immunofluorescence

Tissues were fixed overnight in formaldehyde 4% stabilized, buffered (VWR) and then transferred to 70% ethanol. Tissue processing was done using a Tissue-TeK VIP infiltration processor (Sakura) followed by embedding in paraffin wax. Tissues were sectioned at 5 µm thickness, dried at 37 °C and then deparaffinized and rehydrated. For each sample, one section was stained with haematoxylin and eosin. Additional sections were used for either standard IHC or immunofluorescence analyses. The following primary antibodies were used: KRT5 (rabbit; Abcam 52635; 1:200 or chicken; BioLegend 905903; 1/200); LY6D (rabbit; Atlas HPA024755; 1:200); TWIST (mouse; Santa Cruz 81417, 1:200); EPCAM (rabbit; Abcam 71916; 1:200); HNF4A (rabbit; CST 3113; 1:500); ATRX (mouse; Sigma MABE1798; 1:500); β-catenin (mouse; BD 610154; 1:50); E-cadherin (rabbit; CST 3195, 1:200); and CDX2 (mouse; Atlas AMAb 91828, 1:1,000). IHC secondary detection was achieved using EnVision+ System-HRP labelled polymer (Dako). Positive signals were visualized using DAB substrate (Epredia and 2b scientific) for 5–10 min. Sections were counterstained with haematoxylin. For immunofluorescence studies, the following secondary antibodies (1:400)

were used: anti-rabbit-594 (Invitrogen, A21207); anti-chicken-488 (Invitrogen, A78948); anti-streptavidin-647 (Invitrogen, S32357); and anti-rabbit-488 (Abcam, 150073) Representative images are shown for each stain. The slides were digitized using a Nanozoomer digital slide scanner (Hamamatsu) with NDP.view2plus software and analysed using QuPATH (v.0.2.3). The commercial colon cancer human tissue array used was CO804b (Biomax) (Extended Data Fig. 1b–d). The TMA in Fig. 5 and Extended Data Fig. 10 includes patients with stage I–III CRC who underwent potentially curative resection between 1997 and 2013 at Glasgow Royal Infirmary. Three cores were taken from each donor block at the invasive edge, capturing a small amount of tumour and the environment around it. Patient tissue access was authorized by the NHS Greater Glasgow and Clyde Biorepository under their NHS Research Ethics Committee, with ethics approval granted in biorepository application number 845, West of Scotland Ethics 22/WS/0207 in accordance with recognized ethical guidelines as described in the Declaration of Helsinki. Overall, 17 patients undergoing synchronous resection of primary CRC and CRC liver metastases with curative intent between April 2002 and June 2010 at Glasgow Royal Infirmary were analysed. Patient tissue access was authorized by the NHS Greater Glasgow and Clyde Biorepository under their NHS Research Ethics Committee, with ethical approval granted in biorepository application number 357, West of Scotland Ethics 22/WS/0207 in accordance with recognized ethical guidelines as described in the Declaration of Helsinki.

H-scores were generated for HNF4A, ATRX and CDX2 tumour cell expression using QuPath-0.2.3. Tumour cores with >2% tumour cells positive for LY6D and KRT5 expression were designated as LY6D$^+$ and KRT5$^+$, respectively. Tumour cores with <2% tumour cells positive for LY6D or KRT5 expression were designated as LY6D$^-$ and KRT5$^-$, respectively. Analysis was performed using QuPath-0.2.3. H-scores were generated for CDH1 cell expression using QuPath-0.2.3.

## Mouse organoid studies

Organoids were resuspended in Cultrex reduced growth factor BME type 2 (Bio-Techne), plated in a 10 µl drop on a culture plate and cultured in organoid culture medium containing advanced DMEM–F12 (Gibco) medium supplemented with 100 units ml$^{-1}$ penicillin, 100 µg ml$^{-1}$ streptomycin, 2 mM L-glutamine, 10 mM HEPES (all from Life Technologies), 1 ml Primocin (Invivogen), N2, B27 (both from Gibco), 50 ng ml$^{-1}$ EGF (Peprotech) and 1% Noggin conditioned medium. The Noggin-producing cell line was a gift from H. Clevers' group (Hubrecht Institute). Organoids were passaged by mechanical fragmentation with a p1000 and p200 pipette. All organoids were grown in a humidified incubator at 37 °C supplemented with 5% $CO_2$ Single cells were generated by incubating organoids in TrypLE Express (Life Technologies) for 15 min at 37 °C and passed through a 40 µm strainer. Cell counting was performed using a Countess II automated cell counter (Invitrogen). To study the effect of TGFβ, dissociated cells (10,000 (or 5,000 for the $Tgfbr2^{KO}$) cells in 10 µl BME) were cultured in organoid culture medium in the presence of 5 ng ml$^{-1}$ TGFβ (Peprotech). TGFβ treatment in Extended Data Fig. 3 was performed using a different range of concentrations as specified in the figure. On day 3 the medium was replaced with fresh organoid culture medium containing TGFβ. Cells were collected on day 13 for RNA extraction. Alternatively, cells were stained with calcein (Abcam) or phalloidin-568 (VWR) within 14 days of treatment.

For ITGA5$^+$ cell sorting and TGFβ treatment, AKP control and AKP $Atrx^{KO}$ organoids were digested as described above, and ITGA5 FACS was carried out as described in the FACS section below. In brief, 10,000 ITGA5$^+$ or ITGA5$^-$ cells were plated in 10 µl BME and treated with 5 ng ml$^{-1}$ TGFβ as described above. On day 13, organoids were fixed with 4% paraformaldehyde, stained with phalloidin-568 and analysed for the presence of spindle-like structures.

For the tail-vein injections, organoids were dissociated in TrypLE Express (Life Technologies) supplemented with a Rho kinase inhibitor (Y-27632, Tocris) for 15 min at 37 °C, passed through a 40 µm strainer,

counted and resuspended in PBS. For splenic injections, organoids were collected, washed in PBS, mechanically fragmented by vigorous pipetting and then incubated for 7 min at 37 °C in 0.25% trypsin in EDTA–PBS. After quenching of trypsinization by immersion in 10% FBS, cells were passed through a 40 μm strainer, counted using a haemocytometer and resuspended in PBS to achieve a final volume of $1 \times 10^7$ cells per ml. Cells were routinely tested for mycoplasma contamination.

## Human patient-derived organoids

The patient-derived organoids used in this study were generated by F.V.N.D. and M.G.D. MD20043 is an 81-year-old man with stage 4 rectal cancer: T3, N2, M1 (where 'T' is tumour, 'N' is nodes and 'M' is metastases). Ethics approval for human CRC organoid derivation was carried out under NHS Lothian Ethical Approval Scottish Colorectal Cancer Genetic Susceptibility Study 3 (SOCCS3) (REC reference: 11/SS/0109). The patient provided fully informed consent for the use of their tissues.

## Human organoid culture medium and *ATRX* KO generation

Human carcinoma organoids were cultured in advanced DMEM–F12 (Gibco) medium supplemented with 100 units ml$^{-1}$ penicillin, 100 μg ml$^{-1}$ streptomycin, 2 mM L-glutamine, 10 mM HEPES (all from Life Technologies), 1 ml Primocin (Invivogen), 1% Noggin conditioned medium (The Noggin-producing cell line was a gift from H. Clevers' group, Hubrecht Institute), B27 (Gibco), 50 ng ml$^{-1}$ EGF (Peprotech), 10 nM gastrin (Sigma), 10 nM PGE$_2$ (Tocris), 10 mM nicotinamide (Sigma), 10 μM SB202190 (Sigma), 600 nM A83-01 (Biotechne) and 12.5 mM *N*-acetylcysteine (Sigma).

For transduction conditions, human organoids were pretreated with IntestiCult (Stem Cell Tech) organoid growth medium and 1 mM valproic acid (Merk) for 48 h after 2 days post-split (day 1). On transduction day (day 1), human organoids were digested into a single-cell suspension in TrypLE Express (Life Technologies) with 10 μM Y-27632 (Tocris) for 8 min at 37 °C with mechanical dissociation every 4 min. Single-cell suspensions were combined with viral particles containing a non-targeting or *ATRX^KO* sgRNA (hATRX: 5′-GCTATAAACAGAAAAAGAAA-3′; pLentiV2-Addgene) and placed on a BME layer. On day 2, medium was changed into IntestiCult supplemented with 1 mM valproic acid and 10 μM Y-27632. On day 3, antibiotic selection started with IntestiCult supplemented with 10 μg ml$^{-1}$ blasticidin (Gibco) for 3 weeks. IntestiCult medium was used exclusively during transduction and the single-cell stage of organoid development for clone selection. Otherwise, the above-described medium was used for maintaining the organoids.

## In vitro drug treatment and cell-proliferation assay

Organoids were dissociated in TrypLE Express (Life Technologies) for 15 min at 37 °C, passed through a 40 μm strainer and resuspended in BME. Next, 1,000 single cells in 10 μl BME were plated in 24-well plates and treated with 250 nM JQ1 (Stratech), 0.25 ng ml$^{-1}$ IFNγ (Thermo Fisher Scientific), 10 nM FK228 (Stratech) or (0.5 ng ml$^{-1}$) TNF (Peprotech) for 7 days.

For Resazurin cell viability assays, Resazurin (R&D systems) was added at a volume equal to 10% of the cell culture medium volume and incubated at 37 °C. Fluorescence was read using 544 nm excitation and 590 nm emission wavelengths.

## RNA extraction, RT–qPCR and RNA-seq

Total RNA was isolated using a RNeasy Mini kit (Qiagen) accordingly to the manufacture's protocol. RNA was then subjected to DNA-free DNase treatment (Invitrogen). cDNA was generated using 1 μg RNA by reverse transcription using qScript cDNA SuperMix (Quantabio). RT–qPCR was performed using SYBR Select master mix (Applied Biosystems). $C_t$ values were normalized to β-actin or 18S rRNA. The $\Delta\Delta C_t$ method was used to calculate relative gene expression values. Oligonucleotides used in this study are listed in Supplementary Table 11. For the RNA-seq experiments, RNA integrity was evaluated using an Agilent 2200 Bioanalyser. Truseq mRNA-seq libraries were prepared from total AKP and AKP *ATRX^KO* RNA, and these were then sequenced using NovaseqS1 Illumina sequencing at Edinburgh Genomics Facility. AKP and AKP *Hnf4a^KO* libraries were prepared from 100 ng of each total RNA sample using a NEBNEXT Ultra II Directional RNA Library Prep kit (NEB 7760) and the Poly-A mRNA magnetic isolation module (NEB E7490) according to the provided protocol. Sequencing was performed on a NextSeq 2000 platform (Illumina, 20038897) using NextSeq 2000 P3 reagents (200 cycles) (20040560). RNA-seq analysis was carried out using the RaNA-seq pipeline with default settings[32].

## Western blotting

Cells were lysed using RIPA buffer (Sigma) supplemented with 1% phosphatase and protease inhibitors (Sigma). Protein concentration was measured using a BCA Protein Assay kit (Pierce). A total of 20 μg protein lysate was resuspended in 4× LDS sample buffer (Invitrogen) supplemented with sample reducing agent (Invitrogen) and denatured at 100 °C for 5 min. Proteins were separated by electrophoresis on NuPAGE 3–8% Tris-acetate protein gels (Invitrogen) using Tris-acetate buffer and blotted onto an activated PVDF or nitrocellulose (Cytiva) membrane at 100 V for 1.15 h. Membranes were incubated in blocking solution (5% milk, 0.1% Tween-20–PBS) for 1 h at room temperature, and then in primary antibody. The following primary antibodies were used: β-actin (Cell Signalling Technology, 1:5,000); ATRX (MABE1798, Sigma; 1:500); and HNF4A (C11F12, Cell Signalling Technology, 1:,1,000). After 3×10-min washes in 0.1% Tween-20–PBS, the membrane was incubated in HRP-linked secondary antibody for 1 h at room temperature. The following secondary antibodies were used: anti-rabbit or anti-mouse IgG HRP-linked (Cell Signalling Technology, 1:5,000). Following 3× 10-min washes in 0.1% Tween-20–PBS, antibody signals were detected by using ECL Plus Western blotting substrate (Pierce) and visualized using an ImageQuant 800 (GE Healthcare). Full scans are provided in Supplementary Information 1 and 2.

## CRISPR–Cas9 genome editing

sgRNAs (mAtrx: 5′-ACGGCGCATTAAGGTTCAAG-3′; mHnf4a 5′-CGG GCCACCGGCAAACACTA-3′, mTgfbr2: 5′-AAGCCGCATGAAGTC TGCG-3′, non-targeting controls 5′- GCTTTCACGGAGGTTCGACG-3′ or 5′-ATGTTGCAGTTCGGCTCGAT-3′) were cloned individually into lentiCRISPR v.2 plasmids (Addgene) following Addgene's protocol. Lentiviral particles were generated using HEK293T cells (provided by J. C. Acosta (IGMM, Edinburgh), originally obtained from the American Type Culture Collection): 10 μg gene-specific lentiviral vector was mixed with 7.5 μg lentiviral packaging vector psPAX2 and 2.5 μg envelope-protein-producing vector pCMV-VSV-G (both from Addgene) and transfected into HEK293T cells in a 10 cm$^2$ dish using polyethylenimine as the transfection reagent (Polysciences). After 48 h, the supernatant medium was filtered using a 0.45 μm syringe filter and concentrated using a Lenti-X concentrator (Takara Bio). Lentiviral transduction was carried out as previously described[33]. In brief, AKP organoids were expanded and cultured in organoid culture medium supplemented with 10 μM Rho kinase inhibitor (Y-27632, Tocris) and 1 mM valproic acid (Sigma) for 48 h. Spheroids were enzymatically dissociated with StemPro Accutase cell dissociation reagent (Gibco) supplemented with 10 μM Y-27632 for 3 min at 37 °C. Dissociated organoids were then washed twice with advanced DMEM–F12 medium. Cells were counted using a Countess II Automated cell counter (Invitrogen). Next, $5 \times 10^5$ cells were plated on a 150 μl bed of BME in a 6-well plate in the presence of Y-27632, 1 mM valproic acid and 4 μg ml$^{-1}$ polybrene (Sigma). Virus was removed 24 h after transduction, and adhered organoids were overlaid with 150 μl BME. To select transduced cells, 2 μg ml$^{-1}$ of puromycin or 10 μg ml$^{-1}$ blasticidin (both from Gibco) was added to the organoid culture medium supplemented with 10 μM Y-27632. Multiple deleted clones were generated. Editing of clonal lines was confirmed by genomic sequencing or western blotting.

## TissueEnrich analysis

TissueEnrich analysis[34] (https://tissueenrich.gdcb.iastate.edu/) was carried out using the web-based tool with the following settings: gene symbol, *Homo sapiens*, Human Protein Atlas, All. Lists of gene upregulated or downregulated by >2 fold in the AKP versus AKP *Atrx*[KO] dataset were used for input. In Fig. 2g, TissueEnrich analysis was performed on the list of genes upregulated in scRNA-seq cluster 4 and cluster 15 (fold change > 1.5).

## *ATRX* mutation analysis in CRC transcriptional subtypes

For *ATRX* mutation enrichment analysis, we used previously designated CRIS subtypes[14] and analysed TCGA mutational data for *ATRX* mutations for which the CRIS designation is known. The number of tumours carrying *ATRX* mutations was calculated for the CRIS-B subtype and compared with all other subtypes using Fishers' exact tests. For survival analysis, only CRIS-B tumours were analysed using TCGA survival data downloaded from the TCGA data portal. For iCMS, the same analysis was carried out but with TCGA data separated on the basis of iCMS designation.

## HNF4A overexpression

HNF4A overexpression was generated using the pGCDNsam-HNF4A-IRES-GFP plasmid (Addgene). In brief, AKP *Atrx* organoids were transduced as above. To select transduced cells, FACS analysis for GFP positivity was performed.

## Collection of human samples and processing for FACS

Normal colorectal mucosa and tumour were sampled from freshly resected surgical specimens from patients diagnosed with CRC. Ethics approval was carried out under NHS Lothian Ethical Approval Scottish Colorectal Cancer Genetic Susceptibility Study 3 (SOCCS3 REC: 11/SS/0109, IRAS: 9556). All patients provided fully informed consent for the use of their tissues. Tissues were cut into small pieces and then incubated in Advanced DMEM–F12 supplemented with 1 mg ml$^{-1}$ collagenase type IV (Sigma), 0.5 mg ml$^{-1}$ hyaluronidase (Sigma) and 10 μM Y-27632 (Tocris) at 37 °C with vigorous shaking until the tissue was completely disaggregated (60–90 min). The digested reaction was then filtered through a 70 μm cell strainer. The filtered cells were centrifuged at 500*g* for 5 min, washed twice in Advanced DMEM–F12 and once in 0.1% BSA in PBS. Single-cell suspensions were then analysed by FACS.

## FACS

Pelleted organoids were resuspended in 1 ml TrypLE Express (Gibco) and incubated at 37 °C for 15 min. Cells were vigorously dissociated by pipetting, resuspended in 10 ml advanced DMEM–F12, passed through a 40 μm cell strainer and centrifuged at 300*g* for 5 min at 4 °C. Single cells were washed with 0.1% BSA in PBS and stained with the following antibodies: EPCAM–APC (BioLegend, 118213; 1:200); LY6D–PE (BioLegend, 138603; 1:200) or LY6D–APC (Miltenyi, 130-115-313; 1:50); and ITGA5–PE (BioLegend, 103805; 1:200). Human single-cell suspensions were stained with EPCAM–APC (BioLegend, 324207; 1:50) and LY6D–FITC (Cusabio Biotech, CSB-PA613492LC01HU; 1:50). Cells were then washed twice in 0.1% BSA in PBS before being subjected to FACS (BD FACSARIA II/BD LSR-Fortessa X-20). Single viable cells were gated by negative staining for DAPI. The gating strategy is provided in Supplementary Information 3. Analyses were performed using FlowJo (v.10.8) software.

## scRNA-seq data processing

Raw sequencing reads were processed and aligned to the mouse reference genome (mm10) using the 10x Genomics CellRanger pipeline (v.7.2.0). The gene expression matrices obtained from CellRanger were analysed using the R package Seurat (v.5). Cell barcodes with <200 unique genes and >10% mitochondrial gene expression were removed. The filtered matrices were normalized to the total unique molecular

identifier counts per cell, and cell cycle effects were regressed out. Data integration was performed with Harmony, and the subsequent Louvain clustering resulted in 18 clusters. Differentially expressed genes between clusters were obtained using Wilcoxon rank-sum tests implemented by the FindAllMarkers function in Seurat.

## ATAC–seq

ATAC–seq was performed using an Active Motif commercially available kit as per the manufacturer's instructions. In brief, 50,000 cells were lysed with ATAC lysis using a pestle and dounced slowly for 25 strokes on ice followed by Tn5 tagmentation for 30 min at 37 °C. Size selection was performed using the SPRIselect protocol (Beckam). After indexing and PCR amplification, DNA libraries were multiplexed and sequenced with an Illumina Nextseq 2000 on a 100-cycles kit by Edinburgh Clinical Research Facility–Wellcome Trust CRF.

## ATAC–seq analysis

Raw sequencing output fastq files were inputed into an nf-core ATAC–seq pipeline using default pipeline settings (https://nf-co.re/atac-seq/2.1.2)[35]. ATAC–seq library quality was assessed for the presence of contaminating mitochondrial DNA sequences. Pipeline output bigWig files were used for creating locus-specific genome images from the IGV browser (https://www.igv.org/). ATAC–seq peaks were analysed for differential enrichment between samples using DESeq2 (ref. 36). Principal component analysis was performed on tables of ATAC-seq read counts (DESeq2 output) using base R scripts. Subsequently, DESeq2 ATAC–seq read counts were input into the monaLisa R package[37] to determine the presence of differentially enriched TF-binding or DNA-binding site motifs between sample groups.

## CUT&RUN

Cells were digested with TrypLE Express at 37 °C for 15 min. Next, $1 × 10^6$ viable cells were fixed in PBS containing 0.1% formaldehyde at room temperature for 1 min. Formaldehyde was quenched with glycine at 0.125 M, followed by 5 min of incubation at room temperature. Fixed cells were pelleted and washed twice with room temperature wash buffer (20 mM HEPES, 150 mM NaCl, 0.5 mM spermidine (Sigma-Aldrich), 1% Triton X-100, 0.05% SDS and 1× protease inhibitor cocktail (Roche)). Cells were pelleted and resuspended in 100 μl wash buffer. The cell suspension was added at a 10:1 ratio to concanavalin A beads (CST) activated as per the manufacturer's instructions, gently mixed and incubated at room temperature for 20 min. The cell–bead slurry was placed on a magnetic rack until clear, the supernatant removed and samples resuspended in cold antibody buffer (20 mM HEPES, 150 mM NaCl, 0.5 mM spermidine, 0.0025% digitonin (Thermo Fisher), 2 mM EDTA, 1% Triton X-100, 0.05% SDS and 1× protease inhibitor cocktail (Roche)). The following antibodies were added and incubated at 4 °C overnight: XP isotype control DA1E (rabbit; CST 1:10) or anti-histone H3, acetyl K27 (rabbit; Abcam 1:250). Samples were washed twice with digitonin buffer (20 mM HEPES, 150 mM NaCl, 0.5 mM spermidine, 0.0025% digitonin, 1% Triton X-100, 0.05% SDS and 1× protease inhibitor cocktail (Roche)) and resuspended in digitonin buffer. 1× CUTANA pAG-MNase (EpiCypher) was added and incubated for 10 min at room temperature. Samples were washed twice with digitonin buffer. Chromatin was cleaved by the addition of 2 mM CaCl$_2$ and incubation for 2 h at 4 °C. Digestion was halted by the addition of stop buffer (340 mM NaCl, 20 mM EDTA, 4 mM EGTA and 50 μg ml$^{-1}$ glycogen (ThermoFisher)) and chromatin released by incubation at 37 °C for 10 min. The supernatant was collected and SDS added to a final concentration of 0.09%. Proteinase K (CST) was added and incubated overnight at 55 °C. DNA was purified using MaXtract High Density columns (Qiagen) as per the manufacturer's instructions and precipitated with GlycoBlue (ThermoFisher) as per the manufacturer's instructions. DNA was eluted in TE buffer (ThermoFisher). Samples were quantified and assessed for size distribution using an Agilent 2100 Electrophoresis Bioanalyser

Instrument with a DNA HS kit (Agilent Technologies). Libraries were generated using a Simple-ChIP DNA library prep kit (CST) and quantified by fluorometry using a Qubit dsDNA HS assay and assessed for size distribution on an Agilent Bioanalyser with a DNA HS kit. Next, 100 bp paired-end sequencing was performed on a NextSeq 2000 platform (Illumina) using a NextSeq 1000/2000 P1 reagents (300 cycles) kit.

## CUT&RUN data analysis

The nf-core CUT&RUN analysis pipeline (https://doi.org/10.5281/zenodo.7715959) was used to process CUT&RUN data using default parameters. Reads were normalized to spike in DNA (CST) and aligned to the mm10 reference genome. Regions in the mm10 ENCODE blacklist were removed[38]. MACS2 (ref. 39) was used for peak calling and peak calls were normalized to IgG controls. Differential analysis was conducted using DiffBind[40] with default parameters. BEDTools[41] was used to identify overlapping regions of significant difference in the H3K27ac CUT&RUN with the ATAC dataset.

## TCGA patient stratification based on *ATRX* transcriptional signatures

Bulk RNA transcripts per kilobase million values for patients with colorectal adenocarcinoma from TCGA[27] were obtained from the Genomic Data Commons Portal and were $\log_2$ normalized. Patients who had ATAC−seq data ($n = 36$) were stratified into high, medium or low for *Atrx$^{KO}$* and *Atrx$^{WT}$* signatures. This was performed by calculating the single-sample GSEA score using GSVA in R and selecting the samples above the third, second or first quantile for the respective category. These patients were also assigned an iCMS class as previously described[23].

## Differential analysis of TCGA bulk ATAC−seq data

Normalized ATAC−seq peak counts were obtained from the Genomic Data Commons Portal for 77 samples from 36 patients with colorectal adenocarcinoma; details on the normalization can be found the original publication[27]. In brief, the peak counts matrix was normalized using 'cpm(matrix, log = TRUE, prior.count = 5)' in edgeR followed by quantile normalization using normalize.quantiles of preprocessCore in R. A SummarizedExperiment object was created with the normalized count matrix and metadata for the patients. A two-tailed $t$-test was used to identify the peaks that have a significantly different mean count between the samples from patients categorized as HiSquam (high for *Atrx$^{KO}$* and low for *Atrx$^{WT}$*) or HiCol (low for *Atrx$^{KO}$* and high for *Atrx$^{WT}$*). The selection of differential peaks was based on FDR < 0.01 and $\Delta\log_2$counts > 1 cut-off values. Counts from all samples for the set of significantly differential peaks were plotted into a heatmap using ComplexHeatmap in R.

## HOMER TF motif enrichment

TF motif enrichment analysis was performed on the set of peaks that were differentially accessible for patients categorized as HiSquam or HiCol by using HOMER. The peaks were first annotated using ChIPeakAnno and then formatted into HOMER input style. The analysis was performed with the command findMotifsGenome.pl, genome "hg38" and "-size 200 -mask" as options. TF motif enrichment was presented as previously described[42].

## Analysis of bulk gene expression data with *Atrx$^{WT}$* and *Atrx$^{KO}$* signature genes

Normalized gene expression data of samples from patients with CRC were obtained from the Gene Expression Omnibus (accession GSE39582), and 557 patients with relapse survival information were used for the downstream analysis. HiCol, intermediate and HiSquam groups were identified by hierarchical clustering using *Atrx$^{WT}$* and *Atrx$^{KO}$* signature genes. The relapse-free survival of three patient groups was determined by Kaplan–Meier survival with log-rank tests.

## Identification of *Atrx$^{WT}$* and *Atrx$^{KO}$* signatures from single-cell resolution

To understand the association between *ATRX$^{KO}$*-driven gene expression and iCMS classification at the single-cell level, we performed the analysis using scRNA-seq data after downloading from syn26844071 (ref. 23) and used tumour cells with iCMS classification from primary tumour tissue samples after excluding patients with low numbers of cells. The dimension reduction analysis was performed using 715 iCMS-associating genes, and patient batch was corrected using Harmony[43]. *Atrx$^{WT}$* and *Atrx$^{KO}$* scores for each cell were calculated using the average expression level of each signature and then subtracted with the aggregated expression of the control gene score sets. The control gene score was calculated using the average expression of 100 randomly selected genes, replicated 10 times.

## Statistical analysis

Statistical analysis was performed using GraphPad Prism 9. Statistical tests used are indicated in figure legends and exact $P$ values shown throughout.

## Reporting summary

Further information on research design is available in the Nature Portfolio Reporting Summary linked to this article.

## Data availability

The CUT&RUN, ATAC−seq, scRNA-seq and RNA-seq data generated in this study have been deposited into the Genome Sequence Archive (https://ngdc.cncb.ac.cn/gsa/) website with the following accession numbers: CRA024850 (H3K27ac AKP versus AKP ATRX); CRA024849 (ATAC-seq AKP versus AKP ATRX); CRA024816 (scRNA-seq AKP versus AKP ATRX); CRA024804 (RNA-seq AKP versus AKP HNF4A); and CRA024763 (RNA-seq AKP versus AKP ATRX). Source data are provided with this paper.

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

**Acknowledgements** This work was funded by the Cancer Research UK (CRUK) under a Career Development Fellowship (A19166 to K.B.M.) and a Small Molecule Drug Discovery Project Award (A25808 to K.B.M.), a European Research Council under Starting Grant (COLGENES–715782 to K.B.M.) and the MRC (project grant MR/X008762/1 to K.B.M.). C.W.S. is funded by a UKRI Future Leaders Fellowship (MR/W007851/1). O.J.S. is funded by a CRUK ACRCelerate: Colorectal Cancer Stratified Medicine Network Accelerator Award (A26825). We thank technical staff at the University of Edinburgh's Institute of Genetics and Cancer (IGC) for providing support for some of the experiments; staff at the Flow Cytometry service for support with FACS; the animal technicians at the Biomedical Research Facility (BRF) facility for animal husbandry support; NHS Lothian staff and patients for support obtaining human tissue biopsies; B. Callaghan for helping N.P. with cell proliferation assays; J. Roper and O. Yilmaz for the gift of AKP organoids; and G. Brien for advice on CUT&RUN assays.

**Author contributions** The project was conceived and experiments planned by M.R., P.C. and K.B.M. Experiments were conducted by P.C., M.R., N.P., J.F., M.W., C.V.B., S.Ö.-G.P., N.J.D., J.Q., A.B.A. and K.B.M. Bioinformatic analyses were conducted by Y.H. and F.D.V.-B. (human data), D.S.D. (ATAC–seq), N.P. (CUT&RUN), N.T.Y. and Y.Z. (scRNA-seq), S.B.M., P.D.D. and K.P. J.Q. stained the in-house TMAs. C.V.B. performed mouse work, prepared tissue histology and lentivirus. P.C., N.J.D. and C.V.B. performed the orthotopic transplantations. J.F. and M.W. performed mouse work. R.G., M.G.D. and F.V.N.D. provided the patient-derived samples. R.R.M., O.J.S., J.E., M.G.D., F.V.N.D., S.T., C.W.S. and K.B.M. supervised the project. The manuscript was prepared by P.C. and K.B.M., and all authors read and approved it.

**Competing interests** The authors declare no competing interests.

**Additional information**
**Correspondence and requests for materials** should be addressed to Kevin B. Myant.

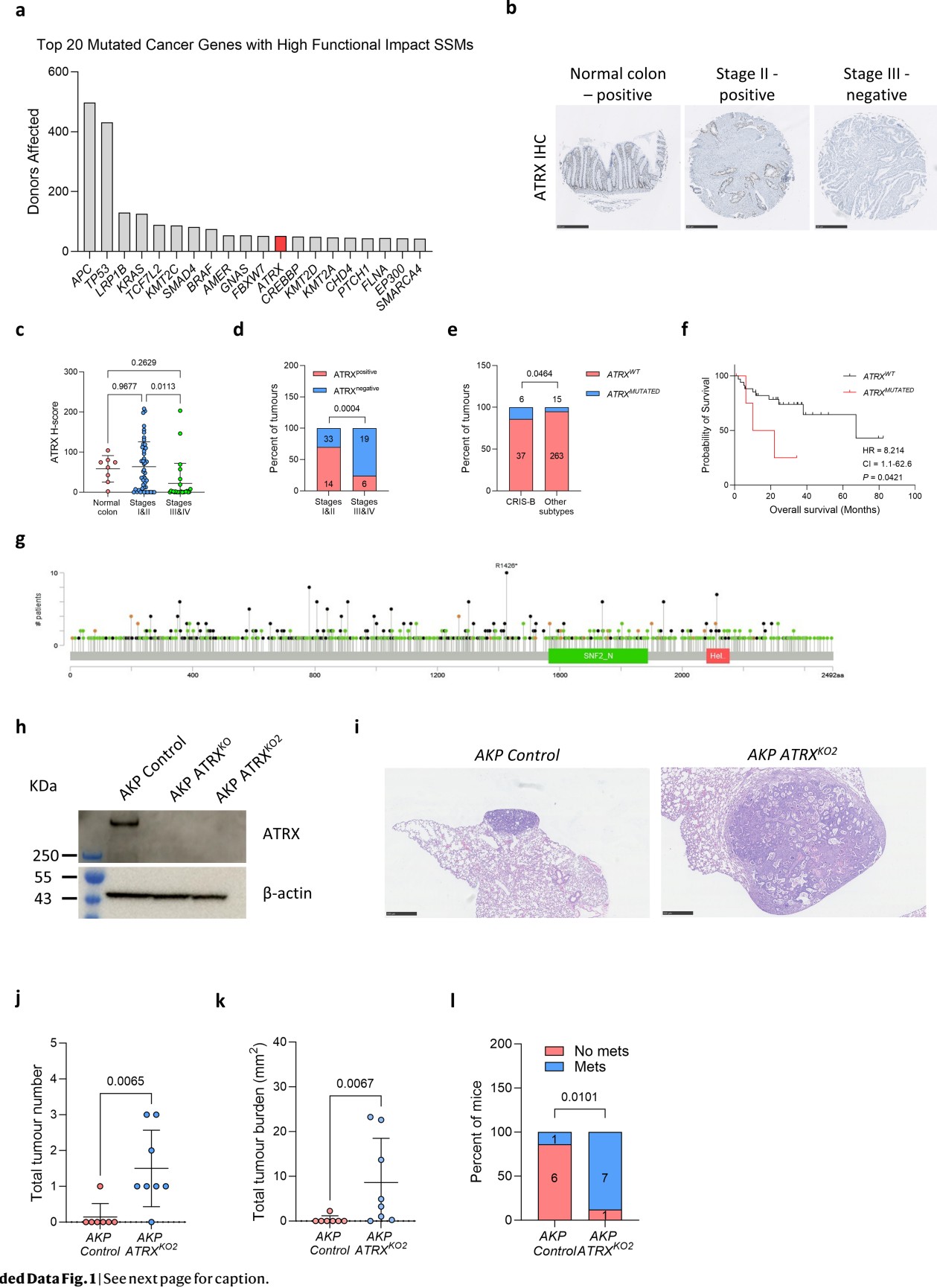

**a** Top 20 Mutated Cancer Genes with High Functional Impact SSMs

**b** ATRX IHC: Normal colon – positive; Stage II – positive; Stage III – negative

**Extended Data Fig. 1** | See next page for caption.

**Extended Data Fig. 1 | Loss of ATRX expression is associated with metastasis.** (a) International Cancer Genome Consortium data of top 20 mutated cancer genes with high functional impact mutations in colorectal cancer (CRC). (b) Representative ATRX staining of human normal and CRC tissue microarray. Examples of positive and negative staining are shown. Scale bars, 500 μm. (c) Quantification of ATRX expression using immunohistochemistry H-score method analysed using QuPath. Samples are separated into normal, non-metastatic (stage I and II) and metastatic (stage III and IV) groups, n = 8 vs 47 vs 25 tumours. (d) Summary data indicating presence (H-score > 10) or absence (H-score <10) of ATRX staining in non-metastatic and metastatic samples. Number of tumours in each group indicated on graph, n = 47 vs 25 tumours. (e) Summary data indicating presence or absence of *ATRX* mutation in CRIS-B vs all other CRIS transcriptional subtypes. Data extracted from TCGA dataset where CRIS tumour annotation is known. Number of tumours in each group indicated on graph, n = 43 vs 278 tumours. (f) Overall survival data of patients with CRIS-B tumours separated on presence or absence of *ATRX* mutation. Data extracted from TCGA dataset, n = 37 vs 6 patients. For (c) data are mean ± SD. P values were calculated using ordinary one-way ANOVA with multiple comparisons. For (d) and (e) p values were calculated using two-sided Fisher's exact test. For (f) P value was calculated using Log-rank (Mantel-Cox) test. (g) Lollipop plot of TCGA PanCancer mutational data for *ATRX*. *ATRX* mutations were analysed using cBioPortal (07/12/23) with TCGA PanCancer Atlas Studies selected. (h) Western-blot analysis of *AKP* ATRX[KO] organoids for ATRX and β-actin. n = 2 technical replicates. (i) Representative images of haematoxylin and eosin (H&E) stained lung metastases in mice injected with *AKP Control* or *AKP Atrx[KO2]* organoid cells via tail vein. Scale bars, 500 μm. (j) Quantification of number of lung metastases per mouse, n = 7 vs 8 mice. (k) Quantification of total lung tumour burden per mouse, n = 7 vs 8 mice. (l) Summary data indicating presence or absence of lung metastases. Number of mice with lung metastases or no metastases indicated on graph, n = 7 vs 8 mice. For (j) and (k) data are mean ± SD. P values were calculated using two-tailed Mann-Whitney test. For (l) p value was calculated using two-sided Fisher's exact test.

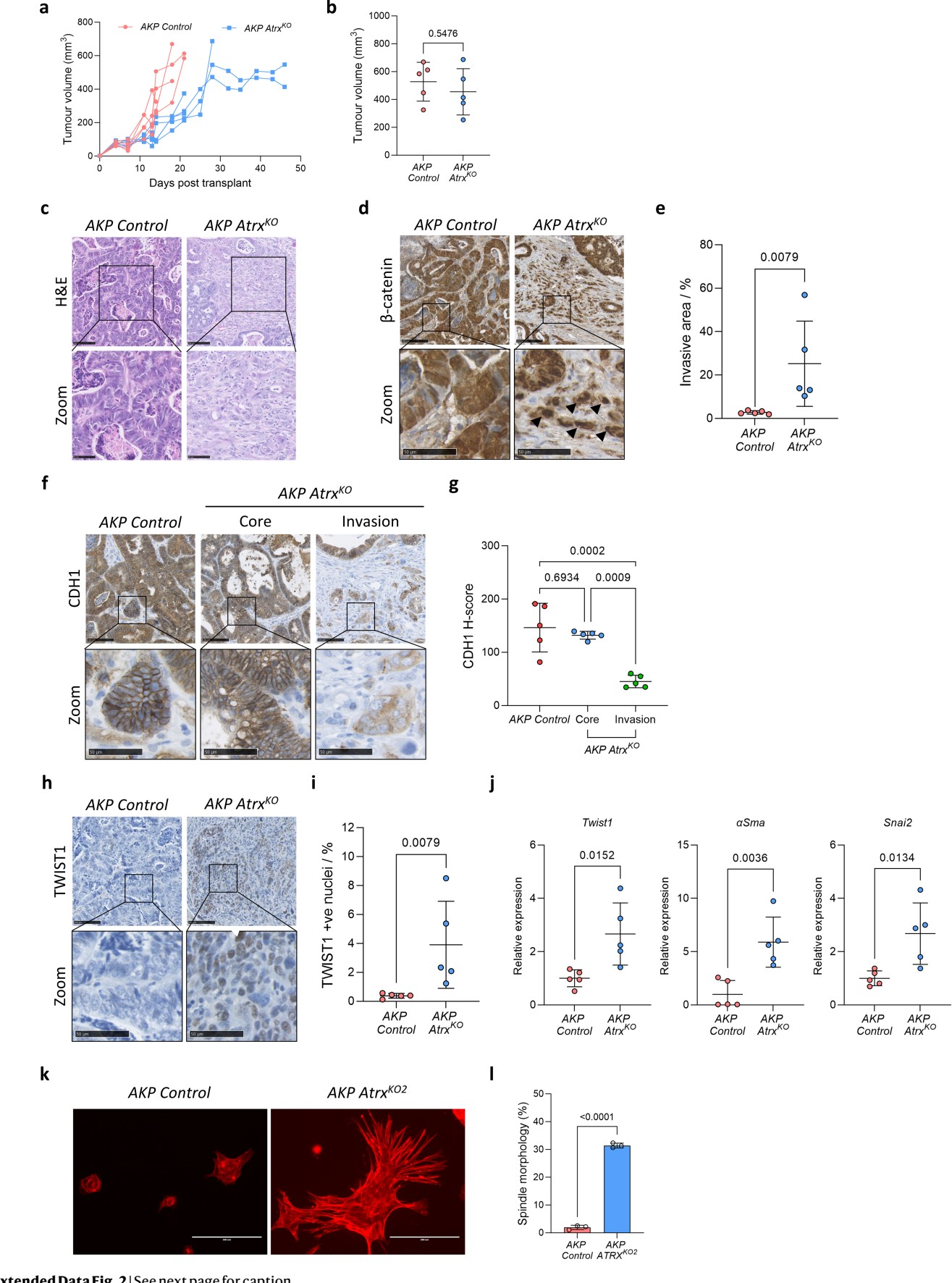

**Extended Data Fig. 2** | See next page for caption.

**Extended Data Fig. 2 | *Atrx* deletion leads to tumour invasion.**
(a) Subcutaneous tumour volume growth of mice injected with *AKP Control* or *AKP Atrx$^{KO}$* organoid cells. (b) Volumes of tumours generated via subcutaneous injection of *AKP Control* or *AKP Atrx$^{KO}$*, n = 5 vs 5 tumours. (c) Representative images of H&E-stained subcutaneous tumours in mice subcutaneously injected with *AKP Control* or *AKP Atrx$^{KO}$* organoid cells. Scale bars, 100 μm overview and 50 μm zoom. (d) Representative images of β-catenin-stained subcutaneous tumours in mice subcutaneously injected with *AKP Control* or *AKP Atrx$^{KO}$* organoid cells. β-catenin staining is used to identify tumour cells. Black arrows indicate β-catenin positive tumour cells invading into surrounding stroma. Scale bars, 100 μm overview and 50 μm zoom. (e) Quantification of invasive area of *AKP Control* vs *AKP Atrx$^{KO}$* tumours, n = 5 vs 5 tumours. (f) Representative images of CDH1 stained subcutaneous tumours in mice subcutaneously injected with *AKP Control* or *AKP Atrx$^{KO}$* organoid cells. Tumour core and invasive region of *AKP Atrx$^{KO}$* tumour shown. Scale bars, 100 μm overview and 50 μm zoom. (g) Quantification of CDH1 staining intensity (H-score analysed with QuPath) in *AKP Control* vs *AKP Atrx$^{KO}$* tumours, n = 5 vs 5 tumours. (h) Representative images of TWIST1 stained subcutaneous tumours in mice subcutaneously injected with *AKP Control* or *AKP Atrx$^{KO}$* organoid cells. Scale bars, 100 μm overview and 50 μm zoom. (i) Quantification of TWIST1 staining in *AKP Control* vs *AKP Atrx$^{KO}$* tumours, n = 5 vs 5 tumours. (j) RT-qPCR analysis of EMT marker expression in *AKP Control* vs *AKP Atrx$^{KO}$* tumours, n = 5 vs 5 tumours. For (b), (e) and (i) data are mean ± SD. P values were calculated using two-tailed Mann-Whitney test. For (g) data are mean ± SD. P values were calculated using ordinary one-way ANOVA with multiple comparisons. For (j) gene expression was normalised to *Actb* and levels relative to *AKP Control* tumours calculated using the ΔΔCt method. Data are mean ± SD. P values were calculated using two-tailed student's t test. (k) Fluorescence microscopy of phalloidin stained *AKP Control* and *AKP Atrx$^{KO2}$* organoids following treatment with 5 ng/ml TGF-beta. Scale bars, 200 μm. (l) Quantification of the percentage of *AKP Control* vs *AKP Atrx$^{KO2}$* organoids adopting spindle-like morphology following TGF-beta treatment, n = 3 vs 3 independent experiments. Data are mean ± SD. P value was calculated using two-tailed student's t test.

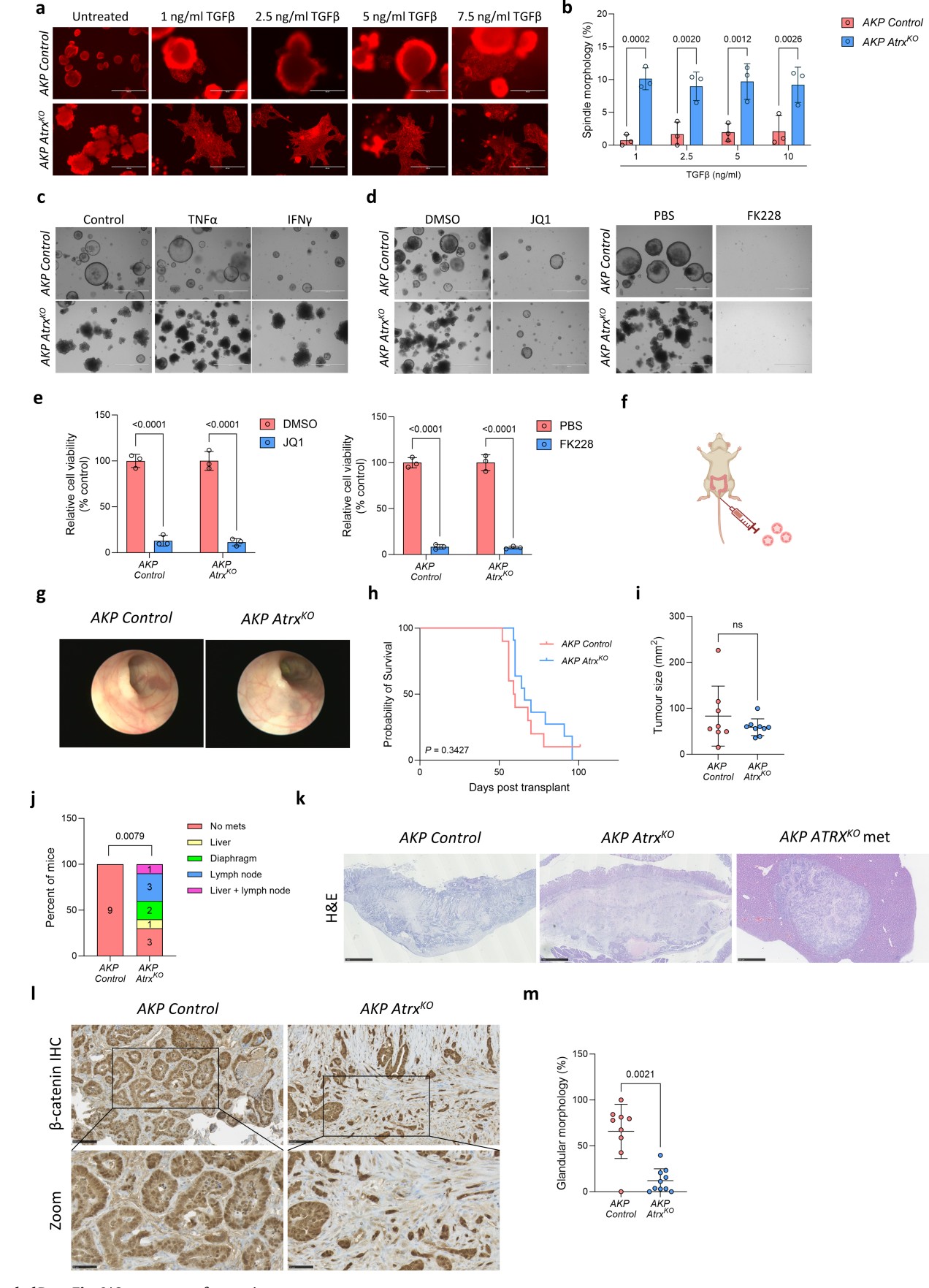

**Extended Data Fig. 3** | See next page for caption.

**Extended Data Fig. 3 | Deletion of *Atrx* promotes metastasis.**
(a) Fluorescence microscopy of phalloidin stained *AKP Control* and *AKP Atrx^{KO}* organoids following concentration gradient of TGF-beta (TGFβ). Scale bars, 400 μm. (b) Quantification of the percentage of *AKP Control* vs *AKP Atrx^{KO}* organoids adopting spindle-like morphology following TGF-beta gradient treatment, n = 3 vs 3 independent experiments. Data are mean ± SD. P values were calculated using ordinary two-way ANOVA with Sidak's multiple comparisons. (c) Representative images of *AKP Control* and *AKP Atrx^{KO}* organoids treated with TNFα (0.5 ng/mL), IFNγ (0.25 ng/mL) or control for 8 days, n = 3 vs 3 independent experiments. (d) Representative images of *AKP Control* and *AKP Atrx^{KO}* treated with JQ1 (250 nm), FK228 (10 nM) or control for 8 days. n = 3 vs 3 independent experiments. (e) Relative cell viability following JQ1 or FK228 treatment in *AKP Control* vs *AKP Atrx^{KO}* organoids, n = 3 vs 3 independent experiments. Data are mean ± SD. P values (DMSO vs JQ1 AKP control (p = 0.00000043), DMSO vs JQ1 AKP ATRX (p = 0.00000037). DMSO vs FK228 AKP control (p = 0.00000003), DMSO vs FK228 AKP ATRX (p = 0.00000003) were calculated using ordinary two-way ANOVA with multiple comparisons. (f) Cartoon illustrating organoids orthotopic colonic injection. Created with BioRender. (g) Representative colonoscopy images of mice orthotopically transplanted with *AKP Control* or *AKP Atrx^{KO}* organoids, 7-weeks post transplantation, n = 9 vs 10 mice. (h) Survival plot for mice orthotopically transplanted with *AKP Control* or *AKP Atrx^{KO}* aged until clinical end-points. (i) Total tumour area of primary tumours per mouse of the shown genotypes, n = 8 vs 9 mice. (j) Summary data indicating absence, presence, and site of metastases per genotype. Number of mice with metastases or no metastases indicated on graph, n = 9 vs 10 mice. (k) Representative images of H&E-stained primary tumours and liver metastasis in mice orthotopically transplanted with *AKP Control* and *AKP Atrx^{KO}* organoid cells. n = 9 vs 10 biologically independent samples. Scale bars, 1 mm for primary tumours and 500 μm for liver metastasis. (l) Representative β-catenin-stained images of primary tumours in mice orthotopically transplanted with *AKP Control* or *AKP Atrx^{KO}* organoids. β-catenin is used to identify tumour cells. Magnification highlights the distinct morphological features. Scale bars, 100 μm overview, 50 μm zoom. (m) Quantification of glandular morphology areas of *AKP Control* vs *AKP Atrx^{KO}* primary tumours, n = 9 vs 10 mice. For (h) p value was calculated using Long-rank (Mantel-Cox) test. For (i) and (m) data are mean ± SD. P values were calculated using two-tailed Mann-Whitney test. For (j) p value is calculated using Fisher's exact test.

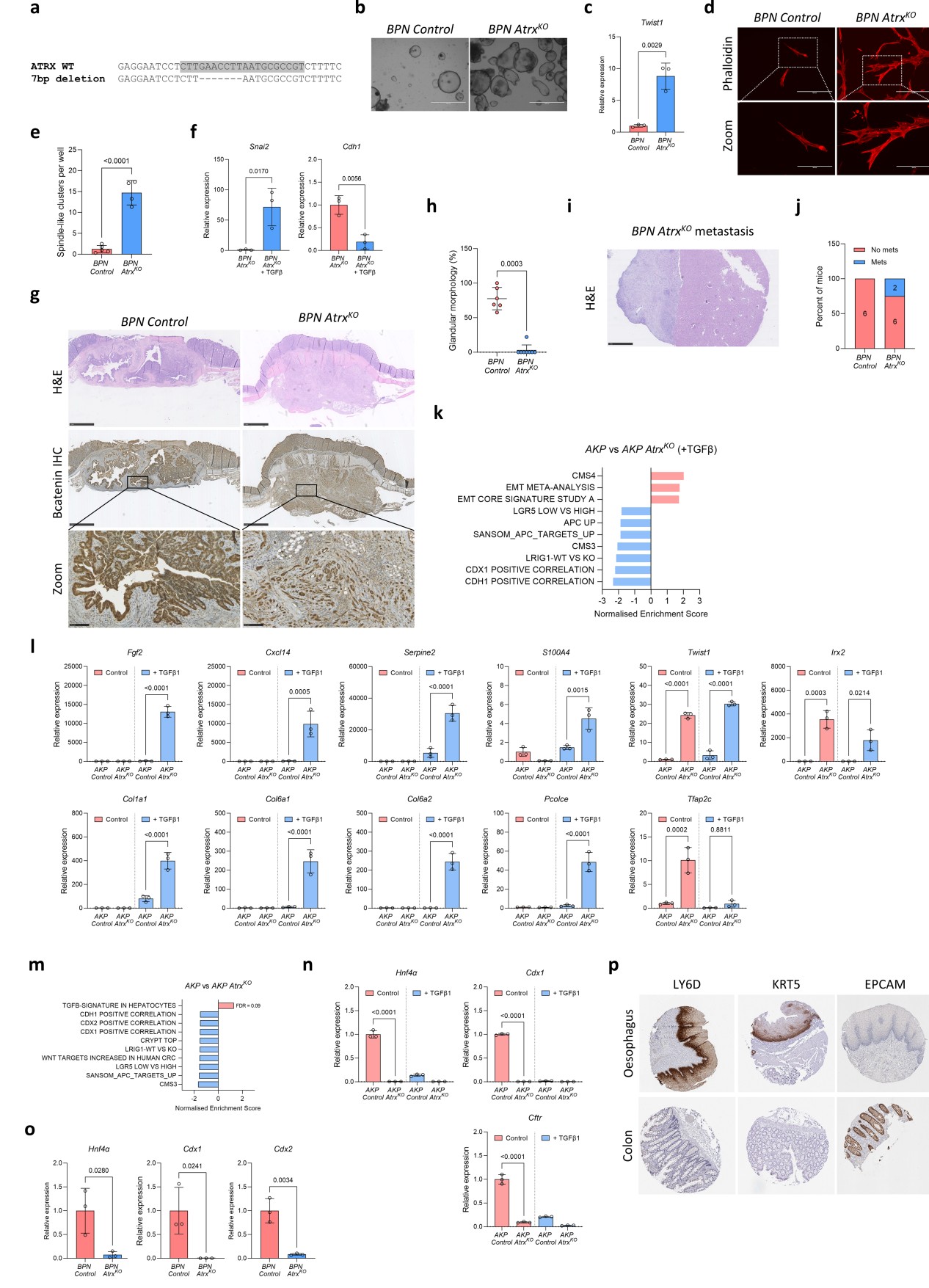

**Extended Data Fig. 4** | See next page for caption.

**Extended Data Fig. 4 | *Atrx* deletion in *BPN* model results in tumour invasion.** (a) Targeted DNA sequencing of targeted *Atrx* locus in *BPN Atrx^{KO}* organoid lines. (b) Representative pictures of *BPN Control* and *BPN Atrx^{KO}* organoid culture. Scale bars, 1000 μm. (c) RT-qPCR analysis of the EMT marker *Twist1* in *BPN Control* vs *BPN Atrx^{KO}* organoid culture, n = 3 vs 3 independent experiments. (d) Fluorescence microscopy of phalloidin stained *BPN Control* and *BPN Atrx^{KO}* organoids following treatment with 5 ng/ml TGF-beta. Zoomed area outlined in white box. Scale bars, 400 μm overview and 200 μm zoom. (e) Average number of clusters of *BPN Control* and *BPN Atrx^{KO}* organoids per well exhibiting a spindle-like morphology following TGF-beta treatment, n = 5 vs 4 independent experiments. (f) RT-qPCR analysis of EMT and epithelial marker expression in *BPN Atrx^{KO}* organoids either untreated or treated with 5 ng/ml TGF-beta, n = 3 vs 3 independent experiments. (g) Representative H&E and β-catenin-stained images of primary tumours in mice orthotopically transplanted with *BPN Control* or *BPN Atrx^{KO}* organoid cells. β-catenin is used to identify tumour cells. Magnification highlights the distinct morphological features. Scale bars, 1 mm overview and 100 μm zoom, n = 6 vs 8 mice. (h) Quantification of tumour areas with glandular morphology in *BPN Control* and *BPN Atrx^{KO}* primary tumours, n = 6 vs 8 mice. (i) Representative H&E-stained image of *BPN Atrx^{KO}* metastasis. Scale bars, 1 mm. (j) Summary data indicating presence or absence of metastases. Number of mice with metastases or no metastases indicated on graph, n = 6 vs 8 mice. For (c) and (f) gene expression was normalised to *Actb* and relative levels were calculated using the ΔΔCt method. Data are mean ± SD. For (c), (e) and (f) p values were calculated using two-tailed student's t test. For (e) p = 0.000023. For (h) p value was calculated using two-tailed Mann-Whitney test. (k) Normalised enrichment scores of significantly enriched gene sets in TGF-beta treated *AKP Control* vs *AKP Atrx^{KO}* organoids. (l) RT-qPCR analysis of representative EMT markers induced in *AKP Atrx^{KO}* organoids following treatment with TGF-beta, n = 3 vs 3 independent experiments. (m) Normalised enrichment scores of significantly enriched gene sets in untreated *AKP Control* vs *AKP Atrx^{KO}* organoids. (n) RT-qPCR analysis of representative genes important for colonic lineage specification and function, n = 3 vs 3 independent experiments. (o) RT-qPCR analysis of representative genes important for colonic lineage specification and function in *BPN Control* vs *BPN Atrx^{KO}* organoids, n = 3 vs 3 independent experiments. For (l) gene expression was normalised to *18S rRNA* and levels relative to untreated *AKP* control organoids calculated using the ΔΔCt method. *For* Twist1, Irx2 and Tfap2c gene expression was normalised to *Actb* and levels relative to untreated *AKP* control organoids calculated using the ΔΔCt method. For (n) gene expression was normalised to *Actb* and levels relative to untreated *AKP* Control organoids calculated using the ΔΔCt method. For (o) gene expression was normalised to *Actb* and levels relative to *BPN Control* organoids calculated using the ΔΔCt method. For (l) and (n) data are mean ± SD. P values were calculated using ordinary one-way ANOVA with multiple comparisons. For (l) *Fgf2* p = 0.00000006, *Serpine2* p = 0.000006, *Col1a1* p = 0.000005, *Col6a1* p = 0.000045, *Col6a2* p = 0.000004, *Pcolce* p = 0.000015, *Twist1* control p = 0.00000026, *Twist1* + TGFβ1 p = 0.00000004. For (n) Hnf4a control p = 0.00000001, *Cdx1* control p = 0.000000000000051, *Cftr* control p = 0.00000008. For (o) data are mean ± SD. P values were calculated using two-tailed student's t test. (p) Representative images of LY6D, KRT5 and EPCAM stained oesophagus and colon from Human Protein Atlas (Human Protein Atlas proteinatlas.org).

**a**

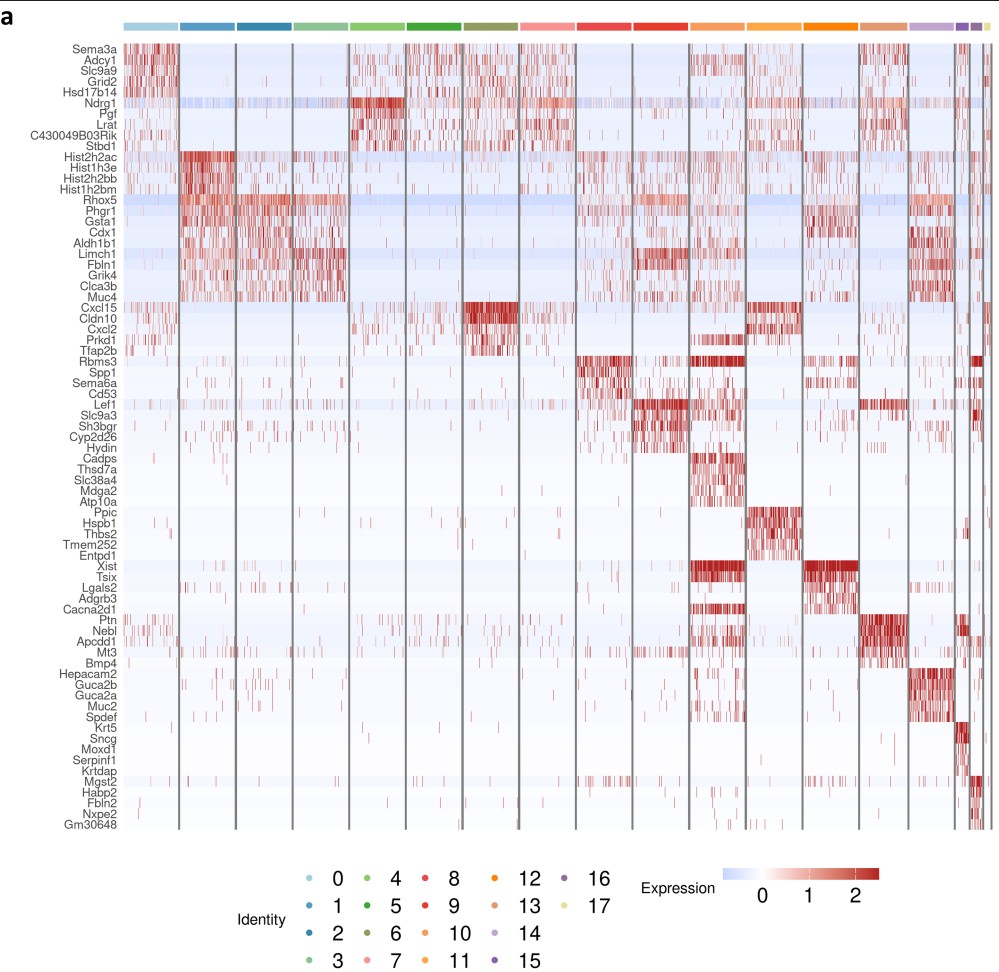

**b**

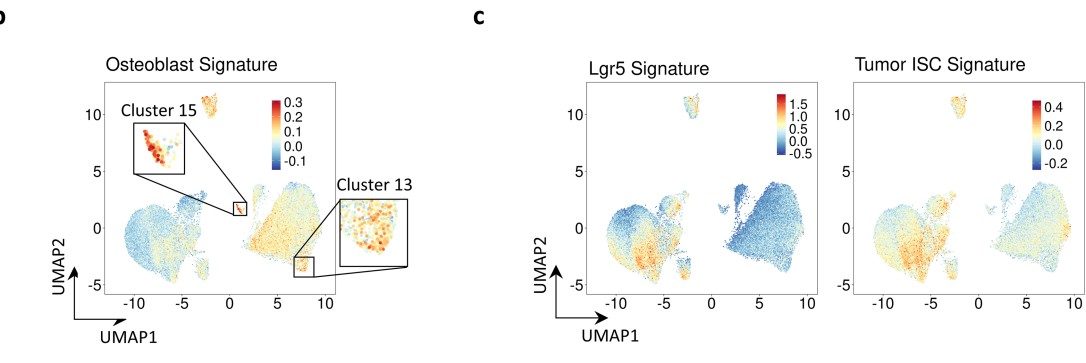

**c**

**d**

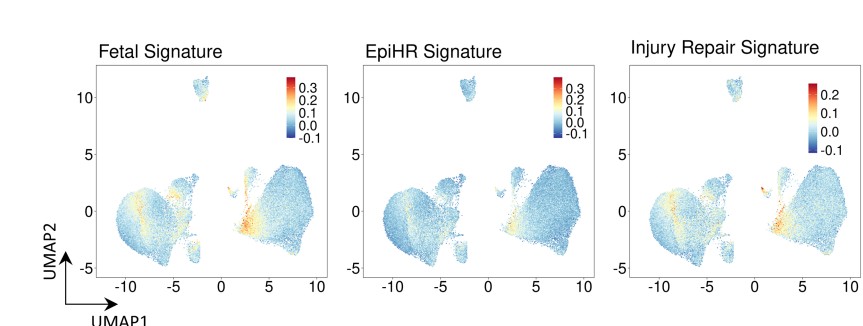

**Extended Data Fig. 5** | See next page for caption.

**Extended Data Fig. 5 | scRNAseq of *AKP* vs *AKP Atrx^{KO}* organoids.**
(a) Heatmap of single-cell RNAseq analysis showing the top 5 differentially expressed genes per cluster in *AKP Control* and *AKP Atrx^{KO}* organoids. (b) UMAP plot of *AKP Control* and *AKP Atrx^{KO}* cells coloured by expression levels of the osteoblast signature. The zoomed area of cluster 13 and 15 are outlined in black boxes. (c) UMAP plots of *AKP Control* and *AKP Atrx^{KO}* cells coloured by expression levels of the Lgr5 and tumour intestinal stem cells (ISC) signatures. (d) UMAP plots of *AKP Control* and *AKP Atrx^{KO}* cells coloured by expression levels of the Fetal, Epithelial-Specific High-Risk (EpiHR) or Injury Repair signatures.

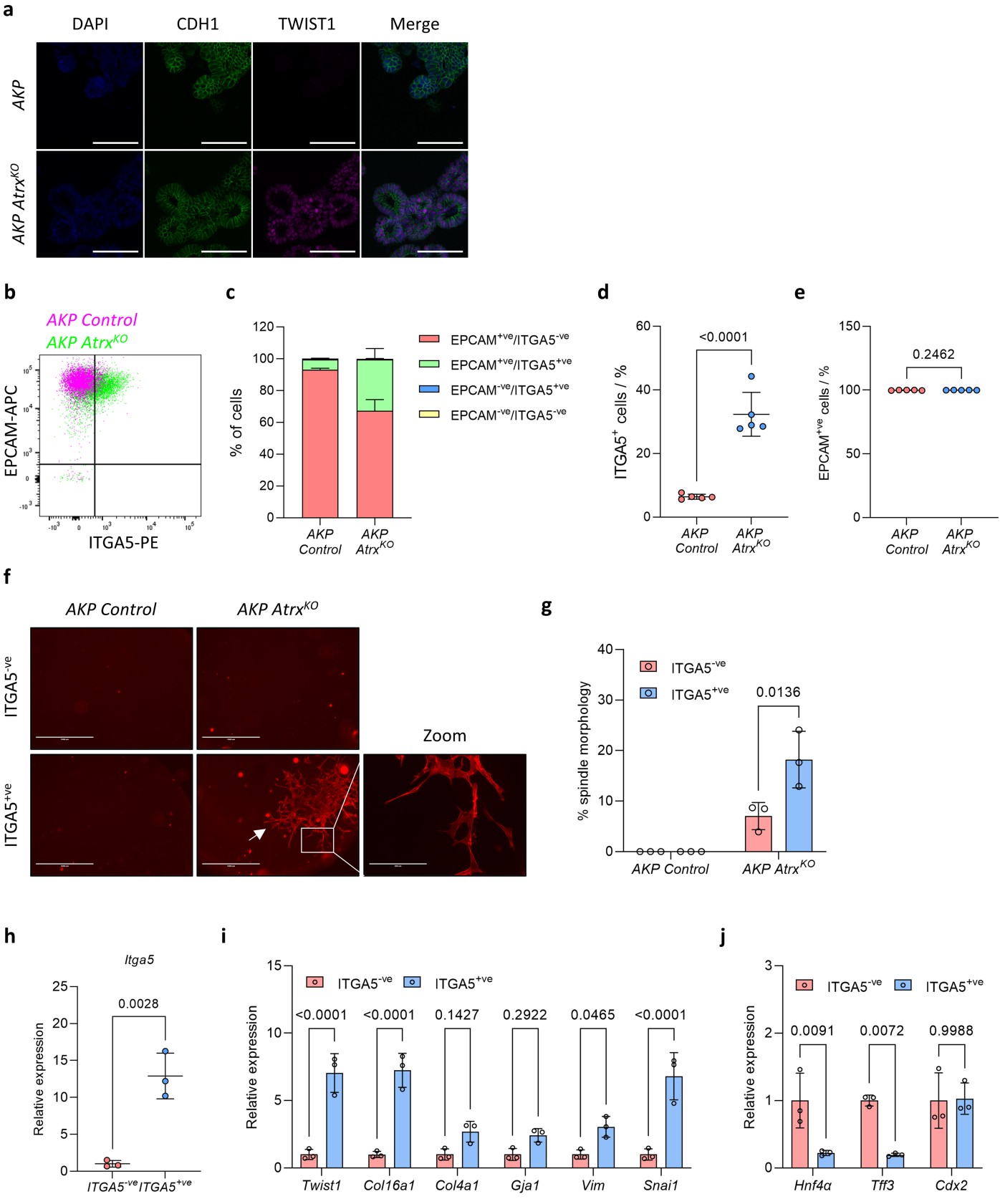

**Extended Data Fig. 6** | See next page for caption.

**Extended Data Fig. 6 | *Atrx* deletion results in hybrid-EMT phenotype.**
(a) Representative images of CDH1/TWIST1 co-immunofluorescence in *AKP Control* vs *AKP Atrx^KO* organoids. n = 1 biological sample examined over three independent experiments. Scale bars, 100 μm. (b) FACS plots for analysing and sorting EPCAM^+ve/ITGA5^+ve cells from *AKP Control* or *AKP Atrx^KO* organoids. (c) Quantification of percentage of cells in each EPCAM/ITGA5 population, n = 5 vs 5 independent experiments. (d) Quantification of percentage of ITGA5^+ve cells in *AKP Control* vs *AKP Atrx^KO* organoids, n = 5 vs 5 independent experiments. (e) Quantification of percentage of EPCAM^+ve cells in *AKP Control* vs *AKP Atrx^KO* organoids, n = 5 vs 5 independent experiments. (f) Fluorescence microscopy of phalloidin stained ITGA5^+ve and ITGA5^-ve cells sorted from *AKP Control* or *AKP Atrx^KO* organoids following treatment with 5 ng/ml TGF-beta, n = 3 vs 3 independent experiments. Spindle-like structures indicated with white arrows. Scale bars, 1000 μm. Zoomed area outlined in white box. (g) Quantification of presence of spindle-like organoid structures formed following treatment of ITGA5^+ve and ITGA5^-ve cells sorted from *AKP Control* or *AKP Atrx^KO* organoids with 5 ng/ml TGF-beta, n = 3 vs 3 independent experiments. (h) RT-qPCR analysis of *Itga5* expression in ITGA5^+ve vs ITGA5^-ve cells sorted from *AKP Atrx^KO* organoids, n = 3 vs 3 independent experiments. (i) RT-qPCR analysis of various mesenchymal marker gene expression in ITGA5^+ve vs ITGA5^-ve cells sorted from *AKP Atrx^KO* organoids, n = 3 vs 3 independent experiments. (j) RT-qPCR analysis of various colonic epithelial marker gene expression in ITGA5^+ve vs ITGA5^-ve cells sorted from *AKP Atrx^KO* organoids, n = 3 vs 3 independent experiments. For (d) and (e) data are mean ± SD. P values were calculated using two-tailed student's ttest. For (d) p = 0.000032. For (g) data are mean ± SD. P values were calculated using two-way ANOVA with Sidak's multiple comparisons test. For (h) gene expression was normalised to *Actb* and levels relative to ITGA5^-ve cells calculated using the ΔΔCt method. Data are mean ± SD. P value was calculated using two-tailed student's ttest. For (i) and (j) gene expression was normalised to *Actb* and levels relative to ITGA5^-ve cells calculated using the ΔΔCt method. Data are mean ± SD. P values were calculated using two-way ANOVA with Sidak's multiple comparisons test. For (i) *Twist1* p = 0.00000006, *Col16a1* p = 0.00000003, *Snai1* p = 0.00000012.

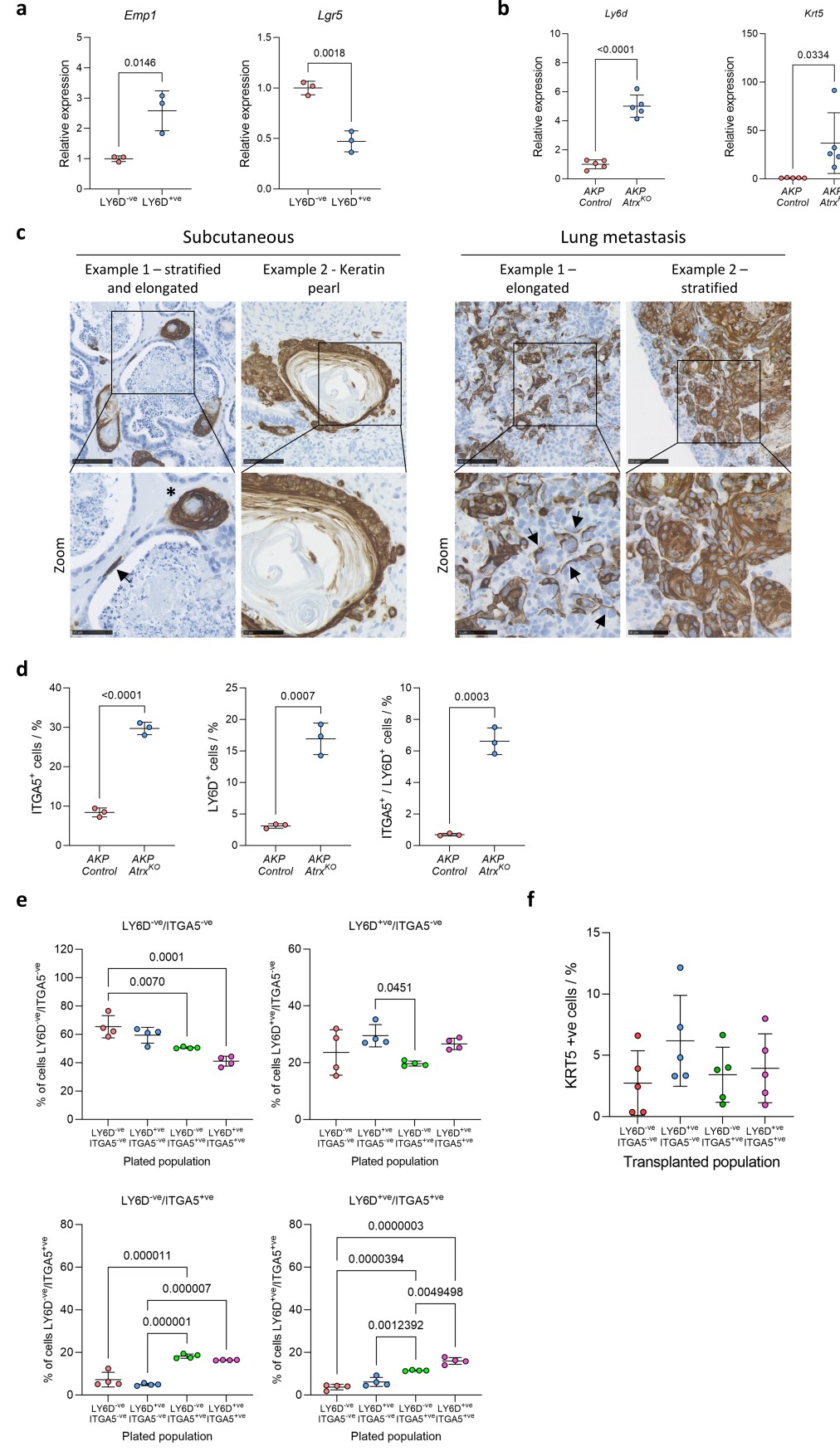

**Extended Data Fig. 7 | Emergence of squamous-like cells following *Atrx* deletion.** (a) RT-qPCR analysis of *Emp1* and *Lgr5* expression in LY6D$^{+ve}$ vs LY6D$^{-ve}$ cells sorted from *AKP Atrx$^{KO}$* organoids, n = 3 vs 3 independent experiments. (b) RT-qPCR analysis of squamous markers *Ly6d* and *Krt5* expression in tumours from mice injected with *AKP Control* or *AKP Atrx$^{KO}$* organoids, n = 5 vs 5 tumours. (c) Representative images of KRT5 staining of *AKP Atrx$^{KO}$* subcutaneous tumours (left) and lung metastases (right). Examples of the different morphologies of KRT5+ cells observed in these tumours are shown. In subcutaneous example 1, stratified epithelium (indicated by a black *) and elongated cells (indicated by black arrow) are shown. In subcutaneous tumour example 2 a keratin pearl is shown. In lung metastasis example 1 elongated cells are shown (indicated by black arrows). In lung metastasis example 2 stratified regions are shown. (d) Quantification of percentage of ITGA5$^{+ve}$ single positive, LY6D$^{+ve}$ single positive, and ITGA5$^{+ve}$/LY6D$^{+ve}$ double positive cells in *AKP Control* vs *AKP Atrx$^{KO}$* organoids, n = 3 vs 3 independent experiments. For (a) gene expression was normalised to *Actb* and levels relative to LY6D$^{-ve}$ cells calculated using the ΔΔCt method. Data are mean ± SD. P values were calculated using two-tailed student's ttest. For (b) gene expression was normalised to *Actb* and levels relative to *AKP Control* tumours calculated using the ΔΔCt method. Data are mean ± SD. P values were calculated using two-tailed student's ttest. For *Ly6d* p = 0.00005. For (d) data are mean ± SD. P values were calculated using two-tailed student's ttest. For ITGA5+ cells (left panel) p = 0.000044. (e) Quantification of percentage of ITGA5$^{-ve}$/LY6D$^{-ve}$ double negative, LY6D$^{+ve}$ single positive, ITGA5$^{+ve}$ single positive, and ITGA5$^{+ve}$/LY6D$^{+ve}$ double positive cells in cultures of *AKP Atrx$^{KO}$* organoids 9 days after plating of different sorted populations. The plated population is indicated on the x-axis and the population being analysed indicated in the graph title and y-axis, n = 4 vs 4 independent experiments. (f) Quantification of percentage of KRT5+ cells in tumours derived from mice injected with ITGA5$^{-ve}$/LY6D$^{-ve}$ double negative, LY6D$^{+ve}$ single positive, ITGA5$^{+ve}$ single positive, or ITGA5$^{+ve}$/LY6D$^{+ve}$ double positive *AKP Atrx$^{KO}$* cells, n = 5 vs 5 mice. For (e) and (f) data are mean ± SD. P values were calculated using one-way ANOVA with multiple comparisons test.

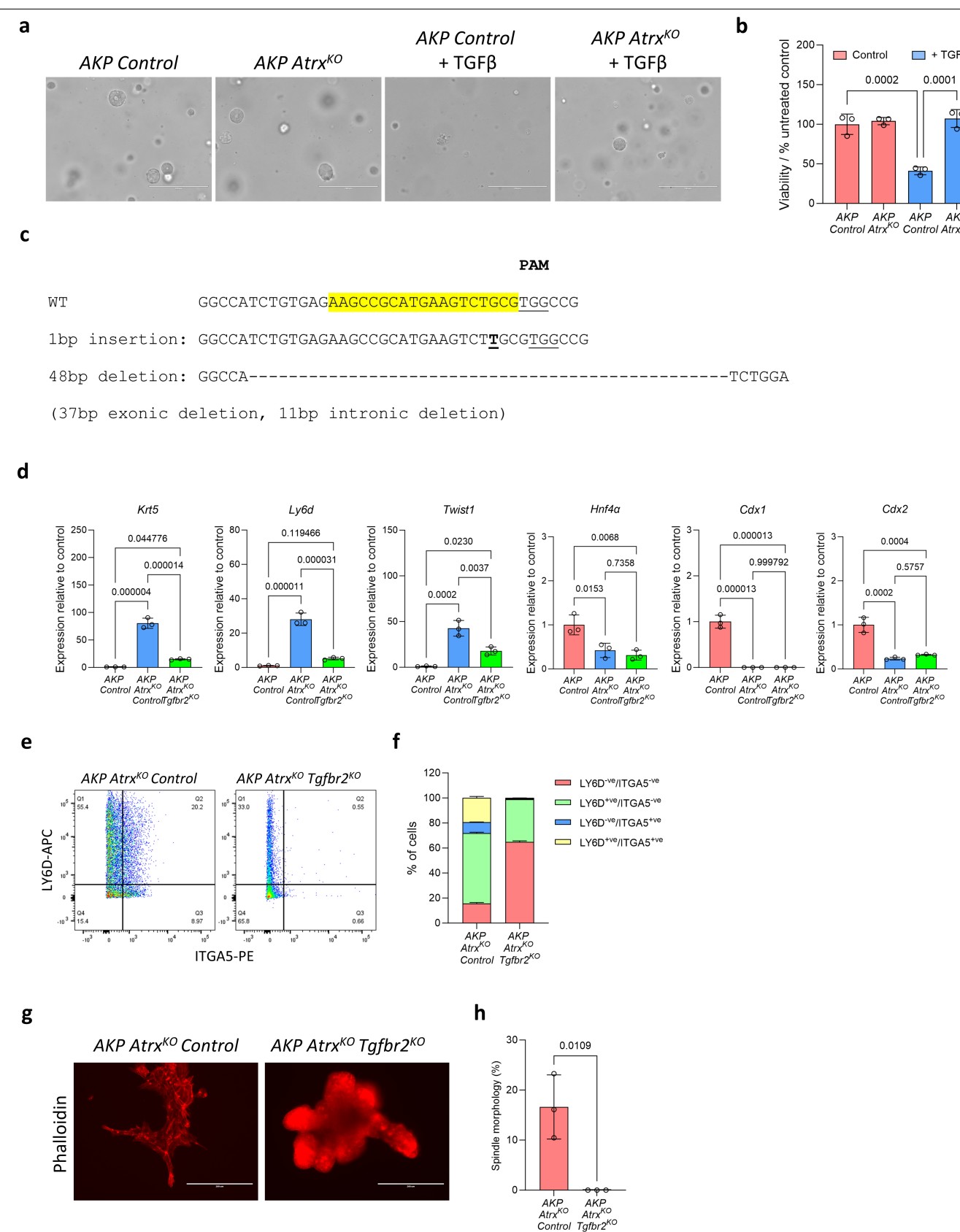

**c**

PAM

WT              GGCCATCTGTGAG<mark>AAGCCGCATGAAGTCTGCG</mark>TGGCCG

1bp insertion: GGCCATCTGTGAGAAGCCGCATGAAGTCT**T**GCGTGGCCG

48bp deletion: GGCCA--------------------------------------------TCTGGA

(37bp exonic deletion, 11bp intronic deletion)

**Extended Data Fig. 8** | See next page for caption.

**Extended Data Fig. 8 | *Tgfbr2* deletion suppresses plasticity in $Atrx^{KO}$ organoids.** (a) Representative images of *AKP Control* and *AKP Atrx*$^{KO}$ cells treated or untreated with 5 ng/ml TGF-beta (TGFβ) for 72 h, n = 3 vs 3 independent experiments. Scale bars, 200 μm. (b) Relative cell viability of *AKP Control* vs *AKP Atrx*$^{KO}$ single cells treated or untreated with TGF-beta, n = 3 vs 3 independent experiments. (c) Targeted DNA sequencing of targeted *Tgfbr2* locus in *AKP Atrx*$^{KO}$ organoid lines. (d) RT-qPCR analysis of representative genes important for squamous, EMT and colonic lineage specification and function in *AKP Control, AKP Atrx*$^{KO}$ *Control* and *AKP Atrx*$^{KO}$ *Tgfbr2*$^{KO}$ organoids. (e) FACS plots for analysing LY6D$^{+ve}$/ITGA5$^{+ve}$ cells from *AKP Atrx*$^{KO}$ *Control* and *AKP Atrx*$^{KO}$ *Tgfbr2*$^{KO}$ organoids, n = 3 vs 3 independent experiments. (f) Quantification of percentage of cells in each LY6D / ITGA5 population, n = 3 vs 3 independent experiments. (g) Fluorescence microscopy of phalloidin stained *AKP Atrx*$^{KO}$ *Control* and *AKP Atrx*$^{KO}$ *Tgfbr2*$^{KO}$ organoids following treatment with 5 ng/ml TGF-beta, n = 3 vs 3 independent experiments. Scale bars, 200 μm. (h) Quantification of presence of spindle-like organoid structures formed following treatment of *AKP Atrx*$^{KO}$ *Control* and *AKP Atrx*$^{KO}$ *Tgfbr2*$^{KO}$ organoids with 5 ng/ml TGF-beta, n = 3 vs 3 independent experiments. For (b) data are presented as mean values ± SD. P values were calculated using one-sided ANOVA multiple comparisons test. For (d) n = 1 biologically independent sample examined over 3 independent experiments. Gene expression was normalised to *Actb* and levels relative to *AKP Control* calculated using the ΔΔCt method. Data are mean ± SD. P values were calculated using one-sided ANOVA multiple comparisons test. For (h) p value was calculated using two-tailed student's t test.

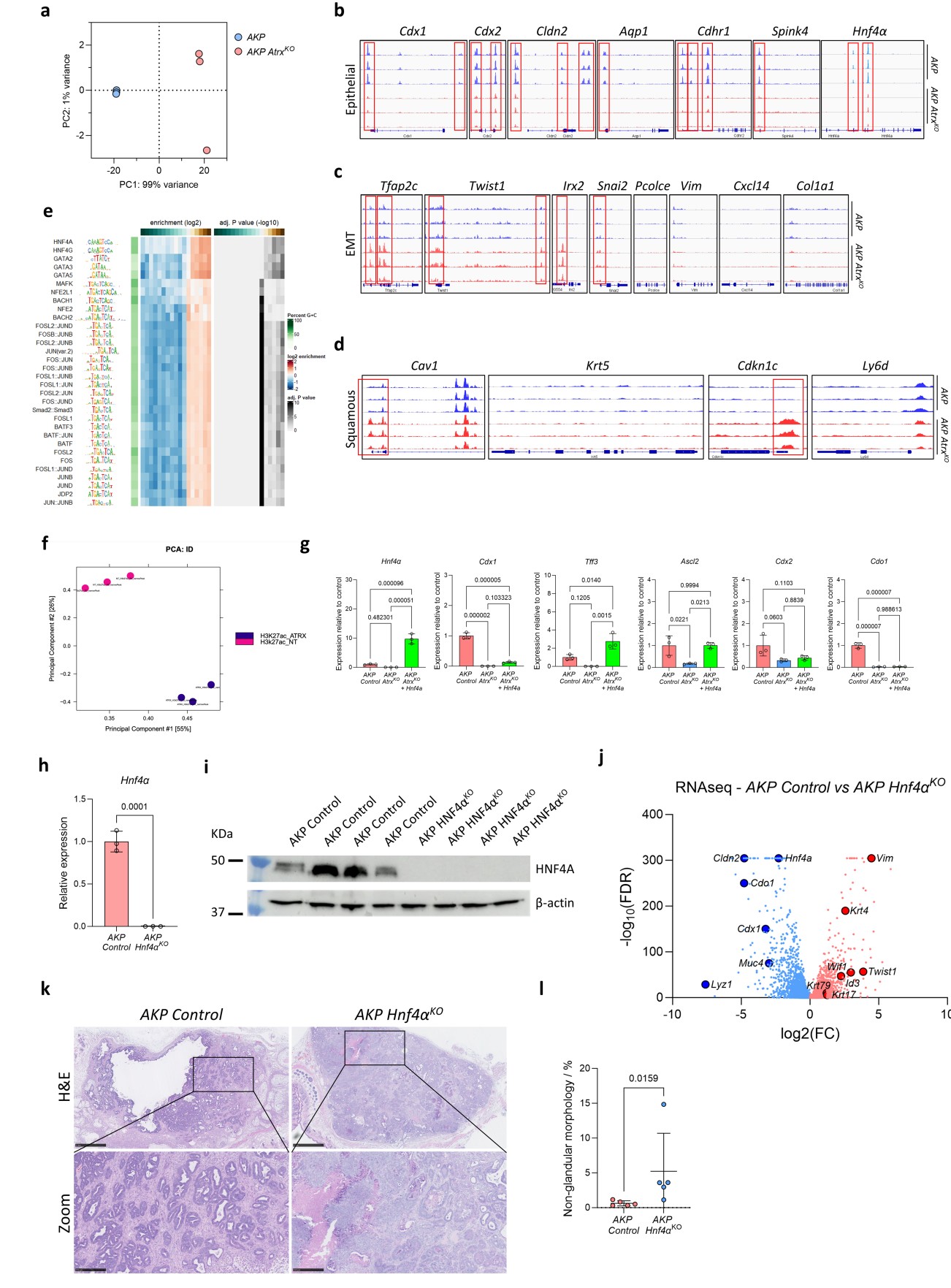

**Extended Data Fig. 9** | See next page for caption.

**Extended Data Fig. 9 | HNF4A mediates *Atrx* loss phenotypes.** a) PCA plot of ATAC-seq data comparing *AKP Control* vs *AKP Atrx^{KO}* organoids, n = 3 vs 3 independent experiments. (b) Representative IGV browser tracks of ATAC peaks at colonic epithelial genes. Significantly altered peaks are outlined with a red box. Blue track refers to *AKP control*, while red track refers to *AKP Atrx^{KO}* organoids. (c) Representative IGV browser tracks of ATAC peaks at mesenchymal genes. Significantly altered peaks are outlined with a red box. Blue track refers to *AKP control*, while red track refers to *AKP Atrx^{KO}* organoids. (d) Representative IGV browser tracks of ATAC peaks at squamous genes. Significantly altered peaks are outlined with a red box. Note lack of significantly different peaks at *Krt5* and *Ly6d* despite altered expression. Blue track refers to *AKP control*, while red track refers to *AKP Atrx^{KO}* organoids. (e) Heatmap showing transcription factor binding sites enriched in regions that lose accessibility in *AKP Atrx^{KO}* organoids. Brown shade gradient indicates regions where accessibility is lost in *AKP Atrx^{KO}* organoids. (f) PCA plot of H3K27ac CUT&RUN comparing *AKP Control* vs *AKP Atrx^{KO}* organoids, n = 3 vs 3 independent experiments. (g) RT-qPCR analysis of colonic epithelial cell marker expression in *AKP Control*, *AKP Atrx^{KO}* and *AKP Atrx^{KO} Hnf4a* overexpressing organoids, n = 3 vs 3 independent experiments. (h) RT-qPCR analysis of Hnf4α in *AKP Control* vs *AKP Hnf4a^{KO}* organoids, n = 3 vs 3 independent experiments (i) Western-blot analysis of *AKP Control* and *AKP Hnf4a^{KO}* organoids for HNF4A and β-actin. n = 4 vs 4 technical replicates. (j) Volcano plot depicting differentially expressed genes in *AKP Control* vs *AKP Hnf4a^{KO}* organoids following RNAseq analysis. Red dots represent genes expressed at higher levels in the *AKP Hnf4a^{KO}* organoids, while blue dots represent genes expressed at higher levels in the *AKP Control*, n = 3 vs 3 technical replicates. (k) Representative H&E images of subcutaneous tumours in mice subcutaneously injected with *AKP Control* or *AKP Hnf4a^{KO}* organoid cells, n = 5 vs 5 mice. Magnification highlights the distinct morphological features. Scale bars, 1 mm overview and 250 μm zoom. (l) Quantification of non-glandular morphology area of *AKP Control* vs *AKP Hnf4a^{KO}* subcutaneous tumours, n = 5 vs 5 mice. For (e) p values were calculated using one-sided Fisher's exact test. For (g) and (h) gene expression was normalised to *Actb* and levels relative to *AKP Control* calculated using the ΔΔCt method. Data are mean ± SD. For (g) p values were calculated using one-way ANOVA with multiple comparisons test. For (h) p value was calculated using two-tailed student's t test. For (l) data are mean ± SD. P values were calculated using two-sided Mann-Whitney test.

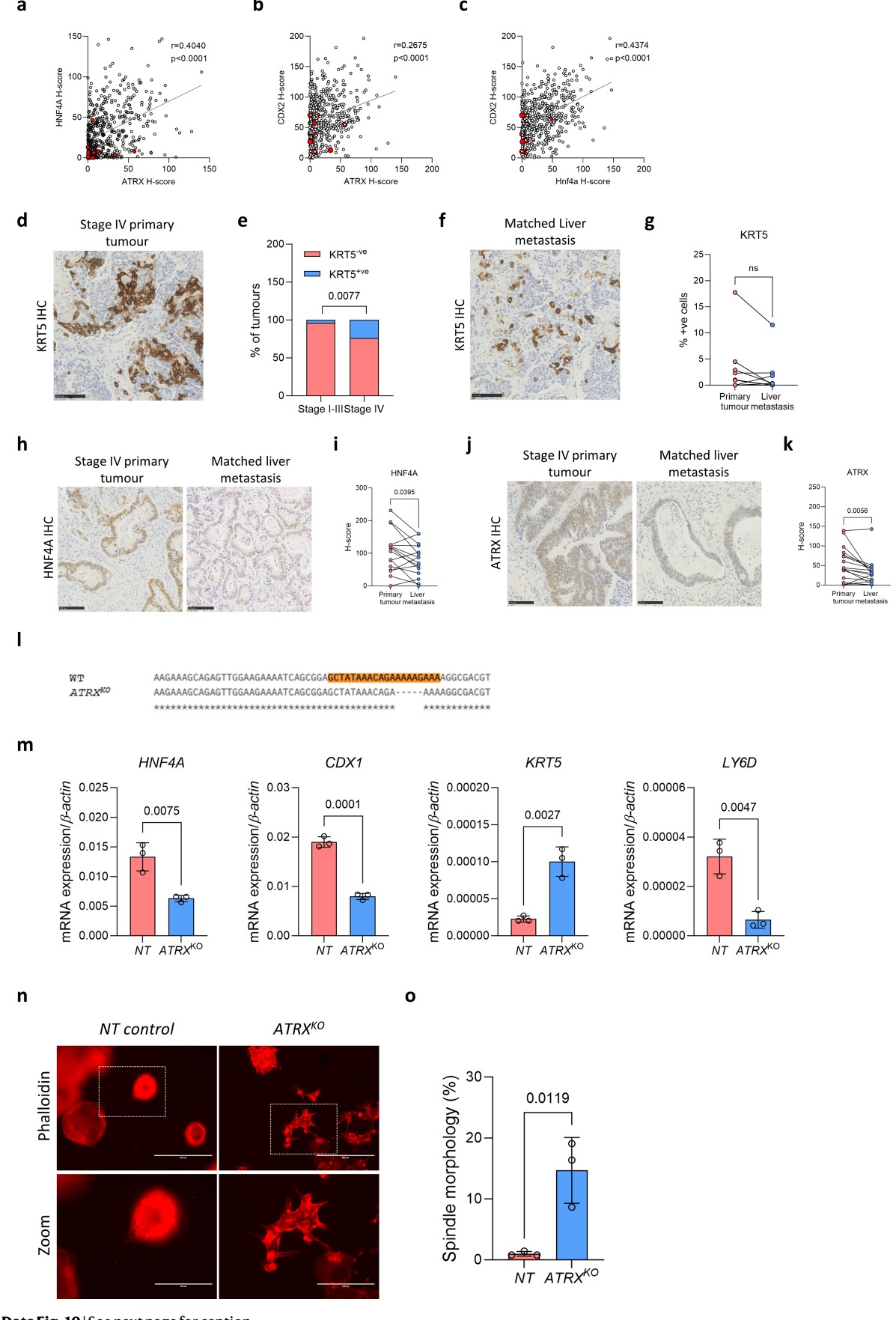

**Extended Data Fig. 10** | See next page for caption.

**Extended Data Fig. 10 | Squamous-like plasticity observed in stage IV CRC samples.** (a) Scatter plot of individual tumour H-score values for HNF4A and ATRX of stained human CRC tissue microarray. (b) Scatter plot of individual tumour H-score values for CDX2 and ATRX of stained human CRC tissue microarray. (c) Scatter plot of individual tumour H-score values for CDX2 and HNF4A of stained human CRC tissue microarray. For (a), (b) and (c) samples staining positive for LY6D ( > 2% cells LY6D[+ve]) highlighted in red. Data is individual value X/Y scatter plot. P values were calculated using two-sided Pearson's correlation test. For (a) p < 0.000000000000001, (b) p = 0.0000000009, (c) p < 0.000000000000001. (d) Representative IHC staining of a human primary stage IV CRC tissue microarray for KRT5. (e) Summary data indicating the presence of KRT5 positive cells (>2% cells KRT5[+ve]) in human stage I-III vs stage IV primary CRC. (f) Representative IHC staining of a human matched liver metastasis tissue microarray for KRT5 expression. (g) Summary data indicating the percentage of KRT5 positive cells in matched human primary tumours and liver metastases. (h) Representative image of HNF4A IHC staining in matched human stage IV primary tumour and matched liver metastasis. Scale bars, 100 μm. (i) Summary data indicating the percentage of HNF4A positive cells in matched human primary tumours and liver metastases. (j) Representative image of ATRX IHC staining in matched human stage IV primary tumour and liver metastasis. Scale bars, 100 μm. (k) Summary data indicating the percentage of ATRX positive cells in matched human primary tumours and liver metastasis. For (e) p value was calculated using two-sided Fisher's exact test. For (g), (i) and (k) p values were calculated using two-tailed Wilcoxon matched-pairs signed rank test. n = 17 vs 17 matched human primary stage IV tumours and liver metastases. (l) Targeted DNA sequencing of targeted *ATRX* locus in human CRC organoid line. (m) RT-qPCR analysis of colonic epithelial and squamous marker gene expression in human *ATRX*[KO] CRC organoids and control (NT), n = 3 vs 3 independent experiments. (n) Fluorescence microscopy of phalloidin stained human control (*NT*) and *ATRX*[KO] CRC organoids following treatment with 5 ng/ml TGF-beta, n = 3 vs 3 independent experiments. Zoomed area outlined in white box. Scale bars, 400 μm overview, 200 μm zoom. (o) Quantification of the percentage of human control (*NT*) and *ATRX*[KO] CRC organoids adopting spindle-like morphology following TGF-beta treatment. For (m) gene expression was normalised to *ACTB*. For (m) and (o) n = 1 biologically independent sample examined over 3 independent experiments. Data are mean ± SD. P values were calculated using two-tailed student's t test.

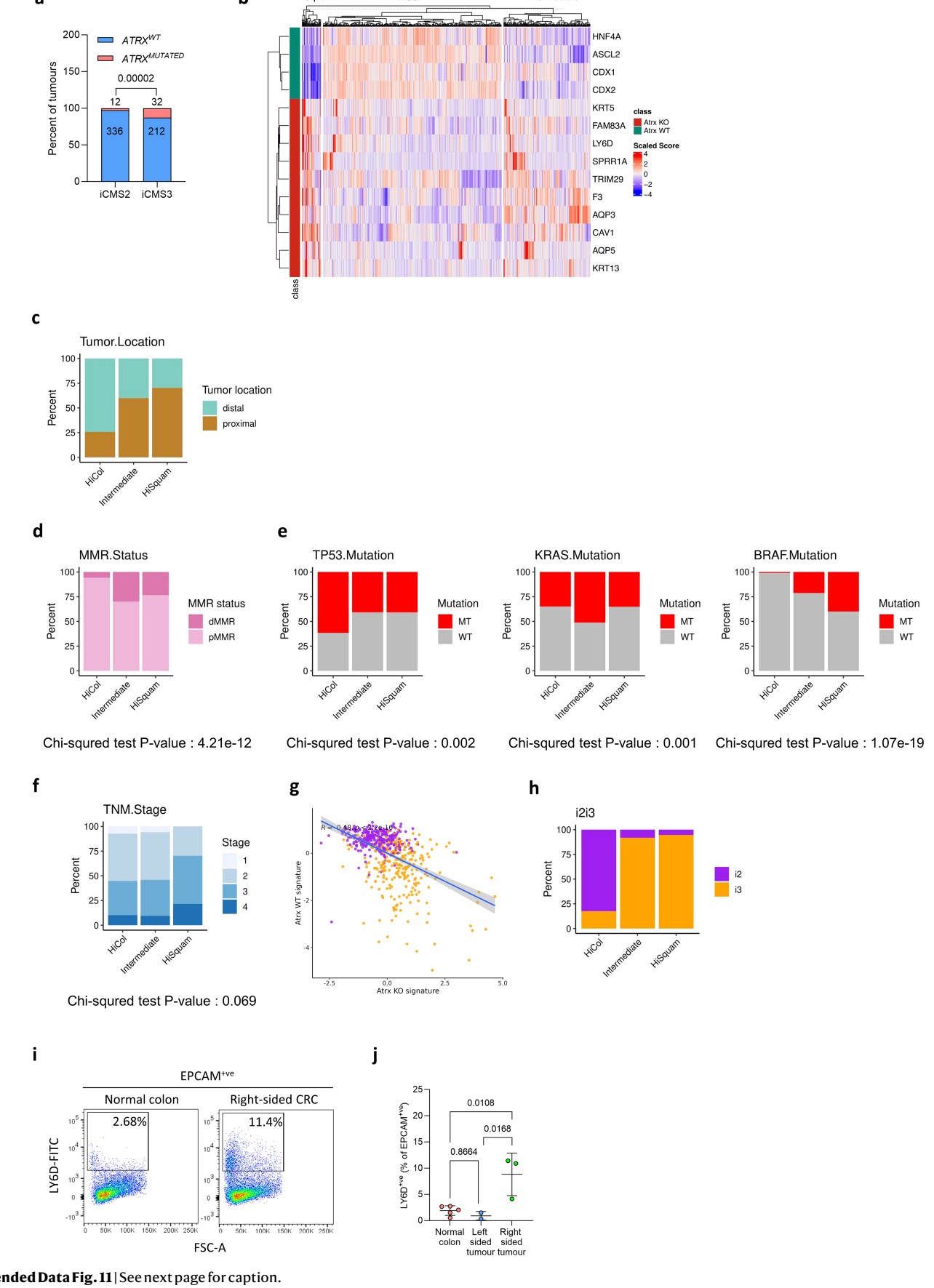

**Extended Data Fig. 11** | See next page for caption.

**Extended Data Fig. 11 | *ATRX* mutation correlates with iCMS3 transcriptional subtype.** (a) Summary data indicating presence or absence of *ATRX* mutation in iCMS2 vs iCMS3 transcriptional subtypes. Data extracted from TCGA dataset. Number of tumours in each group indicated on graph, n = 348 vs 244 tumours. (b) Heatmap clustering of Marisa CRC patient dataset of 557 tumours based on squamous-like (*Atrx*^KO^) and colonic epithelial (*Atrx*^WT^) gene expression signatures. (c) Percentage of tumours located in the distal (left sided) or proximal (right sided) colon in different *Atrx* expression clusters. One-tailed Chi-squared test, p value = 1.72e-16. (d) MMR status of tumours in HiCol, Intermediate and HiSquam expression clusters. (e) Mutational status of tumours in HiCol, Intermediate and HiSquam expression clusters. (f) TNM stage of tumours in HiCol, Intermediate and HiSquam expression clusters. (g) Correlation of *Atrx*^WT^ (colonic epithelial-like) and *Atrx*^KO^ (squamous-like) gene expression signatures in GSE39582 dataset. iCMS status overlayed in purple (iCMS2) and orange (iCMS3). (h) Percentage of tumours designated as iCMS2 (i2) or iCMS3 (i3) transcriptional subtype in different *Atrx* expression clusters. Chi-squared test, p value = 2.2e-16. (i) FACS plots for analysing EPCAM^+ve^/LY6D^+ve^ cells from normal colon and colon cancer samples. Box highlights the presence of EPCAM^+ve^/LY6D^+ve^ cells. (j) Quantification of EPCAM^+ve^/LY6D^+ve^ cells from normal colon and colon cancer samples, separated into left and right sided groups. For (a) p value was calculated using two-sided Fisher's exact test. For (c), (d), (e), (f) and (h) p values were calculated using one-tailed Chi-Square test. For (g) Pearson's correlation coefficient and two-sided p-value are described. The blue line and gray shaded area represent the regression line and 95% confidence interval. For (j) n = 5 (normal colon), n = 2 (left sided tumour), n = 3 (right sided tumour) biological independent samples. Data are mean ± SD. P values were calculated using ordinary one-way ANOVA with multiple comparisons.

**a**

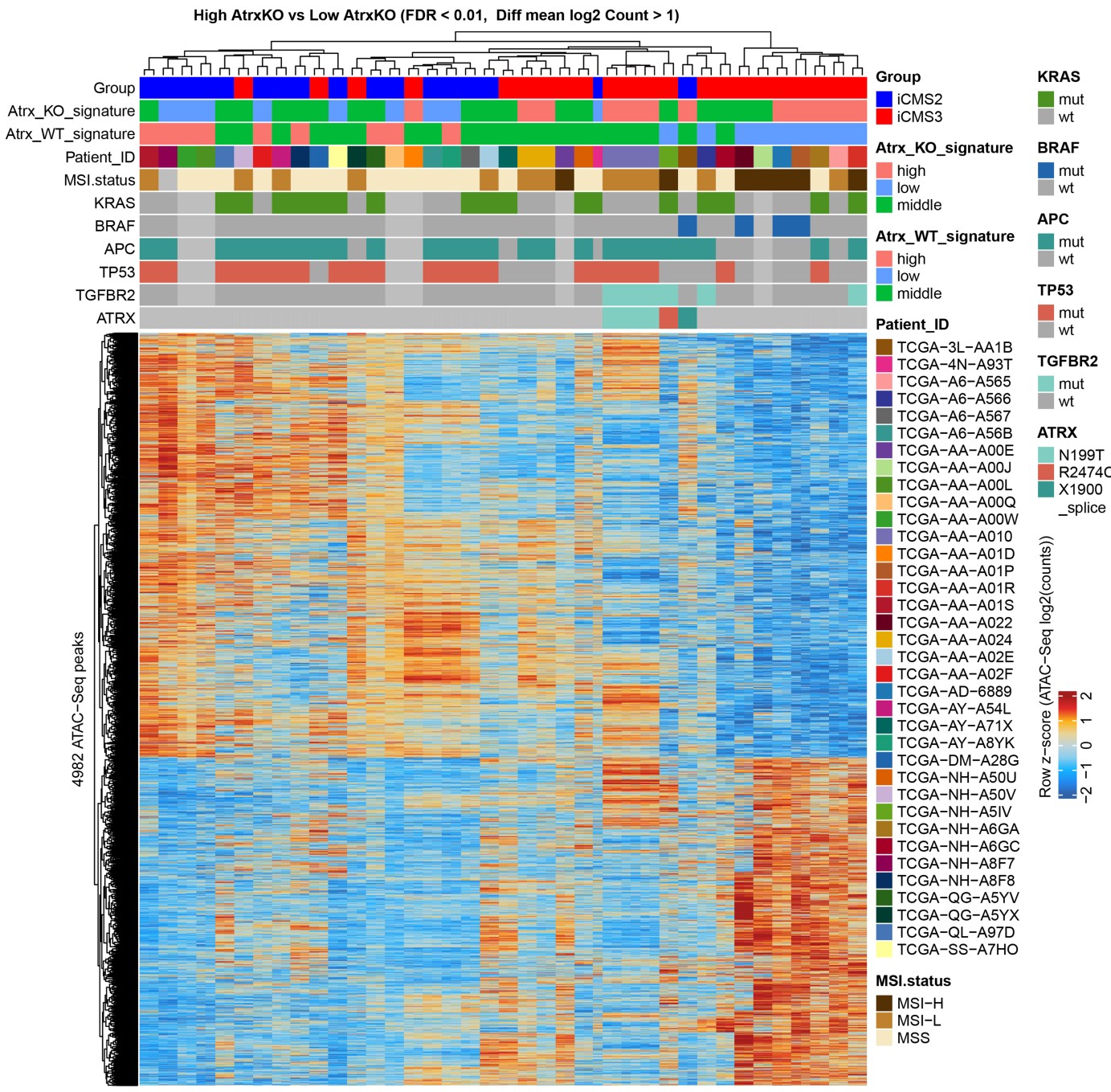

**Extended Data Fig. 12 | *Atrx^KO* expression signature is associated with loss of HNF4A activity.** Heatmap of *z*-scores of ATAC-seq log2 normalised counts across 77 samples from 37 CRC patients from the TCGA. ATAC-seq peaks shown correspond to the significantly differential accessible regions in patients with high *Atrx^KO* expression score vs high *Atrx^WT* patients. FDR was calculated using two tailed ttest.

# Reporting Summary

## Statistics

For all statistical analyses, confirm that the following items are present in the figure legend, table legend, main text, or Methods section.

| n/a | Confirmed | |
|---|---|---|
| ☐ | ☒ | The exact sample size (*n*) for each experimental group/condition, given as a discrete number and unit of measurement |
| ☐ | ☒ | A statement on whether measurements were taken from distinct samples or whether the same sample was measured repeatedly |
| ☐ | ☒ | The statistical test(s) used AND whether they are one- or two-sided *Only common tests should be described solely by name; describe more complex techniques in the Methods section.* |
| ☒ | ☐ | A description of all covariates tested |
| ☐ | ☒ | A description of any assumptions or corrections, such as tests of normality and adjustment for multiple comparisons |
| ☐ | ☒ | A full description of the statistical parameters including central tendency (e.g. means) or other basic estimates (e.g. regression coefficient) AND variation (e.g. standard deviation) or associated estimates of uncertainty (e.g. confidence intervals) |
| ☐ | ☒ | For null hypothesis testing, the test statistic (e.g. *F*, *t*, *r*) with confidence intervals, effect sizes, degrees of freedom and *P* value noted *Give P values as exact values whenever suitable.* |
| ☒ | ☐ | For Bayesian analysis, information on the choice of priors and Markov chain Monte Carlo settings |
| ☒ | ☐ | For hierarchical and complex designs, identification of the appropriate level for tests and full reporting of outcomes |
| ☐ | ☒ | Estimates of effect sizes (e.g. Cohen's *d*, Pearson's *r*), indicating how they were calculated |

*Our web collection on statistics for biologists contains articles on many of the points above.*

## Software and code

Policy information about availability of computer code

| Data collection | RNASeq, ATAC-seq, CUT&RUN, 10x scRNAseq: NovaseqS1 Illumina sequencing, NextSeq 2000 platform. qPCR: BioRad CFX Connect Histology: Hamamatsu Nanozoomer XR Flow cytometry and cell sorting (FACS): BD FACS ARIA II/BD LSR-Fortessa X-20 |
|---|---|
| Data analysis | RNAseq analysis - RaNASeq pipeline, GSEA, TissueEnrich. ATAC-seq - DESeq2, monaLisa R package, IGV browser. CUT&RUN - MACS2, Diffbind, BEDTools. Histology - NDP.view2 U12388-01, QuPath 0.2.3. Graphs and statistics - Microsoft Office Excel 365, GraphPad Prism 9.0. 10x scRNAseq - Cell Ranger 7.2.0, R package Seurat (v5), TissueEnrich. FACS - FlowJo v10.8.0. |

For manuscripts utilizing custom algorithms or software that are central to the research but not yet described in published literature, software must be made available to editors and reviewers. We strongly encourage code deposition in a community repository (e.g. GitHub). See the Nature Portfolio guidelines for submitting code & software for further information.

## Data

Policy information about availability of data

All manuscripts must include a data availability statement. This statement should provide the following information, where applicable:
- Accession codes, unique identifiers, or web links for publicly available datasets
- A description of any restrictions on data availability
- For clinical datasets or third party data, please ensure that the statement adheres to our policy

The CUT&RUN, ATAC-seq, scRNA seq and RNAseq data generated in this study are deposited in the Genome Sequence Archive (https://ngdc.cncb.ac.cn/gsa/) website. The accession numbers for the data are: CRA024850 (H3K27ac AKP vs AKP ATRX); CRA024849 (ATACseq AKP vs AKP ATRX); CRA024816 (scRNAseq AKP vs AKP ATRX; CRA024804 (RNAseq AKP vs AKP HNF4A); CRA024763 (RNAseq AKP vs AKP ATRX).

Extended data Figure 4p available from Human Protein Atlas proteinatlas.org

All data are provided within the article and Supplementary Information.

## Research involving human participants, their data, or biological material

Policy information about studies with human participants or human data. See also policy information about sex, gender (identity/presentation), and sexual orientation and race, ethnicity and racism.

| | |
|---|---|
| Reporting on sex and gender | Tissues were obtained from both males and females. |
| Reporting on race, ethnicity, or other socially relevant groupings | n/a |
| Population characteristics | This study did not involve human research participants, but utilised human derived biospecimens to generate primary organoids or for FACS analysis. Patient samples were collected from daily surgery and researchers were blinded to the identification of donors. Available patients information are reported in supplementary table and method section. |
| Recruitment | Patients were recruited as having primary colorectal tumours in order to derive tumour material to investigate mechanisms important for this disease. No biases were present that might likely impact results. |
| Ethics oversight | Ethical approval for human CRC organoid derivation was carried out under NHS Lothian Ethical Approval Scottish Colorectal Cancer Genetic Susceptibility Study 3 (SOCCS3) (REC reference: 11/SS/0109). All patients provided fully informed consent for use of their tissues. |
| | In-house TMA: patient tissue access was authorized by the NHS Greater Glasgow and Clyde Biorepository under their NHS Research Ethics Committee approval with ethical approval granted in biorepository application #845, West of Scotland Ethics 22/WS/0207 in accordance with recognized ethical guidelines as described in the Declaration of Helsinki. |
| | 17 patients undergoing synchronous resection of primary colorectal cancer and colorectal cancer liver metastases with curative intent between April 2002 and June 2010 at Glasgow Royal Infirmary (UK) were analysed. Patient tissue access was authorized by the NHS Greater Glasgow and Clyde Biorepository under their NHS Research Ethics Committee approval with ethical approval granted in biorepository application #357, West of Scotland Ethics 22/WS/0207 in accordance with recognized ethical guidelines as described in the Declaration of Helsinki. |

Note that full information on the approval of the study protocol must also be provided in the manuscript.

## Field-specific reporting

Please select the one below that is the best fit for your research. If you are not sure, read the appropriate sections before making your selection.

☒ Life sciences          ☐ Behavioural & social sciences          ☐ Ecological, evolutionary & environmental sciences

For a reference copy of the document with all sections, see nature.com/documents/nr-reporting-summary-flat.pdf

## Life sciences study design

All studies must disclose on these points even when the disclosure is negative.

| | |
|---|---|
| Sample size | Sample sizes for each experiment are outlined in the figure legends. |
| | For all animal experiments, n > 5 mice were used for each experimental cohort. Power analyses were carried out prior to experiments being carried out to determine the minimum number of animals required for each experiment. These analyses were informed by previous and / or preliminary experiments (for example Gudino et al Nat Comms, 2021). |

For organoid experiments, all are derived from n = 3 or n > 3 independent experiments unless otherwise stated. Sample sizes were not statistically predetermined and were based on the results of previous published experiments with these models (for example Gudino et al Nat Comms, 2021).

For RNAseq, ATAC-seq and CUT&RUN experiments, all are derived from n = 3 independent samples. Sample sizes were not statistically predetermined and were based on the results of previous experiments with these models (for example Gudino et al Nat Comms, 2021).

For scRNAseq experiment, all are derives from n = 2 independent samples per condition.

| | |
|---|---|
| Data exclusions | No data were excluded from analysis. |
| Replication | Experiments (mouse, RNAseq, ATAC-seq, CUT&RUN, RT-qPCR, organoids) were replicated at least 3 times using the same experimental approach or using multiple biologically independent replicates. scRNAseq : 2 technical replicates per condition. All replication attempts were successful. |
| Randomization | C57/B6J or CD1 nude mice of ages 6-12 weeks were randomly grouped for transplantation experiments. All mice received the same number of cells of different genotype / phenotype. Experimental groups were determined by genotype of injected cells (for example AKP vs AKP AtrxKO). |
| Blinding | Investigators were blinded to the genotype of tumours when monitoring for clinical signs, when carrying out histological analysis and during data collection. IHC analysis of tumour histology was carried out using QuPath software with the investigator blinded to tumour genotype. For the in vitro experiments data collection and analysis were not performed blinded to the condition of the experiments. |

# Reporting for specific materials, systems and methods

We require information from authors about some types of materials, experimental systems and methods used in many studies. Here, indicate whether each material, system or method listed is relevant to your study. If you are not sure if a list item applies to your research, read the appropriate section before selecting a response.

## Materials & experimental systems

| n/a | Involved in the study |
|---|---|
| ☐ | ☒ Antibodies |
| ☐ | ☒ Eukaryotic cell lines |
| ☒ | ☐ Palaeontology and archaeology |
| ☐ | ☒ Animals and other organisms |
| ☒ | ☐ Clinical data |
| ☒ | ☐ Dual use research of concern |
| ☒ | ☐ Plants |

## Methods

| n/a | Involved in the study |
|---|---|
| ☒ | ☐ ChIP-seq |
| ☐ | ☒ Flow cytometry |
| ☒ | ☐ MRI-based neuroimaging |

## Antibodies

| | |
|---|---|
| Antibodies used | Only commercial antibody have been used.<br><br>KRT5 (Rabbit; Abcam 52635 (EP1601Y); 1:200)<br>KRT5 (Chicken; BioLegend 905903; 1:200)<br>LY6D (rabbit; Atlas HPA024755; 1:200)<br>TWIST (mouse; Santa Cruz 81417, 1:200)<br>EpCAM (rabbit; Abcam 71916; 1:200)<br>HNF4α (rabbit; CST 3113 (C11F12), 1:500)<br>ATRX (mouse; Sigma MABE1798, 39F; 1:500)<br>β-catenin (mouse; BD 610154; 1:50)<br>E-cadherin (rabbit; CST 3195, 1:200)<br>CDX2 (mouse; Atlas AMAb 91828, 1:1000)<br>EpCAM-APC (BioLegend 118213; 1:200)<br>LY6D-PE (BioLegend 138603; 1:200)<br>LY6D-APC (Miltenyi 130-115-313; 1:50)<br>ITGA5-PE (BioLegend 103805; 1:200)<br>EpCAM-APC (BioLegend 324207; 1:50)<br>LY6D-FITC (Cusabio Biotech CSB-PA613492LC01HU; 1:50)<br>anti-rabbit-594 (Invitrogen A21207; 1:400)<br>anti-chicken-488 (Invitrogen A78948; 1:400)<br>anti-streptavidin-647 (Invitrogen S32357; 1:400)<br>anti-rabbit-488 (Abcam 150073; 1:400)<br>anti-b-actin (CST 4970, (13E5); 1:5000)<br>anti-rabbit IgG HRP-linked, (CST 7074; 1:5000)<br>anti-mouse IgG HRP-linked, (CST 7076; 1:5000)<br>anti-histone H3, acetyl K27 (rabbit; Abcam 1/250) |

| Validation | KRT5 (Rabbit; Abcam 52635; 1:200) - Validated by Protein Atlas and by positive signal in skin and negative signal in colon<br>KRT5 (Chicken; BioLegend 905903; 1:200) - Validated by positive signal in skin and negative signal in colon<br>LY6D (rabbit; Atlas HPA024755; 1:200) - Validated by Protein Atlas and by positive signal in skin and negative signal in colon<br>TWIST (mouse; Santa Cruz 81417, 1:200) - Validated by positive signal in cells known to express it<br>EpCAM (rabbit; Abcam 71916; 1:200) - Validated by positive signal in cells known to express it<br>HNF4α (rabbit; CST 3113; 1:500) - Validated by positive signal in cells known to express it<br>ATRX (mouse; Sigma MABE1798; 1:500) - Validated by positive signal in cells known to express it and no signal in knockout cells<br>β-catenin (mouse; BD 610154; 1:50) - Validated by positive signal in cells known to express it<br>E-cadherin (rabbit; CST 3195, 1:200) - Validated by positive signal in cells known to express it<br>CDX2 (mouse; Atlas AMAb 91828, 1:1000) - Validated on manufacturers website by the provider<br>EpCAM-APC (BioLegend 118213; 1:200) - Validated on manufacturers website for FACS analysis of mouse tissue<br>LY6D-PE (BioLegend 138603; 1:200) - Validated on manufacturers website for FACS analysis of mouse tissue<br>LY6D-APC (Miltenyi 130-115-313; 1:50) - Validated on manufacturers website for FACS analysis of mouse tissue<br>ITGA5-PE (BioLegend 103805; 1:200) - Validated on manufacturers website for FACS analysis of mouse tissue<br>EpCAM-APC (BioLegend 324207; 1:50) - Validated on manufacturers website for FACS analysis of human tissue<br>LY6D-FITC (Cusabio Biotech CSB-PA613492LC01HU; 1:50) - Validated on manufacturers website for FACS analysis of human tissue<br>anti-rabbit-594 (Invitrogen A21207; 1:400) - Validated on manufacturers website by the provider<br>anti-chicken-488 (Invitrogen A78948; 1:400) - Validated on manufacturers website by the provider<br>anti-streptavidin-647 (Invitrogen S32357; 1:400) - Validated on manufacturers website by the provider<br>anti-rabbit-488 (Abcam 150073; 1:400) - Validated on manufacturers website by the provider<br>anti-b-actin (CST 4970; 1:5000) - Validated on manufacturers website by the provider<br>anti-rabbit IgG HRP-linked, (CST 7074; 1:5000) - Validated on manufacturers website by the provider<br>anti-mouseIgG HRP-linked, (CST 7076; 1:5000) - Validated on manufacturers website by the provider<br>anti-histone H3, acetyl K27 (rabbit; Abcam 1/250) - Validated on manufacturers website by the provider |
|---|---|

# Eukaryotic cell lines

Policy information about cell lines and Sex and Gender in Research

| Cell line source(s) | AKP organoid line a gift from Jatin Roper and Omar Yilmaz lab. HEK293 cell line was used exclusively for the lentivirus production (see method section) and was kindly provided by Dr Juan Carlos Acosta (IGMM, Edinburgh), originally obtained from ATCC. BPN organoid line was a gift from Prof Owen Sansom. |
|---|---|
| Authentication | AKP line was authenticated by confirming mutational status of Apc (growth in absence of Wnt ligand), Kras (growth in absence of EGF) and P53 (growth in presence of nutlin). HEK293 cells were not authenticated independently. |
| Mycoplasma contamination | Cell lines and organoid cultures were routinely tested for mycoplasma contamination and found to be negative. |
| Commonly misidentified lines<br>(See ICLAC register) | Not used. |

# Animals and other research organisms

Policy information about studies involving animals; ARRIVE guidelines recommended for reporting animal research, and Sex and Gender in Research

| Laboratory animals | Mice were purchased from Charles River and maintained at the animal facilities of the University of Edinburgh or Scotland Cancer Institute and were kept in 12 h light–dark cycles and were given access to water and food ad libitum. Mice were maintained in a temperature- (20–26°C) and humidity- (30–70%) controlled environment. Mice were either C57Bl6J or CD1 nude. Female mice were used for all experiments at an age of between 6 and 12 weeks once they had reached a minimum weight of 20 g. At experiment endpoints (15mm for subcutaneous injection) mice were humanely sacrificed by cervical dislocation in line with UK Home Office regulations. |
|---|---|
| Wild animals | Not used in this study. |
| Reporting on sex | Female mice were used in this study as the AKP organoid line is derived from a female mouse thus enabling transplantation into immune competent donors. |
| Field-collected samples | Not used in this study. |
| Ethics oversight | All animal experiments were performed in accordance with a UK Home Office licences (PP9016178, PP7510272, PP3908577), and were subject to review by the animal welfare and ethics board of the University of Edinburgh and University of Glasgow. |

Note that full information on the approval of the study protocol must also be provided in the manuscript.

# Plants

| | |
|---|---|
| Seed stocks | Not used in this study. |
| Novel plant genotypes | Not used in this study. |
| Authentication | Not used in this study. |

# Flow Cytometry

## Plots

Confirm that:

☒ The axis labels state the marker and fluorochrome used (e.g. CD4-FITC).

☒ The axis scales are clearly visible. Include numbers along axes only for bottom left plot of group (a 'group' is an analysis of identical markers).

☒ All plots are contour plots with outliers or pseudocolor plots.

☒ A numerical value for number of cells or percentage (with statistics) is provided.

## Methodology

| | |
|---|---|
| Sample preparation | Pelleted organoids were resuspended in 1 ml TrypLE Express (GIBCO) and incubated at 37°C for 15 minutes. Cells were vigorously dissociated via pipetting, resuspended in 10 mL advanced DMEM/F12, passed through a 40µm cell strainer and centrifuged at 300g for 5 minutes at 4°C. Single cells were washed with 0.1% BSA in PBS and stained with the following antibodies: EpCAM-APC (BioLegend 118213; 1:200), LY6D-PE (BioLegend 138603; 1:200) or LY6D-APC (Miltenyi 130-115-313; 1:50), ITGA5-PE (BioLegend 103805; 1:200).<br>Human samples:<br>Normal colorectal mucosa and tumour were sampled from freshly resected surgical specimens from patients diagnosed with colorectal cancer. Tissues were cut into small pieces and then incubated in Advanced DMEM/F12 supplemented with 1mg/ml collagenase type IV (Sigma), 0.5mg/ml, hyaluronidase (Sigma) and 10µM Y-27632 (Tocris) at 37C with vigorous shaking until the tissue was completely disaggregated (60–90 min). The digested reaction was then filtered through a 70µm cell strainer. The filtered cells were centrifuged at 500g for 5 minutes, washed twice in Advanced DMEM/F12 and once in 0.1% BSA in PBS. Single cell suspension was then analysed by FACS using the following antibody: EpCAM-APC (BioLegend 324207; 1:50) LY6D-FITC (Cusabio Biotech CSB-PA613492LC01HU; 1:50).<br><br>Single viable cells were gated based on FSC and SSC/Single Cells (FSC-A/FSC-H)/Living Cells (DAPI negative). |
| Instrument | BD FACSARIA II/BD LSR-Fortessa X-20 |
| Software | FlowJo v10.8 |
| Cell population abundance | Organoid samples were pure epithelial and cell population abundance (different lineages) ranged from 2-50%. In primary tumour samples, epithelial cell abundance was variable but in all cases sufficient to determine the abundance of different lineages. |
| Gating strategy | We gated cells based on FSC and SSC/Single Cells (FSC-A/FSC-H)/Living Cells (DAPI negative). Then gated on fluorescent markers, for example EPCAM, LY6D, ITGA5. Positive cells were gated by comparing to samples stained with fluorophore conjugated IgG to determine the negative stained population. |

☒ Tick this box to confirm that a figure exemplifying the gating strategy is provided in the Supplementary Information.

