## [Peer Review File · Nature]

Loss of colonic fidelity enables multilineage plasticity and metastasis

Corresponding Author: Dr Kevin Myant

Version 0:

Reviewer comments:

Referee #1

(Remarks to the Author)

In this study, Myant and colleagues explore the role of ATRX in regulating tumor cell plasticity in metastatic colorectal cancers (CRCs). ATRX is a chromatin remodeling protein known to be mutated in a small subset of CRC cases. The authors demonstrate that the knockout of ATRX in experimental CRC models leads to an increased metastatic potential. They provide compelling evidence that ATRX is essential for the expression of the colonic lineage transcription factor HNF4A. Loss of ATRX downregulated HNF4A, triggering the loss of intestinal identity in ATRX mutant CRCs. This effect coincides with an exacerbated Epithelial-to-Mesenchymal Transition (EMT) and the adoption of squamous programs by tumor cells, implying that ATRX limits tumor cell plasticity. Recent studies have highlighted the significance of transcriptional plasticity, driven by non-mutational processes, in the context of metastasis and therapy resistance in CRC. This study represents a pioneering example of a driver mutation impacting tumor cell plasticity, making it intriguing, novel, and relevant. Nevertheless, several aspects warrant further exploration. In particular, some experiments are preliminary or lack the appropriate controls and data should be improved to a large extent as detailed below:

- 1) All the mechanistic data presented in the study has been obtained in the AKP model. To strengthen their findings, the authors need to validate the key observations related to increased plasticity and metastatic potential in additional models, such as the *Braf/Alk5* or the *KPN* model. Furthermore, it would be of interest to investigate whether metastases generated by these alternative mouse models exhibit ATRX silencing or rely on a different mechanism.
- 2) A weakness of the study is the use of intrasplenic or tail injection approaches to model metastasis, whereby millions of cells are inoculated. Authors should provide evidence of the role of ATRX in orthotopic transplantation models.
- 3) The characterization of the cell states within ATRX mutant CRCs, which lose the colonic program and adopt squamous and partial EMT states, appears somewhat rudimentary as it primarily relies on the expression of individual marker genes for each cell state. Authors should significantly enhance these analyses by performing single-cell profiling of ATRX wild-type and mutant metastases. They should explore the relationship between the programs triggered by ATRX deficiency and the crypt stem, HRC, L1CAM and oncofetal-like states previously described by others. scRNAseq may also help define better hybrid EMT/squamous/intestinal states in ATRX mutants.
- 4) The author's findings suggest two potential mechanisms by which ATRX loss impacts metastasis: it leads to the loss of colonic identity while simultaneously exacerbating TGF-beta signaling-driven EMT. Given that loss-of-function mutations in the TGF-beta pathway are linked to poor prognosis in CRC, it is important to determine whether the increased metastatic potential upon ATRX loss is reliant on TGF-beta signaling or if the downregulation of the colonic program alone is sufficient to confer malignancy. To this end, the authors should conduct experiments involving ATRX loss in *Smad4* or *Tgfr2* mutant models to address this question.
- 5) Only a minority of CRC patients harbor ATRX mutations. Indeed, the authors confirm that only a fraction of cases expresses high levels of a squamous signature. This finding implies that the acquisition of a plastic/squamous phenotype may only be relevant in a limited subset of CRC patients. Alternatively (or additionally), ATRX silencing in metastases could potentially occur in a larger fraction of patients through non-genetic mechanisms, which would extend the significance of the authors' observations. This could be readily analyzed by immunohistochemistry on primary tumors and metastatic lesions

from CRC patients, examining ATRX levels alongside squamous and colonic markers. This analysis would strengthen the study to a large extent.

6) Authors demonstrate that ATRX is necessary for the expression of CDX2, another colon epithelium-defining transcription factor. A previous study (Dalerba et al. NJM. 2016) showed that around 4% of all CRCs silence CDX2, a feature that is strongly associated with poor prognosis. Please check if ATRX loss in CRC patient samples coincides with CDX2 silenced tumors reported by Daleberba et al. This association would enhance the relevance of the author's findings.

7) Fig 5f includes a heatmap correlating ATACseq peaks within the patient TCGA database. But this reviewer has trouble understanding its significance possibly because the authors only dedicate one line of text to explain the analysis. Why samples do not cluster according to high or low levels of the ATRX KO signature (or at least some samples appear to express high levels of both the KO and WT signatures?) Also, why authors don't show the ATRX mutational status of these samples, which should be easy to extract from the TGCA data? Also, BRAF mutations seem to be enriched in patients with low levels of the ATRX KO signature.

Minor

8) The authors generated two different Atrx KO AKP clones (Ext. Fig 2b) but I understand that most experiments are performed with only one clone. Given the known variability between cloned lines, they should provide evidence that both Atrx KO clones produce equivalent results in vitro and in vivo. In addition, please specify the KO clone utilized in each experiment in the figure panels or legend. Proper negative control for the CRISPR knockout should also be included (i.e. non-targeting sgRNA-expressing cells obtained through the same approach than KO cells).

9) Quantification of Fig.3m should be included.

10) For improved clarity, labeling the figure panels to indicate different experiments (e.g. liver versus lung metastases in fig 1), would facilitate comprehension. Additionally, enhancing the description in the text, particularly for patient analyses, would improve readability.

Referee #2

(Remarks to the Author)

In this manuscript, Cammareri et al investigated the role of Atrx in colorectal cancer (CRC) metastasis. They found frequent loss of function mutation of ATRX in advanced CRC. Using CRISPR targeting in AKP organoids, the authors showed that Atrx loss increases the metastatic potential in lung and liver metastasis models and is associated with TGF-beta induced EMT induction. Tissue-specific gene enrichment analysis indicates that the Atrx KO organoids are highly enriched in adipose and squamous genes. The Atrx KO organoids show selective loss of colonic markers and acquisition of a highly plastic state. Transcriptomic and epigenetic analysis show loss HNF4 activity upon Atrx loss, which may explain the loss of colonic identity and the enhanced plasticity observed in the Atrx KO organoids. Interestingly, loss of colonic identity and increased squamous-like signatures are associated with poor prognosis in the iCMS3 human CRC samples that are enriched for ATRX mutations, highlighting the clinical relevance of their findings. Overall, the manuscript is written clearly and concisely, and the findings provide new insight into the mechanism of cancer plasticity. However, there are some major concerns that need to be addressed to strengthen the conclusion.

Specific comments:

The manuscript used only 1 AKP organoid line to study the role of Atrx in metastasis. Would the observed effect of Atrx loss be maintained in other genetic backgrounds of clinical relevance? For example, in AK or Apc loss only organoids?

If the Atrx KO organoids have increased plasticity, they could be sensitized to other agents besides TGF-beta. Have the authors studied the effect of other EMT inducers or epigenetic drugs, such as histone deacetylase inhibitors?

The authors showed that Atrx loss promotes a plastic state with enhanced mesenchymal/squamous cell state in response to TGF-beta. However, the TGF-beta receptor is mutated in more advanced human. How would the presented results relate to the other more aggressive AKPT organoid model? Is Atrx loss also required to drive plasticity and squamous-like signatures in this model? Or would the Atrx KO promote an alternate pathway due to the acquired plasticity? Is the observed effect strictly dependent on TGF-beta signalling?

The authors showed loss of HNF4 activity in Atrx KO organoids based on ATAC seq and H3K27ac CUT'N'RUN analysis. How does ATRX regulate HNF4 transcription? Does ATRX bind to HNF4 directly? Does ATRX1 occupy HNF4 transcription loci? This part can be strengthened with a bit more mechanistic insight on how ATRX regulates HNF4 transcription.

Some discussion in the introduction appears redundant. For example, the two phrases between lines 42-45 discussing epigenetic modifications and their effect on gene expression. These could be edited to highlight examples of epigenetic modifications and the relevance of these modifications on cellular identity.

The authors show in the extended Fig. 1 that ATRX is one of the top 20 mutated cancer genes with high functional impact. How common is it for KRAS, TP53, and ATRX mutations to co-occur in patients?

Interestingly, ATRX levels are not differentially expressed between the normal colon and the late-stage CRC, although there is a reduction between the early stage and advanced disease. Is ATRX upregulated in early-stage CRC? Including the normal tissue staining for ATRX in the extended data Fig 1b would be important to faithfully compare the expression between normal, early- and late-stage CRC.

The authors showed that subcutaneous primary Atrx KO tumours are more invasive. Is there any differences of tumour size and survival between Atrx WT and KO primary tumours?

For the calcein stained organoids shown in Fig 1i, it'd be good to include untreated organoids as control?

In the extended data Fig 2, the authors show the targeted DNA sequencing of two ATRX KO clones. In some of the main Figures, the results refer to AKP Atrx KO and some to AKP Atrx KO2. Could the authors clarify in the text if this relates to these clones and why one is used for a subset of the experiments?

The expression changes in the classical EMT markers shown in Figure 1k are extremely interesting and fall in line with a hybrid phenotype. Do they have any data on the response to other concentrations of TGF-beta?

In Fig 2d, adipose tissue is the most enriched tissue from the TissueEnrich analysis. Which genes are contributing to this highly significant enrichment score?

In Fig 4e, please include WT AKP organoids as reference to compare the rescue effect by the Hnf4a overexpression.

In line 232, the authors could include that right sided corresponds to the proximal disease in parenthesis. It is included in the figure legend for Fig 5b, but it would improve the interpretation of the text.

Referee #3

(Remarks to the Author)

In this manuscript Cammareri et al. investigated the role of the chromatin remodelling helicase Atrx in colorectal cancer. The authors showed that Atrx loss is associated with late stage, metastatic disease by analysis of a tissue microarray of CRC cases. Deletion of Atrx in Apc, KrasG12D, Trp53 mutant (AKP) organoids led to the formation of highly invasive and poorly differentiated tumors. RNAseq analysis showed that depletion of Atrx caused the suppression of colonic epithelial genes in favor of genes associated with squamous tissue identity. Further characterization revealed that Atrx mutant cancer cells displayed dual mesenchymal and EMT-like states. Next, ATAC-sequencing (ATAC-seq) and CUT&RUN assays on AKP Atrx KO organoids demonstrated that loss of Atrx caused reduced chromatin accessibility and enhancer activity of epithelial genes normally regulated by HNF4A, HNF4G and GATA2/3/5 transcription factors. Rescue experiments by overexpression of HNF4a in Atrx KO organoids partially restored expression of colonic genes, while inversely, KO of Atrx in AKP organoids led to diminished expression of a subset of colonic genes. Finally in patient samples, loss of ATRX predicted aggressive disease and poor prognosis.

This manuscript sheds important new insight into the mechanisms underlying progression of colorectal cancer. In particular the shift in epithelial to mesenchymal identity associated with Atrx loss is striking and overall convincingly documented. This phenotype alone will undoubtedly attract significant interest in the field. However, despite the novelty of these findings and in particular the observation that epithelial identity is altered in a HNF4a dependent fashion, this manuscript does not fully uncover the mechanism by which Atrx loss leads to enhanced aggressiveness of CRC cells. In addition, further evidence is required demonstrating that Atrx confers enhanced metastatic potential in different CRC models. The following points would need to be addressed prior to publication.

Specific comments:

The rescue experiments in Fig 4 demonstrate that HNF4a is an important regulator of epithelial gene expression but falls short of showing that loss of HNF4a is the mechanism by which inactivation of Atrx drives cancer progression. What are the functional consequences of HNF4a loss on AKP organoids? Do the organoids adopt morphological features of mesenchymal cells in vitro or in subcutaneous tumors as shown in Figure 1i or Fig 3? If loss of HNF4a does not recapitulate the aggressive phenotype observed upon loss of Atrx, this may indicate that Atrx performs alternative functions that promote aggressive behaviour irrespective of its impact on maintaining epithelial identity. Clear indications of the downstream genes that promote cancer metastasis in the context of Atrx loss are lacking.

Previous reports have shown that orthotopic injection of AKP organoids in the caecum, for instance, leads to formation of liver metastasis albeit at reduced frequency compared to AKP organoids harboring mutations in the Tgfb pathway. In this manuscript the authors document enhanced metastatic potential of AKP Atrx KO organoids that were injected in the tail vein or intrasplenically. It remains unclear whether loss of Atrx can promote metastatic spread from primary tumors. Thus, an orthotopic model should be developed to address this question.

In silico analysis in Fig 5 indicates that alterations in Atrx are associated with iCMS3 tumours, squamous gene expression profiles, and poor prognosis. Although these data are consistent with functional data in murine AKP organoids, the authors need to further validate these by deleting Atrx in a human CRC organoid model.

Figure 5 shows that Atrx mutations are more frequently associated with iCMS3 type tumors that often harbor mutations in Tgfb pathway and Braf mutations. Yet all functional experiments were done in the AKP model, which resembles iCMS2 tumors. What is the impact of Atrx loss in Tgfbr/Braf mutant organoids? Related to this question, the Tgf-beta pathway is associated with induction of EMT phenotypes as stated in the main text, but it is also potent growth suppressor and for this reason mutations in this pathway are required for tumor progression. What effect does Tgf-beta stimulation have on the growth of Atrx wt vs KO AKP organoids? Are mutant organoids no longer sensitive to the growth suppressive effects of Tgf-beta as would be expected for AKP organoids? This could provide a mechanism underlying their enhanced metastatic potential. A more thorough analysis of the impact of Tgf-beta on AKP Atrx KO organoids is needed.

Version 1:

Reviewer comments:

Referee #1

(Remarks to the Author)

The authors have addressed all my criticisms satisfactorily, and the manuscript has been substantially strengthened. This study presents important findings that contribute to our understanding of CRC plasticity, and in my opinion, it is now ready for publication.

Eduard Batlle

Referee #2

(Remarks to the Author)

The authors have addressed all my concerns in the revised manuscript.

Referee #3

(Remarks to the Author)

The authors have satisfactorily addressed my concerns.

I only have a minor comment regarding Extended Fig 16a, b. The graph in b apparently shows that Atrx KO organoids are resistant to the growth suppressive effects of Tgfb. But one wonders how this data was compiled. The representative images shown in a are very unclear and do not show any difference between mutants and controls. Higher magnification images should be shown.

If this issue can be addressed, I would support publication of this manuscript.

We thank the reviewers for taking the time to review our manuscript, and are pleased they recognise the importance of our work. We have carried out additional experiments and manuscript changes to address all the comments they have made. Detailed responses outlining these follow each specific comment and are written in bold. These detail which Figure and what text has been altered in the manuscript. In the main manuscript and Extended data, text altered during the revision is highlighted in yellow.

Referee #1 (Remarks to the Author):

In this study, Myant and colleagues explore the role of ATRX in regulating tumor cell plasticity in metastatic colorectal cancers (CRCs). ATRX is a chromatin remodeling protein known to be mutated in a small subset of CRC cases. The authors demonstrate that the knockout of ATRX in experimental CRC models leads to an increased metastatic potential. They provide compelling evidence that ATRX is essential for the expression of the colonic lineage transcription factor HNF4A. Loss of ATRX downregulated HNF4A, triggering the loss of intestinal identity in ATRX mutant CRCs. This effect coincides with an exacerbated Epithelial-to-Mesenchymal Transition (EMT) and the adoption of squamous programs by tumor cells, implying that ATRX limits tumor cell plasticity. Recent studies have highlighted the significance of transcriptional plasticity, driven by non-mutational processes, in the context of metastasis and therapy resistance in CRC. This study represents a pioneering example of a driver mutation impacting tumor cell plasticity, making it intriguing, novel, and relevant. Nevertheless, several aspects warrant further exploration. In particular, some experiments are preliminary or lack the appropriate controls and data should be improved to a large extent as detailed below:

We thank the reviewer for their supportive remarks and insightful comments. We address their comments below.

1) All the mechanistic data presented in the study has been obtained in the AKP model. To strengthen their findings, the authors need to validate the key observations related to increased plasticity and metastatic potential in additional models, such as the Braf/Alk5 or the KPN model. Furthermore, it would be of interest to investigate whether metastases generated by these alternative mouse models exhibit ATRX silencing or rely on a different mechanism.

To extend the findings of our study we have deleted *Atrx* in the *Braf*^{V600E} *P53*^{fl/fl} *Notch*^{ICD} (BPN) mouse colorectal cancer model. This Braf driven model is transcriptionally distinct from the AKP mode, resembling the CMS4 CRC subtype (Torang et al., 2025. PMID: 39747069). We find *Atrx* deletion in this model increases the induction of TGFβ mediated EMT (Extended Data Figure 8, page 6) and in an orthotopic transplantation model, BPN *Atrx*^{KO} tumours completely lose their glandular morphology, exhibiting a poorly differentiated phenotype (Extended Data Figure 8, page 6). *Atrx* deletion also leads to metastatic progression in ~25% of mice (Extended Data Figure 8, page 6). In addition, similar to the AKP model, *Atrx* deletion leads to loss of colonic gene expression (*Cdx1*, *Cdx2*, *Hnf4a*) and increased expression of EMT markers (*Twist1*) (Extended Data Figure 9, page 7). Therefore, *Atrx* deletion in the BPN model recapitulates the results of the AKP model, validating our key findings. We do not have data on the expression of *Atrx* in metastases in other mouse models but our human data suggest it is reduced in human liver metastases compared to primary tumours (Extended Data Figure 20a-20h, page 13). This is discussed in more detail to address reviewer point 5 below.

2) A weakness of the study is the use of intrasplenic or tail injection approaches to model metastasis, whereby millions of cells are inoculated. Authors should provide evidence of the role of ATRX in orthotopic transplantation models.

The reviewer makes a valid point regarding the metastatic models used in the original study. To address this, we carried out orthotopic transplantation of our AKP and AKP *Atrx*^{KO} organoids. This confirmed our intrasplenic and tail vein injection models with 70% of AKP *Atrx*^{KO} tumours showing metastatic progression (compared to 0% for controls) (Extended Data Figure 7, pages 5-6).

3) The characterization of the cell states within ATRX mutant CRCs, which lose the colonic program and adopt squamous and partial EMT states, appears somewhat rudimentary as it primarily relies on the expression of individual marker genes for each cell state. Authors should significantly enhance these analyses by performing single-cell profiling of ATRX wild-type and mutant metastases. They should explore the relationship between the programs triggered by ATRX deficiency and the crypt stem, HRC, L1CAM and oncofetal-like states previously described by others. scRNAseq may also help define better hybrid EMT/squamous/intestinal states in ATRX mutants.

To enhance our analyses, we carried out scRNAseq on AKP Control and AKP *Atrx*^{KO} organoids. This validated our bulk RNAseq analysis, identifying loss of colonic and gain of non-colonic cell markers (Figure 2e, pages 7-8). We also identified the induction of multiple non-canonical cell states following *Atrx* deletion, including several populations expressing markers of squamous cells and osteoblasts, alongside previously described fetal, injury repair and EpiHR-like states (Figure 2c-h and Extended Data Figures 11 and 12, pages 7-8). We also found that these non-canonical and fetal-like states were broadly distinct from those expressing *Lgr5* and crypt stem markers (Figure 2h and Extended Data Figure 12, pages 7-8). Furthermore, we found broad overlap between fetal, injury repair and EpiHR cell states in our *Atrx* deletion model which also appear to resemble an oesophageal-like cell state, suggesting a more proximal intestinal phenotype (Figure 2h and Extended Data Figure 12, pages 7-8). Together, these analyses confirm the induction of highly plastic, non-canonical cell lineages following *Atrx* deletion and provide a framework (and model system) for exploring this phenomenon in CRC.

4) The author's findings suggest two potential mechanisms by which ATRX loss impacts metastasis: it leads to the loss of colonic identity while simultaneously exacerbating TGF-beta signaling-driven EMT. Given that loss-of-function mutations in the TGF-beta pathway are linked to poor prognosis in CRC, it is important to determine whether the increased metastatic potential upon ATRX loss is reliant on TGF-beta signaling or if the downregulation of the colonic program alone is sufficient to confer malignancy. To this end, the authors should conduct experiments involving ATRX loss in *Smad4* or *Tgfbr2* mutant models to address this question.

To address this, we deleted *Tgfbr2* from our AKP *Atrx*^{KO} model. This led to reduced expression of squamous (*Krt5*, *Ly6d*) and EMT (*Twist1*, *Snai2*) marker genes and decreased the proportion of LY6D+ and ITGA5+ cells in this model (Extended Data Figure 16, pages 10-11). It also reduced the ability of *Atrx*^{KO} cells to undergo EMT (Extended Data Figure 16, pages 10-11), suggesting TGFβ signalling plays a role in mediating cell plasticity in this model.

5) Only a minority of CRC patients harbor ATRX mutations. Indeed, the authors confirm that only a fraction of cases expresses high levels of a squamous signature. This finding implies that the acquisition of a plastic/squamous phenotype may only be relevant in a limited subset of CRC

patients. Alternatively (or additionally), ATRX silencing in metastases could potentially occur in a larger fraction of patients through non-genetic mechanisms, which would extend the significance of the authors' observations. This could be readily analyzed by immunohistochemistry on primary tumors and metastatic lesions from CRC patients, examining ATRX levels alongside squamous and colonic markers. This analysis would strengthen the study to a large extent.

The reviewer is correct that a minority of patients harbour Atrx mutations and a small proportion of primary tumours express squamous-like markers. Therefore, we extended our analysis to determine whether transcriptional silencing of Atrx may also be important for mediating these phenotypes and whether squamous expression is more prevalent in late-stage disease. We stained matched primary stage IV CRC tumour and liver metastasis samples for epithelial and squamous markers. We found the proportion of tumours expressing the squamous markers LY6D and/or KRT5 was higher in stage IV tumours than earlier disease stages (3% of stage I-III vs 35% of stage IV for LY6D and 4% of stage I-III vs 24% of stage IV for KRT5), suggesting this phenomenon is associated with late-stage disease and liver metastatic progression (Figure 5c-5f and Extended Data Figure 20a-20d, page 13). Interestingly, we did not see increased proportion of LY6D or KRT5 expressing cells in liver metastases compared to matched primary tumour but the expression of both HNF4A and ATRX was decreased (Figure 5c-5f and Extended Data Figure 20a-20h, page 13). Together, this suggests induction of squamous lineages precedes metastatic dissemination and that progressive loss of colonic identity (via loss of ATRX and HNF4A expression) increases metastatic potential.

6) Authors demonstrate that ATRX is necessary for the expression of CDX2, another colon epithelium-defining transcription factor. A previous study (Dalerba et al. NJM. 2016) showed that around 4% of all CRCs silence CDX2, a feature that is strongly associated with poor prognosis. Please check if ATRX loss in CRC patient samples coincides with CDX2 silenced tumors reported by Daleberba et al. This association would enhance the relevance of the author's findings.

We stained our TMA for CDX2 and compared to expression of ATRX, HNF4A and LY6D. We found that CDX2 expression positively correlates with both HNF4A and ATRX and reduced CDX2 expression is associated with emergence of LY6D+ cells (Figure 5a-5b and Extended Data Figure 19a-19c, page 13). However, this was not as binary as 'loss of CDX2 = emergence of LY6D' suggesting a more progressive emergence of non-canonical plasticity (Figure 5a-5b, page 13).

7) Fig 5f includes a heatmap correlating ATACseq peaks within the patient TCGA database. But this reviewer has trouble understanding its significance possibly because the authors only dedicate one line of text to explain the analysis. Why samples do not cluster according to high or low levels of the ATRX KO signature (or at least some samples appear to express high levels of both the KO and WT signatures?) Also, why authors don't show the ATRX mutational status of these samples, which should be easy to extract from the TGCA data? Also, BRAF mutations seem to be enriched in patients with low levels of the ATRX KO signature.

We apologise for the lack of clarity in this section. The figure contains expression of both the KO and WT signatures and as an oversight, we used different colours to indicate high, middle and low expression of each. We agree this is confusing to the reader as it can appear that some samples have both high or low levels of the different signatures. We have updated the Figure to help clarify the different signatures (Extended Data Figure 23). We have also added additional text explaining this analysis (page 15) and have included ATRX mutational status of these samples. Regarding the BRAF mutational data, of the 4 tumours with BRAF mutation, 3 have high expression of the *Atrx*^{KO}

signature, and one has medium expression, consistent with the analysis in Extended Data Figure 21. Again, the confusion here is likely due to the confusing annotation of this figure which we have addressed in this revision.

Minor

8) The authors generated two different *Atrx* KO AKP clones (Ext. Fig 2b) but I understand that most experiments are performed with only one clone. Given the known variability between cloned lines, they should provide evidence that both *Atrx* KO clones produce equivalent results in vitro and in vivo. In addition, please specify the KO clone utilized in each experiment in the figure panels or legend. Proper negative control for the CRISPR knockout should also be included (i.e. non-targeting sgRNA-expressing cells obtained through the same approach than KO cells).

The reviewer is correct that we generated multiple *Atrx*^{KO} clones for this study. We find that both lines have increased capacity for TGFβ induced EMT and enhanced metastatic progression (Extended Figure 2c-2f and Extended Figure 4, pages 4-5). We also clarify the nomenclature throughout the paper, using AKP *Atrx*^{KO} for all experiments utilising the first clone and AKP *Atrx*^{KO2} for experiments utilising the second clone (pages 4-5). In addition, for all experiments carried out with AKP, BPN and human organoids, appropriate negative controls (non-targeting sgRNA-expressing organoids) are used. Details of this are provided on (page 4 and in Methods section).

9) Quantification of Fig.3m should be included.

Quantification of Fig 3m is included as Extended Data Fig 15b.

10) For improved clarity, labeling the figure panels to indicate different experiments (e.g. liver versus lung metastases in fig 1), would facilitate comprehension. Additionally, enhancing the description in the text, particularly for patient analyses, would improve readability.

We have updated the figure panels to outline the different experiments (Figure 1a-1h). The text describing patient analysis has been extended (pages 13, 14 and 15).

Referee #2 (Remarks to the Author):

In this manuscript, Cammareri et al investigated the role of Atrx in colorectal cancer (CRC) metastasis. They found frequent loss of function mutation of ATRX in advanced CRC. Using CRISPR targeting in AKP organoids, the authors showed that Atrx loss increases the metastatic potential in lung and liver metastasis models and is associated with TGF-beta induced EMT induction. Tissue-specific gene enrichment analysis indicates that the Atrx KO organoids are highly enriched in adipose and squamous genes. The Atrx KO organoids show selective loss of colonic markers and acquisition of a highly plastic state. Transcriptomic and epigenetic analysis show loss HNF4 activity upon Atrx loss, which may explain the loss of colonic identity and the enhanced plasticity observed in the Atrx KO organoids. Interestingly, loss of colonic identity and increased squamous-like signatures are associated with poor prognosis in the iCMS3 human CRC samples that are enriched for ATRX mutations, highlighting the clinical relevance of their findings. Overall, the manuscript is written clearly and concisely, and the findings provide new insight into the mechanism of cancer plasticity. However, there are some major concerns that need to be addressed to strengthen the conclusion.

We thank the reviewer for taking their time to review our manuscript and their constructive comments. We outline our response below.

Specific comments:

The manuscript used only 1 AKP organoid line to study the role of Atrx in metastasis. Would the observed effect of Atrx loss be maintained in other genetic backgrounds of clinical relevance? For example, in AK or Apc loss only organoids?

To extend the findings of our study we have deleted *Atrx* in the *Braf*^{V600E} *P53*^{fl/fl} *Notch*^{ICD} (BPN) mouse colorectal cancer model. This clinically relevant Braf driven model is transcriptionally distinct from the AKP mode, resembling the CMS4 CRC subtype (Torang et al., 2025. PMID: 39747069). We find *Atrx* deletion in this model increases the induction of TGFβ mediated EMT (Extended Data Figure 8, page 6) and in an orthotopic transplantation model, BPN *Atrx*^{KO} tumours completely loss their glandular morphology, exhibiting a poorly differentiated phenotype (Extended Data Figure 8, page 6). *Atrx* deletion also leads to metastatic progression in ~25% of mice (Extended Data Figure 8, page 6). In addition, similar to the AKP model, *Atrx* deletion leads to loss of colonic gene expression (*Cdx1*, *Cdx2*, *Hnf4a*) and increased expression of EMT markers (*Twist1*) (Extended Data Figure 9, page 7). Therefore, *Atrx* deletion in the BPN model recapitulates the results of the AKP model, validating our key findings in an additional, clinically relevant CRC mouse model.

If the Atrx KO organoids have increased plasticity, they could be sensitized to other agents besides TGF-beta. Have the authors studied the effect of other EMT inducers or epigenetic drugs, such as histone deacetylase inhibitors?

We tested this by treating our *Atrx*^{KO} organoids with other EMT inducers (TNFα and IFNγ) and epigenetic drugs including the HDAC inhibitor FKK228 and the BET inhibitor JQ1. We find that both AKP control and AKP *Atrx*^{KO} organoids are sensitive to the effects of FKK228 and JQ1, showing a significant loss of viability rather than specific changes in EMT-like phenotypes (Extended Figure 6b-6c, page 5). Neither line shows any viability changes or EMT induction following TNFα treatment and treatment with IFNγ induces a modest reduction in viability of both lines but no

EMT-like phenotypes (Extended Data Figure 6a, page 5). Together, this suggests *Atrx* loss sensitises primarily to TGF β induced EMT induction.

The authors showed that *Atrx* loss promotes a plastic state with enhanced mesenchymal/squamous cell state in response to TGF-beta. However, the TGF-beta receptor is mutated in more advanced human. How would the presented results relate to the other more aggressive AKPT organoid model? Is *Atrx* loss also required to drive plasticity and squamous-like signatures in this model? Or would the *Atrx* KO promote an alternate pathway due to the acquired plasticity? Is the observed effect strictly dependent on TGF-beta signalling?

To address this, we deleted *Tgfbr2* from our AKP *Atrx*^{KO} model. This led to reduced expression of squamous (*Krt5*, *Ly6d*) and EMT (*Twist1*, *Snai2*) marker genes and decreased the proportion of LY6D+ and ITGA5+ cells in this model (Extended Data Figure 16, pages 10-11). It also reduced the ability of *Atrx*^{KO} cells to undergo EMT (Extended Data Figure 16, pages 10-11), suggesting TGF β signalling plays a role in mediating cell plasticity in this model.

The authors showed loss of HNF4 activity in *Atrx* KO organoids based on ATAC seq and H3K27ac CUT'N'RUN analysis. How does ATRX regulate HNF4 transcription? Does ATRX bind to HNF4 directly? Does ATRX1 occupy HNF4 transcription loci? This part can be strengthened with a bit more mechanistic insight on how ATRX regulates HNF4 transcription.

To address this, we attempted to assess ATRX chromatin binding using CUT&RUN followed by sequencing. Despite testing three different ATRX antibodies, we were unable to detect specific ATRX peaks, suggesting available antibodies are not be suitable to determine ATRX chromatin binding in our model system. As a result, we could not determine whether ATRX binds directly to HNF4A target loci.

To further investigate the relationship between ATRX and HNF4A, we generated *Hnf4a* knockout organoids. RNAseq analysis of the *Hnf4a* deleted organoids found a broad downregulation of colonic epithelial genes and a strong overlap with genes downregulated following *Atrx* deletion (Extended Data Figure 18b-18d and Figure 4e, page 12). Indeed, when investigating genes strongly silenced following *Atrx* deletion ($\log_2FC > -2$) around 20% of them showed the same loss of expression following *Hnf4a* deletion (Figure 4e, page 12). Additionally, functional experiments with *Hnf4a*^{KO} organoids showed the same phenotypes of increased TGF β induced EMT and aggressive tumour behaviour as seen with *Atrx*^{KO} (Figure 4f-4h and Extended Data Figure 18e-18f, pages 12-13).

Therefore, although direct chromatin binding could not be assessed, given ATRX's known roles in chromatin remodelling and heterochromatin regulation, our findings are consistent with a model where loss of *Atrx* alters chromatin accessibility and enhancer activity at HNF4A regulated loci, leading to reduced HNF4A function.

Some discussion in the introduction appears redundant. For example, the two phrases between lines 42-45 discussing epigenetic modifications and their effect on gene expression. These could be edited to highlight examples of epigenetic modifications and the relevance of these modifications on cellular identity.

We have altered this part of the introduction expanding on the discussion of epigenetic modification in the context of cellular identity (page 3).

The authors show in the extended Fig. 1 that ATRX is one of the top 20 mutated cancer genes with high functional impact. How common is it for KRAS, TP53, and ATRX mutations to co-occur in patients?

To determine this, we have analysed mutational data from TCGA (CRC pancancer - cbioportal) finding that in this dataset, ATRX is mutated in 36/526 cases (6.8%). ATRX mutation co-occurs with APC mutation in 27 tumours, with KRAS mutation in 14 tumours, with P53 mutation in 17 tumours and with both KRAS and P53 mutation in 8 tumours. Therefore, whilst ATRX mutation is relatively uncommon in CRC, when found, it commonly co-occurs with APC, KRAS and/or P53 mutation.

Interestingly, ATRX levels are not differentially expressed between the normal colon and the late-stage CRC, although there is a reduction between the early stage and advanced disease. Is ATRX upregulated in early-stage CRC? Including the normal tissue staining for ATRX in the extended data Fig 1b would be important to faithfully compare the expression between normal, early- and late-stage CRC.

We do not observe changes in ATRX levels between normal tissue and early-stage disease, rather ATRX expression is specifically reduced in later stage disease compared to early stages. The quantification of this is shown in Extended Data Figure 1c. To enable an accurate comparison, we have included staining of normal tissue in Extended Data Figure 1b.

The authors showed that subcutaneous primary *Atrx* KO tumours are more invasive. Is there any differences of tumour size and survival between *Atrx* WT and KO primary tumours?

There was no difference in size between sampled *Atrx*^{WT} and *Atrx*^{KO} primary subcut transplanted tumours but the *Atrx*^{KO} tumours grew slower than controls. We have included these data as Extended Data Figure 3a-3b, page 5.

For the calcein stained organoids shown in Fig 1i, it'd be good to include untreated organoids as control?

We have included images of untreated control organoids alongside the TGF-beta dose response data (Extended Data Figure 5a).

In the extended data Fig 2, the authors show the targeted DNA sequencing of two ATRX KO clones. In some of the main Figures, the results refer to AKP *Atrx* KO and some to AKP *Atrx* KO2. Could the authors clarify in the text if this relates to these clones and why one is used for a subset of the experiments?

The reviewer is correct that we generated multiple *Atrx*^{KO} clones for this study. We have updated the text to clarify this (pages 4 and 5) and updated the Figure legends to indicate which clone has been used for which experiment. Both generated lines have increased capacity for TGFβ induced EMT and enhanced metastatic progression (Extended Data Figure 2c-2f and 4a-4b).

The expression changes in the classical EMT markers shown in Figure 1k are extremely interesting and fall in line with a hybrid phenotype. Do they have any data on the response to other concentrations of TGF-beta?

To address this, we carried out a dose response to TGF β and analysed EMT induction (Extended Data Figure 5a-5b). Interestingly, *Atrx*^{KO} organoids responded the same to concentrations of TGF β as low as 1ng/ml suggesting a high sensitivity to EMT induction.

In Fig 2d, adipose tissue is the most enriched tissue from the TissueEnrich analysis. Which genes are contributing to this highly significant enrichment score?

This is an interesting point and the reviewer is correct that adipose tissue is the most highly enriched in the TissueEnrich analysis. The enriched genes include those involved in lipid catabolism, storage and localisation such as *Plin5*, *Acacb*, *Cidea* and *Dgat2* suggesting metabolic reprogramming towards lipid metabolism. We have included the adipose gene list (and those of the other enriched tissues – Table S3) and outlined this point in the manuscript (page 7).

In Fig 4e, please include WT AKP organoids as reference to compare the rescue effect by the *Hnf4a* overexpression.

We have included WT AKP organoids as a reference (Extended Data Figure 18a) showing that for some genes, *Hnf4a* overexpression is sufficient to fully rescue gene expression to control levels, but for others the rescue is either absent, or incomplete. This is discussed in the text (page 12).

In line 232, the authors could include that right sided corresponds to the proximal disease in parenthesis. It is included in the figure legend for Fig 5b, but it would improve the interpretation of the text.

We have included this description of right sidedness in the text (page 14).

Referee #3 (Remarks to the Author):

In this manuscript Cammareri et al. investigated the role of the chromatin remodelling helicase Atrx in colorectal cancer. The authors showed that Atrx loss is associated with late stage, metastatic disease by analysis of a tissue microarray of CRC cases. Deletion of Atrx in Apc, KrasG12D, Trp53 mutant (AKP) organoids led to the formation of highly invasive and poorly differentiated tumors. RNAseq analysis showed that depletion of Atrx caused the suppression of colonic epithelial genes in favor of genes associated with squamous tissue identity. Further characterization revealed that Atrx mutant cancer cells displayed dual mesenchymal and EMT-like states. Next, ATAC-seq and CUT&RUN assays on AKP Atrx KO organoids demonstrated that loss of Atrx caused reduced chromatin accessibility and enhancer activity of epithelial genes normally regulated by HNF4A, HNF4G and GATA2/3/5 transcription factors. Rescue experiments by overexpression of HNF4a in Atrx KO organoids partially restored expression of colonic genes, while inversely, KO of Atrx in AKP organoids led to diminished expression of a subset of colonic genes. Finally in patient samples, loss of ATRX predicted aggressive disease and poor prognosis.

This manuscript sheds important new insight into the mechanisms underlying progression of colorectal cancer. In particular the shift in epithelial to mesenchymal identity associated with Atrx loss is striking and overall convincingly documented. This phenotype alone will undoubtedly attract significant interest in the field. However, despite the novelty of these findings and in particular the observation that epithelial identity is altered in a HNF4a dependent fashion, this manuscript does not fully uncover the mechanism by which Atrx loss leads to enhanced aggressiveness of CRC cells. In addition, further evidence is required demonstrating that Atrx confers enhanced metastatic potential in different CRC models. The following points would need to be addressed prior to publication.

We thank the reviewer for their supportive comments and helpful suggestions. We address the specific points below.

Specific comments:

The rescue experiments in Fig 4 demonstrate that HNF4a is an important regulator of epithelial gene expression but falls short of showing that loss of HNF4a is the mechanism by which inactivation of Atrx drives cancer progression. What are the functional consequences of HNF4a loss on AKP organoids? Do the organoids adopt morphological features of mesenchymal cells in vitro or in subcutaneous tumors as shown in Figure 1i or Fig 3? If loss of HNF4a does not recapitulate the aggressive phenotype observed upon loss of Atrx, this may indicate that Atrx performs alternative functions that promote aggressive behaviour irrespective of its impact on maintaining epithelial identity. Clear indications of the downstream genes that promote cancer metastasis in the context of Atrx loss are lacking.

This is an important question and to address it we determined the phenotypic effects of *Hnf4a* deletion. We found that similar to *Atrx*, deletion of *Hnf4a* sensitised AKP organoids to TGFβ induced EMT (Figure 4f-4g, page 12). RNAseq analysis showed this coincided with a loss of expression of colonic epithelial markers (*Cdx1*, *Cdo1m*, *Muc4*, *Lyz1* and others) and a gain of expression of EMT and squamous markers (*Twist1*, *Krt4*, *Krt17*) (Extended Data Figure 18d, pages 12-13). Additionally, comparison to the *Atrx*^{KO} RNAseq dataset found a strong overlap with genes

downregulated following *Atrx* deletion (Figure 4e and Extended Data Figure 17d, page 12). Indeed, when investigating genes strongly silenced following *Atrx* deletion ($\log_2FC > -2$) around 20% of them showed the same loss of expression following *Hnf4a* deletion (Figure 4e, page 12). Additionally, subcutaneous transplant of these tumours demonstrated *Hnf4a* deletion led to the same, loss of glandular morphology, phenotype as seen with *Atrx* deletion (Figure 4h and Extended Data Figure 18e-18f, page 12). Together, these data strongly support loss of *Hnf4a* expression and colonic gene expression as a key mediator of the aggressive tumour behaviour observed following *Atrx* deletion.

Previous reports have shown that orthotopic injection of AKP organoids in the caecum, for instance, leads to formation of liver metastasis albeit at reduced frequency compared to AKP organoids harboring mutations in the *Tgfb* pathway. In this manuscript the authors document enhanced metastatic potential of AKP *Atrx* KO organoids that were injected in the tail vein or intrasplenically. It remains unclear whether loss of *Atrx* can promote metastatic spread from primary tumors. Thus, an orthotopic model should be developed to address this question.

The reviewer makes a valid point regarding the metastatic models used in the original study. To address this, we carried out orthotopic transplantation of our AKP and AKP *Atrx*^{KO} organoids into the colon. We found that metastatic spread from primary tumours can be promoted by deletion of *Atrx*, with 70% of AKP *Atrx*^{KO} tumours showing metastatic progression (compared to 0% for controls) (Extended Data Figure 7, pages 5-6) confirming the findings from our intrasplenic and tail vein injection models.

In silico analysis in Fig 5 indicates that alterations in *Atrx* are associated with iCMS3 tumours, squamous gene expression profiles, and poor prognosis. Although these data are consistent with functional data in murine AKP organoids, the authors need to further validate these by deleting *Atrx* in a human CRC organoid model.

To address this, we deleted *ATRX* in a human CRC organoid model. We found deletion of *ATRX* led to reduced expression of colonic epithelial genes (*HNF4A* and *CDX1*) and increased expression of *KRT5* (Extended Data Figure 21b, page 14). Additionally, treatment of *ATRX* deleted organoids with TGF β led to emergence of spindle-like morphology, similar to that found in our mouse models (Extended Data Figure 21c-21d, page 14).

Figure 5 shows that *Atrx* mutations are more frequently associated with iCMS3 type tumors that often harbor mutations in *Tgfb* pathway and *Braf* mutations. Yet all functional experiments were done in the AKP model, which resembles iCMS2 tumors. What is the impact of *Atrx* loss in *Tgfb*/*Braf* mutant organoids? Related to this question, the *Tgf*-beta pathway is associated with induction of EMT phenotypes as stated in the main text, but it is also potent growth suppressor and for this reason mutations in this pathway are required for tumor progression. What effect does *Tgf*-beta stimulation have on the growth of *Atrx* wt vs KO AKP organoids? Are mutant organoids no longer sensitive to the growth suppressive effects of *Tgf*-beta as would be expected for AKP organoids? This could provide a mechanism underlying their enhanced metastatic potential. A more thorough analysis of the impact of *Tgf*-beta on AKP *Atrx* KO organoids is needed.

To extend the findings of our study and more directly link it to the iCMS3/CMS4 tumour phenotype we have deleted *Atrx* in the *Braf*^{V600E} *P53*^{fl/fl} *Notch*^{1CD} (BPN) mouse colorectal cancer

model. This Braf driven model is transcriptionally distinct from the AKP model, resembling the CMS4 CRC subtype (Torang et al., 2025. PMID: 39747069). We find *Atrx* deletion in this model increases the induction of TGF β mediated EMT (Extended Data Figure 8, pages 5-6) and in an orthotopic transplantation model, BPN *Atrx*^{KO} tumours completely lose their glandular morphology, exhibiting a poorly differentiated phenotype (Extended Data Figure 8, pages 5-6). *Atrx* deletion also leads to metastatic progression in ~25% of mice (Extended Data Figure 8, pages 5-6). In addition, similar to the AKP model, *Atrx* deletion leads to loss of colonic gene expression (*Cdx1*, *Cdx2*, *Hnf4a*) and increased expression of EMT markers (*Twist1*) (Extended Data Figure 9, page 7). Therefore, *Atrx* deletion in the BPN model recapitulates the results of the AKP model, validating our key findings in an additional, iCMS3/CMS4 like CRC mouse model.

To address the impact of *Atrx* deletion in *Tgfbr* mutant organoids, we deleted *Tgfbr2* from our AKP *Atrx*^{KO} model. This led to reduced expression of squamous (*Krt5*, *Ly6d*) and EMT (*Twist1*, *Snai2*) marker genes and decreased the proportion of LY6D+ and ITGA5+ cells in this model (Extended Data Figure 16, pages 10-11). It also reduced the ability of *Atrx*^{KO} cells to undergo EMT (Extended Data Figure 16, pages 10-11), suggesting TGF β signalling plays a role in mediating cell plasticity in this model.

We also investigated the effects of TGF β stimulation on organoid growth, finding deletion of *Atrx* partially suppressed the growth suppression effects seen in AKP organoids (Extended Data Figure 16a-16b, page 10). Together, these results suggest that *Atrx* loss promotes metastasis by enabling organoid growth in the context of active TGF β signalling.

We thank the reviewers again for taking the time to review our manuscript and are very pleased they now support publication of our work. As requested by Referee #3 we have updated the requested figure as outlined below.

Referee #1 (Remarks to the Author):

The authors have addressed all my criticisms satisfactorily, and the manuscript has been substantially strengthened. This study presents important findings that contribute to our understanding of CRC plasticity, and in my opinion, it is now ready for publication.

Eduard Batlle

Referee #2 (Remarks to the Author):

The authors have addressed all my concerns in the revised manuscript.

Referee #3 (Remarks to the Author):

The authors have satisfactorily addressed my concerns.

I only have a minor comment regarding Extended Fig 16a, b. The graph in b apparently shows that Atrx KO organoids are resistant to the growth suppressive effects of Tgfb. But one wonders how this data was compiled. The representative images shown in a are very unclear and do not show any difference between mutants and controls. Higher magnification images should be shown.

If this issue can be addressed, I would support publication of this manuscript.

We apologise for the poor image quality and have replaced the images with higher magnification ones that depict the growth suppressive effects of Tgfb in AKP control organoids more clearly. These data can now be found in Figure Extended Data 8a.